# Causality-Induced Positional Encoding for Transformer-Based Representation Learning of Non-Sequential Features

**Kaichen Xu**[1][*], **Yihang Du**[2], **Mianpeng Liu**[2], **Zimu Yu**[2], **Xiaobo Sun**[3][*][†]

[1] Department of Computer Science, Emory University
[2] School of Statistics and Mathematics, Zhongnan University of Economics and Law
[3] School of Medicine, Department of Human Genetics, Emory University

## Abstract

Positional encoding is essential for supplementing transformer with positional information of tokens. Existing positional encoding methods demand predefined token/feature order, rendering them unsuitable for real-world data with non-sequential yet causally-related features. To address this limitation, we propose **CAPE**, a novel method that identifies underlying causal structure over non-sequential features as a weighted directed acyclic graph (DAG) using generalized structural equation modeling. The DAG is then embedded in hyperbolic space where its geometric structure is well-preserved using a hyperboloid model-based approach that effectively captures two important causal graph properties (**causal strength** & **causal specificity**). This step yields causality-aware positional encodings for the features, which are converted into their rotary form for integrating with transformer's self-attention mechanism. Theoretical analysis reveals that CAPE-generated rotary positional encodings possess three valuable properties for enhanced self-attention, including *causal distance-induced attenuation, causal generality-induced attenuation*, and *robustness to positional disturbances*. We evaluate CAPE over both synthetic and real-word datasets, empirically demonstrating its theoretical properties and effectiveness in enhancing transformer for data with non-sequential features. Our code is available at `https://github.com/Catchxu/CAPE`.

## 1 Introduction

Transformer [1] has become the cornerstone in modern deep learning models, powering advances in natural language processing (NLP) [2–5], computer vision [6–8], speech and audio processing [9, 10], and multimodal learning [11–13]. At the core of transformer is the self-attention mechanism, which effectively captures dependencies among sequential elements. However, this mechanism is inherently position-agonistic and permutation-invariant [14]. Positional information is crucial in learning semantics because it encodes sequential dependencies analogous to directional *causal structure* [3], as opposed to the undirected associations captured by self-attention. To inject positional information into the transformer architecture, a plethora of strategies have been proposed to generate positional encodings that are integrated with contextual embeddings. These approaches include fixed sinusoidal functions [1], trainable absolute or relative positional encodings [2, 5, 15–17], and more recent rotary positional encodings (RoPE) [18, 19]. Notably, these methods generally assume natural, inherent ordering in the data, such as the sequence of words in a sentence or the spatial arrangement of image patches.

---

[*]These authors contributed equally.
[†]Correspondence: `xsun28@emory.edu`

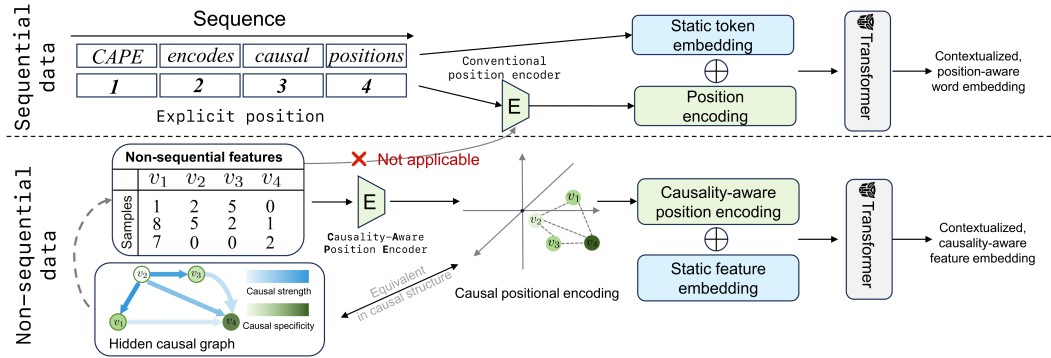

Figure 1: Causal position information can be utilized in place of explicit position information for transformer-based representation learning of data with non-sequential yet causally-related features.

However, this assumption breaks down in the context of many real-world datasets, where rows represent independent observations, columns represent non-sequential features, and entries capture the quantity or state of each feature for a given observation. For example, in biomedical sciences, multi-omics data, such as transcriptomics [20] and proteomics [21], measure gene and protein expression levels within samples. The expressions of genes and proteins lack a predefined sequence, despite their intricate causal order. Similarly, economic studies often involve non-sequential but causally linked economic indicators collected across different regions or countries. Therefore, existing positional encoding methods struggle to capture these underlying causal structures, limiting the applicability of transformers to such data. In response, some preliminary efforts have emerged within specialized domains. For example, in single-cell transcriptomics, transformer-based foundational models seek to generate distributed gene representations from large corpus of scRNA-seq datasets. To impose a pseudo-order on genes, some of these models [22, 23] organize trainable gene embeddings into sequences based on their expression levels, while others [24, 25] use pretrained, static gene embeddings as pseudo-positional encodings. However, a critical limitation of these methods is that they neglect the underlying causal structure among genes.

In this study, we propose **C**ausality-**A**ware **P**osition **E**ncoder (**CAPE**), a novel method for generating causality-aware positional encodings that extend the transformer architecture to data with non-sequential yet causally-related features. Initially, CAPE leverages a generalized structural equation model (SEM) to model the hidden causal structure among features as a weighted directed acyclic graph (DAG), which is efficiently identified through neural variational inference with a constraint-based, continuous optimization technique [26–28]. Next, inspired by the theory of special relativity, which links causal connections between events to their relative positions in hyperbolic spacetime [29, 30], we utilize the hyperboloid model[3] to embed the DAG into the hyperbolic space, which is known for its ability to model tree-like networks commonly seen in DAGs [31]. Specifically, nodes in the DAG are represented as points on a Riemannian manifold, with their positions learned through regularized graph contrastive learning, optimized via Riemannian stochastic gradient descent (RSGD) [32]. This approach ensures that the learned embeddings capture two critical causal graph properties, including **causal strength** and **causal specificity** (see Section 3.4) [33], thus preserving the original causal structure of the DAG. Finally, CAPE converts the hyperbolic positional encodings into rotary form, a causality-induced version of RoPE [18]. This form offers several key benefits, including compatibility with linear self-attention [18] and enhanced understanding of contextual knowledge [19]. We further theoretically demonstrate that the causality-aware, rotary positional encodings offers three valuable properties in computing self-attention: Attention strength attenuates with increasing causal distance (*causal distance-induced attention attenuation*, Section 4.1 ) or decreasing causal specificity (*causal generality-induced attention attenuation*, Section 4.2), and attention scores exhibit *robustness to positional disturbances* (Section 4.3). In summary, our main contributions include:

- We propose CAPE, a novel method for generating causality-aware positional encodings for data with non-sequential yet causally-related features. It eliminates the need for predefined feature ordering required by conventional positional encoding methods, while incorporating causal structure information into transformer-based representation learning.

---

[3]https://en.wikipedia.org/wiki/Hyperboloid_model

- CAPE adopts a hyperboloid model-based approach to embed causal graphs, effectively capturing two fundamental causal graph properties: causal strength and causal specificity.
- We theoretically demonstrate that CAPE-generated rotary positional encodings possess several valuable properties that enhance the effectiveness of self-attention.
- We empirically validate CAPE's theoretical properties using synthetic data, and evaluate its effectiveness in enhancing representation learning of non-sequential data using various real-world multi-omics datasets.

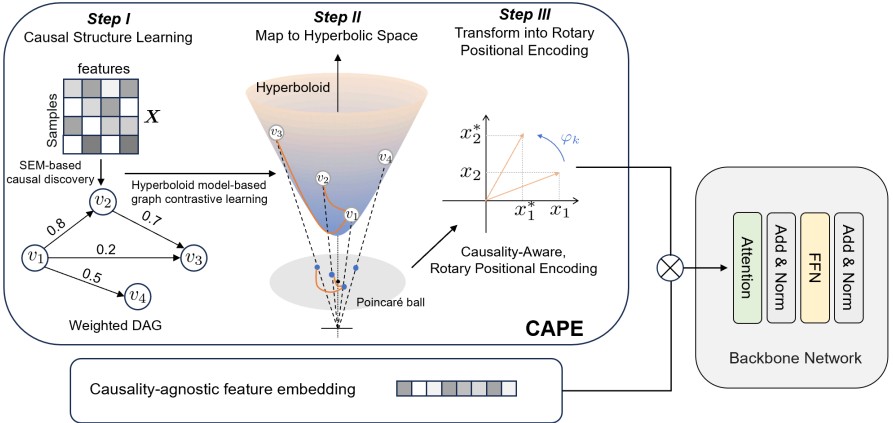

Figure 2: Overview of CAPE.

## 2 Related Work

We postpone this section to Supp. C due to limited space.

## 3 Methodology

### 3.1 Preliminary

Let $\mathcal{V} = \{v_j\}_{j=1}^M$ be a sequence of $M$ input tokens (e.g., words and image patches). The conventional procedure of applying a standard transformer [1] to $\mathcal{V}$ can be described as:

$$\widehat{\boldsymbol{v}}_1, \cdots, \widehat{\boldsymbol{v}}_M = \mathcal{T}(\mathcal{A}(\mathcal{F}(\boldsymbol{v}_1, \boldsymbol{\varphi}_1), \cdots, \mathcal{F}(\boldsymbol{v}_M, \boldsymbol{\varphi}_M))). \tag{1}$$

Here, $\forall j \in [1 \cdots M]$, $\widehat{\boldsymbol{v}}_j$ and $\boldsymbol{v}_j$ represent the position-aware, contextualized embedding and static, pretrained embedding of token $v_j$, respectively. $\boldsymbol{\varphi}_j$ is the positional encoding of the $j$-th position. $\mathcal{F}$ denotes the function for fusing $\boldsymbol{v}$ and $\boldsymbol{\varphi}$, $\mathcal{A}$ represents the self-attention function, and $\mathcal{T}$ is the transformer function.

When $\mathcal{V}$ consists of non-sequential features, $\boldsymbol{\varphi}$ cannot be directly derived from a predefined sequential order. In such cases, we assume that $\{v_j\}_{j=1}^M$ are causally related and organized into a tabular measurement dataset $\boldsymbol{X} \in \mathbb{R}^{N \times M}$, where row $\boldsymbol{x}_i$ represents the $i$-th observation, the $j$-th column corresponds to $v_j$, and $X_{ij}$ denotes the quantity of $v_j$ measured in $\boldsymbol{x}_i$. For example, $v_j$ might represent gene $j$ and $X_{ij}$ the read counts of gene $j$ in cell $i$. We aim to derive causality-aware positional encodings from $\boldsymbol{X}$ as:

$$\{\boldsymbol{\varphi}_{v_j}\}_{j=1}^M := \mathcal{P}(\boldsymbol{X}), \tag{2}$$

where $\mathcal{P}$ denotes the causality-aware positional encoding function. We then inject $\{\boldsymbol{\varphi}_{v_j}\}_{j=1}^M$ into the transformer architecture as:

$$\widehat{\boldsymbol{v}}_1^i, \cdots, \widehat{\boldsymbol{v}}_M^i = \mathcal{T}(\mathcal{A}(\mathcal{F}(\mathcal{G}(\boldsymbol{v}_1, \boldsymbol{x}_i), \boldsymbol{\varphi}_{v_1}), \cdots, \mathcal{F}(\mathcal{G}(\boldsymbol{v}_M, \boldsymbol{x}_i), \boldsymbol{\varphi}_{v_M}))), \quad \forall i = 1, 2, \cdots, N \tag{3}$$

where $\widehat{\boldsymbol{v}}_j^i$ represents the contextualized, causality-aware embedding of $v_j$ within the $i$-th observation. $\mathcal{G}$ is a function to generate contextualized, causality-agonistic intermediate feature embeddings (see

Supp. F.2 for examples of $\mathcal{G}$). We can further obtain observation-level embeddings as:

$$\boldsymbol{h}^i = \text{Agg}(\widehat{\boldsymbol{v}}_1^i, \cdots, \widehat{\boldsymbol{v}}_M^i), \tag{4}$$

where Agg denotes an aggregate function (e.g., mean or max pooling), $\boldsymbol{h}^i$ the embedding of $i$-th observation. These contextualized, causality-aware feature embeddings and observation-level embeddings can be used in downstream tasks for improved performance, as shown in Section 5.2.

## 3.2 Methodology Overview

As stated in the previous section, given a set of non-sequential, causally-related features $\mathcal{V}$ and their associated measurement data $\boldsymbol{X}$, our goal is to generate contextualized, causality-aware positional encoding $\boldsymbol{\varphi}_{v_j} \in \mathbb{R}^d$ for each $v_j \in \mathcal{V}$. To this end, CAPE introduces an integrated three-step framework. In *Step I* (Section 3.3), CAPE identifies the causal structure over $\mathcal{V}$ as a weighted DAG $G(\mathcal{V}, \mathcal{E})$, where $\mathcal{V}$ is the node set representing features, $\mathcal{E}$ is the edge set representing causal relationships, and the edge weights quantify causal strengths. The

Table 1: Summary of main notations.

| Notations | Descriptions |
|---|---|
| $N$ | Number of observations. |
| $M$ | Number of variables. |
| $\boldsymbol{A} \in \mathbb{R}^{M \times M}$ | Adjacency matrix of causal graph. |
| $d$ | Dimensionality of positional encoding. |
| $D$ | Dimensionality of feature embedding. |
| $\boldsymbol{p}_{v_j} \in \mathbb{R}^{d+1}$ | Hyperbolic embedding. |
| $\boldsymbol{e}_{v_j} \in \mathbb{R}^d$ | Poincaré ball embedding. |
| $\boldsymbol{\varphi}_{v_j} \in \mathbb{R}^d$ | Rotary positional encoding. |
| $\boldsymbol{v}_j \in \mathbb{R}^D$ | Static feature embedding. |

presence and weights of edges are inferred using a non-linear SEM that captures complex, potentially non-linear causal dependencies. In *Step II* (Section 3.4), the identified $G$ is embedded in hyperbolic space using the hyperboloid model, assigning each $v_j \in \mathcal{V}$ a positional embedding $\boldsymbol{p}_{v_j} \in \mathbb{R}^{d+1}$, where $d + 1$ is the dimensionality of the hyperbolic space. These embeddings are optimized using a regularized graph contrastive loss that accounts for both causal strength and causal specificity, effectively preserving the original causal structure of $G$. In *Step III* (Section 3.5), hyperboloid positional embeddings are mapped into a unit Poincaré ball via diffeomorphism before being transformed into their rotary form for modulating feature-wise attention scores in the transformer. We will elaborate each of these steps in the following sections.

## 3.3 Causal Structure Learning (Step I)

CAPE infers the causal structure over $\mathcal{V}$ using a generalized nonlinear SEM defined as:

$$f(\boldsymbol{X}) = f(\boldsymbol{X})\boldsymbol{A} + g(\widetilde{\boldsymbol{Z}}), \tag{5}$$

where $f : N \times M \to N \times M$ is a nonlinear function that models the functional relationships among observed features (endogenous variables), $\boldsymbol{A} \in \mathbb{R}^{M \times M}$ is a directed, weighted adjacency matrix representing the hidden causal graph $G(\mathcal{V}, \mathcal{E})$, and $g : N \times M \to N \times M$ models the distribution of unseen noises $\widetilde{\boldsymbol{Z}} \in \mathbb{R}^{N \times M}$ (exogenous variables). Given the universal approximation theorem [34], we let $f$ be a multi-layer perceptron (MLP), and $g(\widetilde{\boldsymbol{Z}}) = \boldsymbol{Z} \in \mathbb{R}^{N \times M} \sim \mathcal{N}(\boldsymbol{0}, \boldsymbol{I})$ for simplicity, where $\mathcal{N}$ denotes the Gaussian distribution. Then Eq. (5) can be reformed as:

$$\boldsymbol{Z} = \text{Enc}(\boldsymbol{X}|\boldsymbol{A}, \boldsymbol{W_e}) \coloneqq f(\boldsymbol{X})(\boldsymbol{I} - \boldsymbol{A}), \tag{6}$$

$$\boldsymbol{X} = \text{Dec}(\boldsymbol{Z}|\boldsymbol{A}, \boldsymbol{W_d}) \coloneqq f^{-1}\left(\boldsymbol{Z}(\boldsymbol{I} - \boldsymbol{A})^{-1}\right), \tag{7}$$

where Enc acts as an encoder mapping $\boldsymbol{X}$ to Gaussian noises $\boldsymbol{Z}$, while Dec serves as a decoder that recovers $\boldsymbol{X}$ from $\boldsymbol{Z}$, similar to the encoder and decoder of a variational autoencoder (VAE) [35]. Here, $\boldsymbol{A}, \boldsymbol{W_e}, \boldsymbol{W_d}$ can be estimated using a variational inference approach, with an evidence lower bound (ELBO) training objective [27]:

$$\mathcal{L}_{\text{ELBO}} = \underbrace{\mathbb{E}_{q(\boldsymbol{Z}|\boldsymbol{X})}\left[\log p(\boldsymbol{X}|\boldsymbol{Z})\right]}_{\text{Reconstruction Loss}} - \text{KL}\left[q(\boldsymbol{Z}|\boldsymbol{X}) \| p(\boldsymbol{Z})\right], \tag{8}$$

where $p(\boldsymbol{X}|\boldsymbol{Z})$ is the reconstruction distribution, $q(\boldsymbol{Z}|\boldsymbol{X})$ is the variational posterior, and $p(\boldsymbol{Z})$ is the prior distribution. We also impose a regularization term $\|\boldsymbol{A}\|_1$ to encourage sparsity, and a smooth

constraint $h(\boldsymbol{A}) : \mathrm{tr}\left(e^{\boldsymbol{A} \odot \boldsymbol{A}}\right) - M = 0$ to ensure the acyclicity of $\boldsymbol{A}$ [26], where $\odot$ denotes the element-wise product (see Supp. A.1). The overall loss function then reads:

$$\min_{\boldsymbol{W_e}, \boldsymbol{W_d}, \boldsymbol{A}} \; -\mathcal{L}_{\mathrm{ELBO}} + \lambda_{\mathrm{s}} \|\boldsymbol{A}\|_1 \quad \mathrm{s.t.} \quad h(\boldsymbol{A}) = 0, \tag{9}$$

where $\lambda_{\mathrm{s}} \geq 0$ is the regularization coefficient.

To solve Eq. (9) efficiently, we employ the augmented Lagrangian method [36], yielding the following unconstrained subproblem:

$$\min_{\boldsymbol{W_e}, \boldsymbol{W_d}, \boldsymbol{A}} \; \mathcal{L}_{\mathrm{DAG}} \coloneqq -\mathcal{L}_{\mathrm{ELBO}} + \lambda_{\mathrm{s}} \|\boldsymbol{A}\|_1 + \frac{\rho}{2}|h(\boldsymbol{A})|^2 + \alpha h(\boldsymbol{A}), \tag{10}$$

where $\alpha$ is the Lagrange multiplier and $\rho > 0$ is the penalty parameter. The optimization proceeds by alternating updates [27]. Finally, a threshold $\tau > 0$ is applied to $\boldsymbol{A}$ to prune noisy, false-positive causal edges, as $\boldsymbol{A} \leftarrow \boldsymbol{A} \odot \mathbb{I}(|\boldsymbol{A}| > \tau)$. See Supp. G.3 for sensitivity analysis of $\tau$.

### 3.4 Mapping Causal Structure to Hyperbolic Space (Step II)

To translate the identified causal graph into spatial positions while preserving its geometric structure, we project $G(\mathcal{V}, \mathcal{E})$ into a hyperbolic space using the hyperboloid model. Specifically, each node $v_j \in \mathcal{V}$ is assigned an embedding $\boldsymbol{p}_{v_j} \in \mathbb{R}^{d+1}$ in a Riemannian manifold $\mathcal{L}^d \coloneqq (\mathcal{H}^d, \boldsymbol{g}_l)$, where $\boldsymbol{g}_l \coloneqq \mathrm{diag}(-1, 1, \cdots, 1) \in \mathbb{R}^{(d+1) \times (d+1)}$ denotes the metric tensor and where

$$\mathcal{H}^d \coloneqq \{\boldsymbol{p} \coloneqq (p^{(0)}, \widetilde{\boldsymbol{p}}) \in \mathbb{R}^{d+1} : \langle \boldsymbol{p}, \boldsymbol{p} \rangle_l = -1, p^{(0)} > 0\}, \tag{11}$$

$$\langle \boldsymbol{p}_{v_m}, \boldsymbol{p}_{v_n} \rangle_l \coloneqq \boldsymbol{p}_{v_m}^{\top} \boldsymbol{g}_l \, \boldsymbol{p}_{v_n} = -p_{v_m}^{(0)} p_{v_n}^{(0)} + \widetilde{\boldsymbol{p}}_{v_m}^{\top} \widetilde{\boldsymbol{p}}_{v_n}, \tag{12}$$

denotes the upper sheet of a two-sheeted hyperboloid with an origin $\boldsymbol{p_o} = (1, \boldsymbol{0}_d)^{\top}$ in a $(d+1)$-dimensional Minkowski space [37]. The distance between two points $\boldsymbol{p}_{v_m}, \boldsymbol{p}_{v_n} \in \mathcal{H}^d$ reads $d_l(\boldsymbol{p}_{v_m}, \boldsymbol{p}_{v_n}) = \mathrm{arcosh}(-\langle \boldsymbol{p}_{v_m}, \boldsymbol{p}_{v_n} \rangle_l)$ [38], based on which two critical causal graph properties are defined as:

$$\textbf{Causal strength} : \sigma(v_m, v_n) \propto \frac{1}{d_l(\boldsymbol{p}_{v_m}, \boldsymbol{p}_{v_n})}, \tag{13}$$

$$\textbf{Causal specificity} : \ell(v_m) \propto d_l(\boldsymbol{p}_{v_m}, \boldsymbol{p_o}) = p_{v_m}^{(0)} = \sqrt{1 + \|\widetilde{\boldsymbol{p}}_{v_m}\|}, \tag{14}$$

where $\|\cdot\|$ denotes the Euclidean norm. Intuitively, as shown in Fig. 2, the strength of the causal relationship between $v_m$ and $v_n$ attenuates as their hyperbolic distance increases, reflecting their weaker connection in the causal graph. Meanwhile, since hyperbolic space can be thought of as a continuous analogue to discrete trees with roots near the origin [39], a causally general feature (e.g., a root feature that is causally related to many its causal descendants in the causal graph) should poise close to the origin. To implement these properties into the positional encodings, we adopt a regularized graph contrastive learning framework, with the objective:

$$\min_{\boldsymbol{p}_{v_1}, \dots, \boldsymbol{p}_{v_M} \in \mathcal{H}^d} \mathcal{L}_{\mathcal{H}} = \frac{1}{M} \sum_{j=1}^{M} \mathcal{L}_{\mathrm{con}}(\boldsymbol{p}_{v_j}) + \lambda_{\mathrm{g}} \Omega(\boldsymbol{p}_{v_j}), \tag{15}$$

$$\mathcal{L}_{\mathrm{con}}(\boldsymbol{p}_{v_m}) = -\sum_{n \in N_k^+(v_m)} |\boldsymbol{A}_{mn}| \log \frac{e^{-d_l(\boldsymbol{p}_{v_m}, \boldsymbol{p}_{v_n})}}{e^{-d_l(\boldsymbol{p}_{v_m}, \boldsymbol{p}_{v_n})} + \sum_{n' \in N_k^-(m)} e^{-d_l(\boldsymbol{p}_{v_m}, \boldsymbol{p}_{v_{n'}})}}, \tag{16}$$

$$\Omega(\boldsymbol{p}_{v_m}) = \boldsymbol{\pi}_{v_m} d_l(\boldsymbol{p}_{v_m}, \boldsymbol{p_o}), \tag{17}$$

where $\mathcal{L}_{\mathrm{con}}$ is the contrastive term, $\Omega$ is the regularization term, and $\lambda_g$ is the regularization weight. In $\mathcal{L}_{\mathrm{con}}$, $N_k^+(v_m) \subset \mathcal{V}$ denotes the set of positive samples of $v_m$, consisting of those connected to $v_m$ via $k$-hop ($k$ defaults to 2, see sensitivity analysis in Supp. G.3) causal paths in $G$, while $N_k^-(v_m) = \mathcal{V} \setminus N_k^+(m)$ denotes its set of negative samples. $|\boldsymbol{A}_{mn}|$ reflects the estimated causal strength between $v_m$ and $v_n$. By pulling closer features with strong causal relationships (positive pairs), while distancing those with weak causal relationships (negative pairs), $\mathcal{L}_{\mathrm{con}}$ ensures the causal strength property. In $\Omega$, $\boldsymbol{\pi}_{v_m} \in \mathbb{R}^+$ is the $m$-th value in the *PageRank* vector $\boldsymbol{\pi} \in (0, 1)^M$ as:

$$\boldsymbol{P}^{\top} \coloneqq |\boldsymbol{A}| \boldsymbol{D}_{\mathrm{in}}^{-1}, \quad \widehat{\boldsymbol{P}} \coloneqq (1 - w)\boldsymbol{P} + w\frac{\boldsymbol{1}}{M},$$

$$\widehat{\boldsymbol{P}}^{\top} \boldsymbol{\pi} = \boldsymbol{\pi} \quad \mathrm{s.t.} \quad \boldsymbol{\pi}^{\top} \overline{\boldsymbol{1}}_M = 1, \tag{18}$$

where $\boldsymbol{D}_{\text{in}}$ denotes the diagonal in-degree matrix of $|A|$, $\boldsymbol{P}$ is the in-degree normalized transition matrix, and $\frac{1}{M} := [\frac{1}{M}]^{M \times M}$ is the transition restart matrix to ensure $\widehat{\boldsymbol{P}}$ is strongly connected and an ergodic Markov chain[4]. $w \in (0, 1)$ is the relative weight for the restart matrix. $\boldsymbol{\pi}$ is essentially a steady-state probability vector with larger values for more causally general nodes (e.g., those with more outgoing edges[5]). Consequently, features with larger causal generality are more penalized by $\Omega$, forcing their positions to be close to the origin $\boldsymbol{p_o}$, thus ensuring the causal specificity property.

To minimize $\mathcal{L}_{\mathcal{H}}$ in Eq. (15), we use RSGD to update $\boldsymbol{p}_{v_j}$ in iterations. Specifically, we first compute the Euclidean gradient $\nabla^{\mathbb{E}}_{\boldsymbol{p}_{v_j}} \mathcal{L}_{\mathcal{H}}$ at $\boldsymbol{p}_{v_j}$, which is then converted into Riemannian gradient as:

$$\nabla^{\mathbb{R}}_{\boldsymbol{p}_{v_j}} \mathcal{L}_{\mathcal{H}} = \boldsymbol{g}_l^{-1} \nabla^{\mathbb{E}}_{\boldsymbol{p}_{v_j}} \mathcal{L}_{\mathcal{H}}. \tag{19}$$

Next, the Riemannian gradient direction is projected onto the tangent space at $\boldsymbol{p}_{v_j}$ via an orthogonal projection (see Prop. A.4) as:

$$\nabla^{\mathbb{T}}_{\boldsymbol{p}_{v_j}} \mathcal{L}_{\mathcal{H}} = \text{proj}_{\boldsymbol{p}_{v_j}} \left( \nabla^{\mathbb{R}}_{\boldsymbol{p}_{v_j}} \mathcal{L}_{\mathcal{H}} \right). \tag{20}$$

$\boldsymbol{p}_{v_j}$ is updated along the direction of $\nabla^{\mathbb{T}}_{\boldsymbol{p}_{v_j}} \mathcal{L}_{\mathcal{H}}$ in the tangent space with a learning rate of $\eta > 0$, and then retracted onto the hyperboloid via an exponential map function (see Prop. A.3) as:

$$\boldsymbol{p}_{v_j} \leftarrow \exp_{\boldsymbol{p}_{v_j}} \left( -\eta \cdot \nabla^{\mathbb{T}}_{\boldsymbol{p}_{v_j}} \mathcal{L}_{\mathcal{H}} \right). \tag{21}$$

Due to the closed-form computation of the geodesics on the hyperboloid, this optimization is computationally efficient (see Supp. A.2 for details).

### 3.5 Transforming Hyperbolic Positional Encoding to Rotary Form (Step III)

As demonstrated in [18] and [19], rotary positional encodings exhibit the advantages of compatibility with linear self-attention and enhanced understanding of contextual knowledge. Adherent to this notion, we first map the optimized positional encodings $\{\boldsymbol{p}_{v_j}\}_{j=1}^{M}$ from the hyperboloid into a Poincaré ball as $\{\boldsymbol{e}_{v_j}\}_{j=1}^{M}$ via the diffeomorphism $f_d : \mathcal{H}^d \to \mathcal{B}^d$ (see Supp. A.2.3 for details). The Poincaré ball is a Riemannian manifold $\mathcal{P}^d := (\mathcal{B}^d, \boldsymbol{g}_p)$, where $\mathcal{B}^d := \{\boldsymbol{e} \in \mathbb{R}^d : \|\boldsymbol{e}\| < 1\}$ represents the open $d$-dimensional unit ball and the metric tensor $\boldsymbol{g}_p = \left( \frac{2}{1-\|\boldsymbol{e}\|^2} \right)^2 \boldsymbol{I}$ is a conformal transformation of the Euclidean metric $\boldsymbol{I}$. This mapping is motivated by the fact that the Poincaré ball, with its spherical geometry centered at the origin $\boldsymbol{0_d}$, is more naturally suited for rotary encodings. Importantly, since it also represents a hyperbolic space, the two causal graph properties (**causal strength** and **causal specificity**) are preserved (see Eqs. (13) and (14)).

To transform the Poincaré ball embeddings $\{\boldsymbol{e}_{v_j}\}_{j=1}^{M}$ into their rotary form for injecting into the transformer architecture, we refine the standard query-key mapping and inner product used in the self-attention mechanism, in a way similar to RoPE[6] [18]:

$$\boldsymbol{q}_{v_m}^i = \mathcal{I}_q(\boldsymbol{v}_m^i, \boldsymbol{e}_{v_m}), \quad \boldsymbol{k}_{v_n}^i = \mathcal{I}_k(\boldsymbol{v}_n^i, \boldsymbol{e}_{v_n}), \tag{22}$$

$$\left\langle \boldsymbol{q}_{v_m}^i, \boldsymbol{k}_{v_n}^i \right\rangle = \mathcal{A} \left( \boldsymbol{v}_m^i, \boldsymbol{v}_n^i, \boldsymbol{\gamma}(\boldsymbol{e}_{v_m}, \boldsymbol{e}_{v_n}) \right), \tag{23}$$

where $\boldsymbol{q}_{v_m}^i$ and $\boldsymbol{k}_{v_n}^i$ are the query and key derived from $v_m$ and $v_n$ in the context of observation $\boldsymbol{x}_i$, respectively. $\boldsymbol{v}_j^i := \mathcal{G}(\boldsymbol{v}_j, \boldsymbol{x}_i) \in \mathbb{R}^D$ denotes the *contextualized, causality-agonistic* embeddings (see Section 3.1). $\mathcal{I}_q$ and $\mathcal{I}_k$ are functions that inject positional encodings into queries and keys, respectively. $\mathcal{A}$ is an attention scoring function that accounts for the *relative* causal position between $v_m$ and $v_n$, which is represented as $\boldsymbol{\gamma}(\boldsymbol{e}_{v_m}, \boldsymbol{e}_{v_n})$. Supp. A.3 gives the explicit solutions to $\mathcal{I}_q, \mathcal{I}_k$, and $\mathcal{A}$ as:

$$\begin{aligned} \mathcal{I}_q(\boldsymbol{v}_m^i, \boldsymbol{e}_{v_m}) &:= \boldsymbol{R}(\boldsymbol{\varphi}_{v_m}) \boldsymbol{W}_q \boldsymbol{v}_m^i = \boldsymbol{R}(\boldsymbol{\varphi}_{v_m}) \boldsymbol{q}_{v_m}^i, \\ \mathcal{I}_k(\boldsymbol{v}_n^i, \boldsymbol{e}_{v_n}) &:= \boldsymbol{R}(\boldsymbol{\varphi}_{v_n}) \boldsymbol{W}_k \boldsymbol{v}_n^i = \boldsymbol{R}(\boldsymbol{\varphi}_{v_n}) \boldsymbol{k}_{v_n}^i, \end{aligned} \tag{24}$$

---

[4]By the Perron-Frobenius theorem, the left eigenvector $\boldsymbol{\pi}$ of the largest eigenvalue ($\lambda_{max} = 1$) of $\widehat{\boldsymbol{P}}$ is unique.

[5]See Supp. B for a discussion of why we use $\boldsymbol{\pi}$ rather than out-degree

[6]Following RoPE, positional encodings are only injected into keys and queries, not values.

$$\mathcal{A}\left(\boldsymbol{v}_m^i, \boldsymbol{v}_n^i, \gamma(\boldsymbol{e}_{v_m}, \boldsymbol{e}_{v_n})\right) = (\boldsymbol{q}_{v_m}^i)^\top \boldsymbol{R}(\boldsymbol{\varphi}_{v_n} - \boldsymbol{\varphi}_{v_m})\boldsymbol{k}_{v_n}^i, \tag{25}$$

where $\boldsymbol{\varphi}_v := c\boldsymbol{e}_v$ denotes a vector whose components are used as the rotation angles in the subsequent rotary embedding, $c = \pi/4$ is a constant scale factor to control the range of angles[7], and $\gamma(\boldsymbol{e}_{v_m}, \boldsymbol{e}_{v_n}) := \boldsymbol{e}_{v_m} - \boldsymbol{e}_{v_n}$. $\boldsymbol{R}(\boldsymbol{\varphi}_v)$ is a rotation matrix induced by $\boldsymbol{\varphi}_v$:

$$\boldsymbol{R}(\boldsymbol{\varphi}_v) := \begin{bmatrix} \boldsymbol{r}(\varphi_v^{(1)}) & \boldsymbol{0} & \cdots & \boldsymbol{0} \\ \boldsymbol{0} & \boldsymbol{r}(\varphi_v^{(2)}) & \cdots & \boldsymbol{0} \\ \vdots & \vdots & \ddots & \boldsymbol{0} \\ \boldsymbol{0} & \boldsymbol{0} & \cdots & \boldsymbol{r}(\varphi_v^{(d)}) \end{bmatrix}, \quad \boldsymbol{r}(\varphi_v^{(t)}) := \begin{bmatrix} \cos(\varphi_v^{(t)}) & -\sin(\varphi_v^{(t)}) \\ \sin(\varphi_v^{(t)}) & \cos(\varphi_v^{(t)}) \end{bmatrix}, \tag{26}$$

where $d = \frac{D}{2}$.

## 4 Theoretical Properties of Causality-Aware, Rotary Positional Encoding

### 4.1 Causal Distance-Induced Attention Attenuation

As demonstrated in Prop. 4.1 and Remark 4.1, injecting CAPE-generated positional encodings into the self-attention calculation allows two features to be assigned reduced attention scores when they are causally distant.

**Proposition 4.1.** *Given $\boldsymbol{v}_m^i$ and $\boldsymbol{v}_n^i$, the attention scoring function $\mathcal{A}$ in Eq. (25) is bounded by $\mathcal{A}^+ > 0$ and $\mathcal{A}^- < 0$, satisfying:*

$$\frac{\partial \mathcal{A}^+\left(\boldsymbol{v}_m^i, \boldsymbol{v}_n^i, \boldsymbol{e}_{v_m} - \boldsymbol{e}_{v_n}\right)}{\partial d_p(\boldsymbol{e}_{v_m}, \boldsymbol{e}_{v_n})} \leq 0, \quad \frac{\partial \mathcal{A}^-\left(\boldsymbol{v}_m^i, \boldsymbol{v}_n^i, \boldsymbol{e}_{v_m} - \boldsymbol{e}_{v_n}\right)}{\partial d_p(\boldsymbol{e}_{v_m}, \boldsymbol{e}_{v_n})} \geq 0, \tag{27}$$

*where $d_p(\boldsymbol{e}_{v_m}, \boldsymbol{e}_{v_n})$ is the distance between $\boldsymbol{e}_{v_m}$ and $\boldsymbol{e}_{v_n}$ on the Poincaré ball manifold, computed as:*

$$d_p(\boldsymbol{e}_{v_m}, \boldsymbol{e}_{v_n}) = \text{arcosh}\left(1 + 2\frac{\|\boldsymbol{e}_{v_m} - \boldsymbol{e}_{v_n}\|^2}{(1 - \|\boldsymbol{e}_{v_m}\|^2)(1 - \|\boldsymbol{e}_{v_n}\|^2)}\right). \tag{28}$$

*The functions $\mathcal{A}^+$ and $\mathcal{A}^-$ are given in Supp. A.4.*

*Proof.* See Supp. A.4. $\qquad\square$

**Remark 4.1.** *As the causal distance $d_p(\boldsymbol{e}v_m, \boldsymbol{e}v_n) \to +\infty$, both $\mathcal{A}^+$ and $\mathcal{A}^-$ attenuate and converge towards smaller magnitudes (though not necessarily to 0). Since $\mathcal{A}$ is bounded between $\mathcal{A}^+$ and $\mathcal{A}^-$, its range of possible variation also shrinks significantly.*

### 4.2 Causal Generality-Induced Attention Attenuation

As discussed in Section 3.5, the causal specificity of $v_m$ increases with $\|\boldsymbol{e}_{v_m}\|$ in hyperbolic space. Here, we further define the causal generality in unit Poincaré ball manifold below.

**Definition 4.1** (**Causal generality in unit Poincaré ball manifold**). *Given a causal graph $G(\mathcal{V}, \mathcal{E})$ embedded in the Poincaré ball manifold $\mathcal{P}^d := (\mathcal{B}^d, \boldsymbol{g}_p)$, where $\mathcal{B}^d := \{\boldsymbol{e} \in \mathbb{R}^d : \|\boldsymbol{e}\| < 1\}$, the causal generality of a node $v_m \in \mathcal{V}$ is defined as $\psi_{v_m} := 1 - \|\boldsymbol{e}_{v_m}\|$.*

Causally general features (e.g., those that influence many other features) distribute their attention more broadly, resulting in lower attention scores for each of their individual causal descendants. Consequently, their attention scores tend to span a narrower range compared to more causally specific features. For example, both the "Big Bang" and "amino acids" are causes of the "emergence of life", but the former is a more causally general event, as it represents the origin of all things. Therefore, the "Big Bang" should receive less attention than "amino acids" when reasoning about the origins of life. This property emerges from the attention scores computed with CAPE-generated rotary positional encodings, as formally demonstrated in Prop. 4.2 and Remark 4.2.

---

[7]$c$ is set to be $\pi/4$ to make $\boldsymbol{\varphi}_k$ a small but not negligible angle

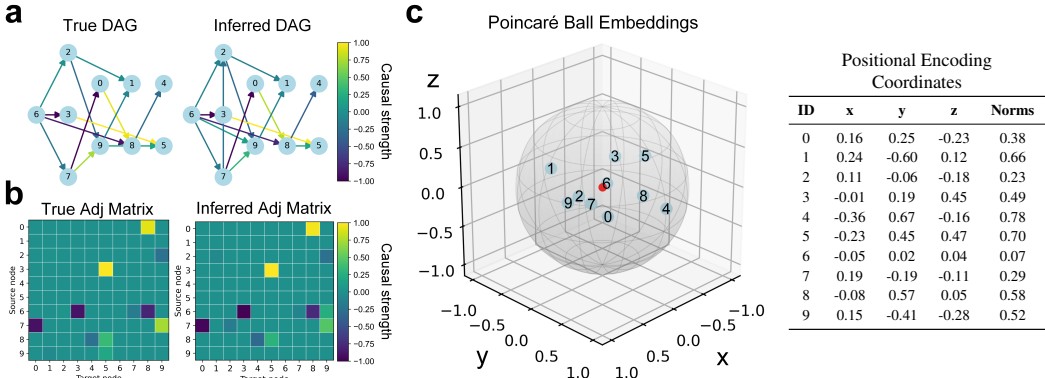

Figure 3: (a) True and inferred DAGs on synthetic data. (b) True and inferred adjacency matrices. (c) 3D visualization of Poincaré ball embeddings of nodes in $\mathcal{P}^3$.

**Proposition 4.2.** *Given $\boldsymbol{v}_m^i$, $\boldsymbol{v}_n^i$, and fixed causal distance $d_p(\boldsymbol{e}_{v_m}, \boldsymbol{e}_{v_n})$ in the Poincaré ball manifold defined in Def. 4.1, $\mathcal{A}^+$ and $\mathcal{A}^-$ satisfy:*

$$\frac{\partial \mathcal{A}^+ \left(\boldsymbol{v}_m^i, \boldsymbol{v}_n^i, \boldsymbol{e}_{v_m} - \boldsymbol{e}_{v_n}\right)}{\partial \psi_{v_m}} \leq 0, \quad \frac{\partial \mathcal{A}^- \left(\boldsymbol{v}_m^i, \boldsymbol{v}_n^i, \boldsymbol{e}_{v_m} - \boldsymbol{e}_{v_n}\right)}{\partial \psi_{v_m}} \geq 0. \tag{29}$$

*The same holds for $\psi_{v_n}$.*

*Proof.* See Supp. A.5 □

**Remark 4.2.** *As the causal generality $\psi_{v_m} \to 1$, $\mathcal{A}$ 's upper boundary $\mathcal{A}^+$ and lower boundary $\mathcal{A}^-$ asymptotically attenuate towards fixed constants $a > 0$ and $-a < 0$, respectively (See Supp. A.5).*

### 4.3 Robustness to Positional Disturbances

In practice, the measurement data $\boldsymbol{X}$ (see Section 3.3) is often subject to measurement errors, leading to biased estimation of the causal structure and perturbed Poincaré ball positional encodings. The following proposition demonstrates the robustness of the resulting attention scores to such disturbances.

**Proposition 4.3.** *Assume that the noise-perturbed Poincaré ball positional encoding of $v_j$ can be represented as $\boldsymbol{e}'_{v_j} := \boldsymbol{e}_{v_j} + \boldsymbol{\varepsilon}_j$, where $\boldsymbol{\varepsilon}_j \sim \mathcal{N}(\boldsymbol{\mu}, \boldsymbol{I}_j)$ is a small random Gaussian disturbance with $\boldsymbol{\mu} \in \mathbb{R}^d$ and $\boldsymbol{I}_j = \mathrm{diag}(\sigma_{j1}^2, \sigma_{j2}^2, \cdots, \sigma_{jd}^2)$. Then, the noise-disturbed attention score $\mathcal{A}'$ remains robust to such disturbances in three aspects, including **Distinguishability** (Prop. A.8), **Unbiasedness** (Prop. A.9), and **Asymptotic Convergence** (Prop. A.10).*

*Proof.* See Supp. A.6. □

## 5 Experiments

Due to space constraints, we defer the dataset descriptions to Supp. D, and the specifics of the evaluation tasks and metrics to Supp. E. Implementation details, including data preprocessing, model architecture, and training procedures, are provided in Supp. F. Finally, Supp. G presents additional results, including a full empirical analysis of CAPE properties, comprehensive multi-omics benchmarks, ablation studies, sensitivity analysis, and complexity analysis.

### 5.1 Empirical Evaluation of CAPE's Properties

**CAPE effectively identifies the causal structure and preserves it in the hyperbolic manifold.** To facilitate this evaluation, we simulate a tabular dataset $\boldsymbol{X}_{\mathrm{syn}} \in \mathbb{R}^{5000 \times 10}$, consisting of $5,000$ observations over a set $\mathcal{V}$ of ten non-sequential features. As shown in Fig. 3a, the underlying causal graph $G(\mathcal{V}, \mathcal{E})$ is generated as a directed adjacency matrix $\boldsymbol{A} \in \mathbb{R}^{10 \times 10}$, using the Barabási-Albert

Table 2: Performance comparison of gene perturbation prediction on scRNA-seq datasets. Mean squared error (MSE) of perturbation predictions for top 20 differentially expressed genes are reported. † indicates the original positional encoding used in this method.

| Methods | Positional Encoding | Single-gene Perturbation | Double-gene Perturbations |
|---|---|---|---|
| scBERT [24] | Static absolute† | 0.224 | 0.230 |
| | Trainable relative | 0.219 $(-0.005)$ | 0.215 $(-0.015)$ |
| | CAPE | 0.193 $(-0.031)$ | 0.189 $(-0.041)$ |
| scGPT [25] | Trainable absolute† | 0.202 | 0.201 |
| | Trainable relative | 0.195 $(-0.007)$ | 0.204 $(+0.003)$ |
| | **CAPE** | **0.182** $(-0.020)$ | **0.176** $(-0.025)$ |

model [40], which is characterized by a *preferential attachment* mechanism. This mechanism assigns a higher probability of gaining new connections to nodes with a larger number of existing connections, thus varying the causal specificity across nodes [41]. We then utilize the *IIDSimulation* package from gCastle [42] to generate $X_{\text{syn}}$ based on $A$, employing MLP-based nonlinear assignments with added Gaussian noises.

CAPE is trained to estimate $A$ from $X_{\text{syn}}$. Fig. 3a and b show that the estimated adjacency matrix $\widehat{A}$ closely approximates the ground truth. Next, CAPE generates $d$-dimensional Poincaré ball positional encoding $e_v$ for each feature $v \in \mathcal{V}$ based on $\widehat{A}$, with $d = 3$ for visualization (Fig. 3c). These embeddings effectively encode causal strengths as pairwise distances and causal specificity as the distance to the origin, thereby accurately preserving the causal structure of $\widehat{A}$. For example, nodes with strong influence in the true DAG, such as node pairs (7, 0) and (3, 5), are embedded in close proximity on the Poincaré ball, demonstrating the model's ability to encode causal strength. Meanwhile, embeddings near the boundary (e.g., nodes 5 and 4) correspond to leaf nodes with high specificity, while those near the origin (e.g., nodes 6) correspond to root nodes with general causal influence, demonstrating the model's ability to model causal specificity.

**CAPE enhances the causality-awareness and robustness of the self-attention mechanism.** See Supp. G.1.

### 5.2 Empirically Evaluating Representation Learning with CAPE over Real Multi-Omics Data

To assess the effectiveness of CAPE in enhancing performance of transformer models over data with non-sequential yet causally-related data, we conduct evaluations using data from multiple omics domains [43], including transcriptomics, epigenomics, and proteomics, (see Supp. D for the data description). Feature and observation representations generated by the CAPE-transformer model are evaluated in various feature-level and observation-level downstream tasks. Here, we focus on the feature-level task, gene perturbation prediction (GPP) with scRNA-seq data [44], and leave the results of other tasks, e.g., cell clustering with proteomics data [45] and age prediction with epigenomics data [46], to Supp. G.2.

GPP aims to leverage the learned gene representations to predict perturbation (e.g., gene knockout or activation)-induced changes in gene expression profiles, facilitating the exploration of gene functions and regulatory networks. Here, we use a human leukemia cell dataset [23, 25, 47, 48], which includes unperturbed and perturbed cells under both single- and double-gene perturbations. Gene representations are learned using two prevalent transformer-based single-cell foundational models, including scBERT [24] and scGPT [25]. Different position encoding approaches, which do not rely on predefined feature order [8], are evaluated with the two models, including CAPE, their default methods, and a trainable, causality-agnostic relative position encoder [49]. The dataset is first preprocessed using standard scRNA-seq workflows [50], including quality control, normalization, and highly variable gene selection, as detailed in Supp. F.1. Following [24, 25], the two methods are trained on unperturbed cells to learn contextualized gene representations, which are fed into GEARS [44], a perturbation prediction model, to predict the perturbation-induced gene expression changes. Detailed implementations of the transformer models and positional encoding mechanisms can be found in Supp. F.2 and Supp. F.3, respectively. The prediction accuracy is measured as the mean squared error

---

[8]This explains why RoPE is not used as benchmark

Table 3: Ablation studies on gene perturbation prediction on scRNA-seq datasets. The standard deviation of each experiment is indicated in parentheses.

| Models | Single-gene Perturbation | Double-gene Perturbations |
|---|---|---|
| CAPE-null | 0.234 ($\pm$0.014) | 0.238 ($\pm$0.017) |
| CAPE-w/o-CSL | 0.209 ($\pm$0.010) | 0.213 ($\pm$0.011) |
| CAPE-w/o-hyperbolic | 0.192 ($\pm$0.008) | 0.196 ($\pm$0.008) |
| CAPE-w/o-rotary | 0.201 ($\pm$0.009) | 0.208 ($\pm$0.010) |
| **CAPE** | **0.182** ($\pm$0.005) | **0.176** ($\pm$0.008) |

(MSE) of the predicted and true expressions of the top 20 differentially expressed genes Tab. 2. We find that both models equipped with CAPE consistently yield substantial performance gains (11.1% average reduction in MSE) compared to their respective default approaches. This contrasts with using the causality-agnostic relative position encoder, which achieves only a 2.7% reduction.

## 5.3 Model Analyses

We provide a comprehensive analysis of CAPE, including ablation studies, sensitivity analysis, and complexity analysis. In this section, we primarily present the ablation studies, while the results of sensitivity and complexity analyses are deferred to Supp. G.3 and Supp. G.4, respectively.

We conduct a series of ablation studies to assess the contributions of CAPE's key components, using the same GPP dataset as described in Section 5.2. In these experiments, we adopt scGPT as the transformer backbone, replacing its default position encoding mechanism with four CAPE ablation variants. Specifically, the first variant completely omits CAPE (CAPE-null). The second variant (CAPE-w/o-CSL) excludes the Causal Structure Learning step (Step I) by replacing the learned causal graph with a similarity graph constructed from pairwise feature correlations. The third variant (CAPE-w/o-hyperbolic) removes the hyperbolic modeling component, replacing the hyperbolic distance ($d_l$ in Eq. (16) and Eq. (17)) with Euclidean distance and using standard SGD instead of RSGD for optimization. Lastly, the fourth variant (CAPE-w/o-rotary) bypasses the rotary form conversion (Step III), directly adding hyperbolic positional encodings to feature embeddings.

As shown in Tab. 3, removing any of these components leads to performance degradation, particularly in the double-gene perturbation prediction task. This task is inherently more challenging than single-gene perturbation prediction, as it requires more semantical informative gene representations. As expected, the CAPE-null variant yields the most significant increase in MSE for both prediction tasks, demonstrating the importance of CAPE's overall design. The second-largest performance drop arises with CAPE-w/o-CSL, where replacing the learned causal DAG with an undirected similar matrix impairs the model's ability to encode causal relationships in the positional encodings. Similarly, APE-w/o-rotary leads to a notable decrease in performance, as omitting the rotary form forfeits the attention mechanism's sensitivity to causal distance and causal generality, two key causal properties for capturing hierarchical causal semantics. Moreover, this omission also undermines CAPE's sensitivity to nuanced changes in underlying causal semantics, consistent with prior findings that highlight the importance of concentrated attention scores in context modeling [19]. Finally, although CAPE-w/o-hyperbolic leads to a more modest decline in performance, we emphasize the essential role of space curvature-aware optimization, which allows the positional encodings to be placed in optimal locations that better reflect the underlying causal graph structure.

## 6 Conclusion

In this study, we present CAPE, a causality-aware positional encoding that enables transformers to handle non-sequential yet causally-related features by modeling their latent structure as a DAG and embedding it in hyperbolic space. By unifying causal discovery, hyperbolic geometry, and rotary attention, CAPE effectively captures core properties of causal graphs, causal strength and causal specificity, and translates them into position-aware attention dynamics. Our theoretical analysis reveals desirable behaviors of the resulting self-attention, and extensive empirical results across synthetic and multi-omics datasets validate the benefits. CAPE opens a pioneering path toward causal representation learning in domains where traditional positional encodings fail.

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

# Supplementary Material

In this supplementary material, we first provide theoretical analysis in Supp. A and justify the use of gregariousness ($\boldsymbol{\pi}_v$) in regularization in Supp. B. Supp. C reviews related work, while Supp. D and Supp. E detail the datasets and evaluation protocols. Supp. F outlines implementation details, including preprocessing, architecture, and training. Finally, Supp. G presents additional results, covering CAPE's empirical properties, multi-omics benchmarks, sensitivity, and complexity analyses. The code for our model and experiments is available at `https://github.com/Catchxu/CAPE`.

## A  Theoretical Analysis

### A.1  Equivalence between Constraint Condition and Acyclicity

**Proposition A.1** ([26]). *A matrix $\boldsymbol{A} \in \mathbb{R}^{M \times M}$ can induce a DAG, if and only if :*

$$h(\boldsymbol{A}) := \mathrm{tr}(e^{\boldsymbol{A} \odot \boldsymbol{A}}) - M = 0, \tag{A.1}$$

*where $\odot$ is the element-wise product. Moreover, the gradient of $h(\boldsymbol{A})$ follows a simple form of $\nabla h(\boldsymbol{A}) = (e^{\boldsymbol{A} \odot \boldsymbol{A}})^\top \odot 2\boldsymbol{A}$.*

*Proof.* Given a non-negative matrix $\boldsymbol{B} \in \mathbb{R}_{\geq 0}^{M \times M}$ as the adjacency matrix of a DAG $G$, $\boldsymbol{B}^k$ reflects its $k$-hop connectivity. Specifically, $\boldsymbol{B}_{uv}^k \neq 0$ indicates that a path of length $k$ exists from $u$ to $v$. Consequently, $\mathrm{tr}(\boldsymbol{B}^k)$ equals to the number of length-$k$ self-cycles in $G$. That is, $G$ is acyclic if and only if:

$$\mathrm{tr}(\boldsymbol{B}^k) = 0, \quad \forall k \in \mathbb{Z}_+. \tag{A.2}$$

This infinite family of constraints can be reformulated using matrix exponential:

$$
\begin{aligned}
\mathrm{tr}(e^{\boldsymbol{B}}) &= \mathrm{tr}(\boldsymbol{I}) + \sum_{k=1}^{+\infty} \frac{\mathrm{tr}(\boldsymbol{B}^k)}{k!} \\
&= M + \sum_{k=1}^{+\infty} \sum_{u=1}^{M} \frac{(\boldsymbol{B}^k)_{uu}}{k!} \\
&= M \iff \mathrm{tr}(e^{\boldsymbol{B}}) - M = 0.
\end{aligned}
\tag{A.3}
$$

Replacing $\boldsymbol{B}$ with the element-wise product matrix $\boldsymbol{A} \odot \boldsymbol{A}$, which ensures the non-negativity, we reach the constraint $\mathrm{tr}(e^{\boldsymbol{A} \odot \boldsymbol{A}}) - M = 0$. In particular, this constraint exhibits a simple form of gradient for easy optimization:

$$\nabla h(\boldsymbol{A}) = \frac{\partial \mathrm{tr}(e^{\boldsymbol{S}})}{\partial \boldsymbol{A}} = \frac{\partial \mathrm{tr}(e^{\boldsymbol{S}})}{\partial \boldsymbol{S}} \odot \frac{\partial \boldsymbol{S}}{\partial \boldsymbol{A}} = (e^{\boldsymbol{A} \odot \boldsymbol{A}})^\top \odot 2\boldsymbol{A}, \tag{A.4}$$

where $\boldsymbol{S} = \boldsymbol{A} \odot \boldsymbol{A}$. This completes the proof. $\square$

### A.2  Optimization and Diffeomorphism of Hyperbolic Models

CAPE involves two hyperbolic models, including the hyperboloid model and Poincaréball model, to embed the identified DAG in a hyperbolic space. On one hand, the hyperboloid model is employed for an efficient RSGD optimization of the node embeddings, due to its simple closed-form computation of geodesics on the hyperboloid. On the other hand, we utilize a diffeomorphism to map hyperboloid embeddings to Poincaré ball embeddings, which are more natural for both conversion into their rotary form and visualization. In the following subsections, we will discuss the geometric properties, the optimization, and the diffeomorphism of the two models.

#### A.2.1  Hyperboloid Model

Recall the definition of the hyperboloid model:

**Definition A.1** ([51]). *d-dimensional hyperboloid model is a Riemannian manifold $\mathcal{L}^d := (\mathcal{H}^d, \boldsymbol{g}_l)$, where $\boldsymbol{g}_l := \mathrm{diag}(-1, 1, \cdots, 1) \in \mathbb{R}^{(d+1) \times (d+1)}$ denotes the metric tensor and where*

$$\mathcal{H}^d := \{\boldsymbol{p} := (p^{(0)}, \widetilde{\boldsymbol{p}}) \in \mathbb{R}^{d+1} : \langle \boldsymbol{p}, \boldsymbol{p} \rangle_l = -1, p^{(0)} > 0\}, \tag{A.5}$$

$$\langle \boldsymbol{p}_{v_m}, \boldsymbol{p}_{v_n} \rangle_l := \boldsymbol{p}_{v_m}^\top \boldsymbol{g}_l \, \boldsymbol{p}_{v_n} = -p_{v_m}^{(0)} p_{v_n}^{(0)} + \widetilde{\boldsymbol{p}}_{v_m}^\top \widetilde{\boldsymbol{p}}_{v_n}, \tag{A.6}$$

*denotes the upper sheet of a two-sheeted hyperboloid with an origin $\boldsymbol{p_o} = (1, \boldsymbol{0}_d)^\top$ in a $(d+1)$-dimensional Minkowski space [37].*

The distance of $\boldsymbol{p}_{v_m}, \boldsymbol{p}_{v_n} \in \mathcal{H}^d$ on $\mathcal{L}^d$ is given as [38]:

$$d_l(\boldsymbol{p}_{v_m}, \boldsymbol{p}_{v_n}) = \mathrm{arcosh}(-\langle \boldsymbol{p}_{v_m}, \boldsymbol{p}_{v_n} \rangle_l). \tag{A.7}$$

Geometrically, the tangent space $\mathcal{T}_{\boldsymbol{p}} \mathcal{H}^d$ at a point $\boldsymbol{p} \in \mathcal{H}^d$ is defined as the set of vectors orthogonal to $\boldsymbol{p}$ [38, 51]:

$$\mathcal{T}_{\boldsymbol{p}} \mathcal{H}^d = \{\boldsymbol{v} \in \mathbb{R}^{d+1} : \langle \boldsymbol{v}, \boldsymbol{p} \rangle_l = 0\}. \tag{A.8}$$

The geodesics of $\mathcal{H}^d$ can then be computed based on the following proposition.

**Proposition A.2.** *Given $\boldsymbol{p} \in \mathcal{H}^d$ and an unit tangent vector $\boldsymbol{v} \in \mathcal{T}_{\boldsymbol{p}} \mathcal{H}^d$ where $\langle \boldsymbol{v}, \boldsymbol{v} \rangle_l = 1$, the unique unit-speed geodesic $\phi_{\boldsymbol{p}, \boldsymbol{v}} : [0, 1] \to \mathcal{H}^d$ can be parameterized as:*

$$\phi_{\boldsymbol{p}, \boldsymbol{v}}(t) = \cosh(t)\boldsymbol{p} + \sinh(t)\boldsymbol{v} \quad \text{s.t.} \quad \phi_{\boldsymbol{p}, \boldsymbol{v}}(0) = \boldsymbol{p}, \dot{\phi}_{\boldsymbol{p}, \boldsymbol{v}}(0) = \boldsymbol{v}, \tag{A.9}$$

*where $t \in [0, 1]$. $\dot{\phi}_{\boldsymbol{p}, \boldsymbol{v}}(0)$ denotes the derivative of $\phi_{\boldsymbol{p}, \boldsymbol{v}}(t)$ to $t$, or the velocity of the geodesic $\phi_{\boldsymbol{p}, \boldsymbol{v}}(t)$ at time $t$.*

*Proof.* According to the definition of geodesic [38, 51], $\phi : [0, 1] \to \mathcal{H}^d$ is a geodesic if and only if it satisfies the equivalent conditions:

$$\nabla \dot{\phi} \equiv 0 \quad \Longleftrightarrow \quad \ddot{\phi} = \langle \dot{\phi}, \dot{\phi} \rangle_l \, \phi. \tag{A.10}$$

This means that the acceleration vector $\nabla \dot{\phi}$ is zero. In other words, along the geodesic $\phi$, the direction of the velocity vector does not "turn". When $\langle \dot{\phi}, \dot{\phi} \rangle_l = 1$, Eq. (A.10) can be formulated as an ordinary differential equation (ODE):

$$\ddot{\phi}(t) = \phi(t), \tag{A.11}$$

with a general solution:

$$\phi(t) = \boldsymbol{\alpha} e^t + \boldsymbol{\beta} e^{-t}, \tag{A.12}$$

where $\boldsymbol{\alpha}, \boldsymbol{\beta} \in \mathbb{R}^{d+1}$ are constant vectors. $\boldsymbol{\alpha}, \boldsymbol{\beta}$ can be determined by the initial conditions as:

$$\begin{aligned} \phi_{\boldsymbol{p}, \boldsymbol{v}}(0) = \boldsymbol{p} &\implies \boldsymbol{\alpha} + \boldsymbol{\beta} = \boldsymbol{p}, \\ \dot{\phi}_{\boldsymbol{p}, \boldsymbol{v}}(0) = \boldsymbol{v} &\implies \boldsymbol{\alpha} e^0 - \boldsymbol{\beta} e^{-0} = \boldsymbol{\alpha} - \boldsymbol{\beta} = \boldsymbol{v}. \end{aligned} \tag{A.13}$$

Then, we have:

$$\boldsymbol{\alpha} = \frac{1}{2}(\boldsymbol{p} + \boldsymbol{v}), \quad \boldsymbol{\beta} = \frac{1}{2}(\boldsymbol{p} - \boldsymbol{v}), \tag{A.14}$$

$$\phi_{\boldsymbol{p}, \boldsymbol{v}}(t) = \cosh(t)\boldsymbol{p} + \sinh(t)\boldsymbol{v}. \tag{A.15}$$

This completes the proof. $\qquad\square$

Eq. (A.10) can be extended to tangent vectors of any length using the exponential map, defined as:

**Proposition A.3 (Exponential map of hyperboloid model).** *Given $\boldsymbol{p} \in \mathcal{H}^d$ and $\boldsymbol{v} \in \mathcal{T}_{\boldsymbol{p}} \mathcal{H}^d$ with a length $\|\boldsymbol{v}\|_l = \sqrt{\langle \boldsymbol{v}, \boldsymbol{v} \rangle_l}$, there exists a unique geodesic $\tilde{\phi} : [0, 1] \to \mathcal{H}^d$ with $\tilde{\phi}(0) = p, \tilde{\phi}'(0) = v$. The exponential map $\exp_{\boldsymbol{p}} : \mathcal{T}_{\boldsymbol{p}} \mathcal{H}^d \to \mathcal{H}^d$ is defined as $\exp_{\boldsymbol{p}}(\boldsymbol{v}) := \tilde{\phi}(1)$, which is the end point of the geodesic on the manifold. Based on Prop. A.2, $\exp_{\boldsymbol{p}}$ can be represented as:*

$$\exp_{\boldsymbol{p}}(\boldsymbol{v}) = \cosh(\|\boldsymbol{v}\|_l)\boldsymbol{p} + \sinh(\|\boldsymbol{v}\|_l)\frac{\boldsymbol{v}}{\|\boldsymbol{v}\|_l}. \tag{A.16}$$

In other words, the exponential map yields the unique point on $\mathcal{H}^d$ reached by traveling along the geodesic originating at $\boldsymbol{p}$ along the direction of $\boldsymbol{v}$ for a hyperbolic distance of $\|\boldsymbol{v}\|_l$.

Moreover, given any Riemannian gradient direction $\boldsymbol{u} \in \mathbb{R}^{d+1}$ at $\boldsymbol{p}$, it can be projected to the tangent space using the following proposition.

**Proposition A.4 (Projecting Riemannian gradient to tangent space).** *Given $\boldsymbol{p} \in \mathcal{H}^d$, a Riemannian gradient $\boldsymbol{u} \in \mathbb{R}^{d+1}$ at $\boldsymbol{p}$ can be projected to the tangent space $\mathcal{T}_{\boldsymbol{p}}\mathcal{H}^d$ via:*

$$\text{proj}_{\boldsymbol{p}}(\boldsymbol{u}) = \boldsymbol{u} + \langle \boldsymbol{p}, \boldsymbol{u} \rangle_l \, \boldsymbol{p}. \tag{A.17}$$

*Proof.* Since $\boldsymbol{p}$ is orthogonal to the tangent space, we can derived the projected $\boldsymbol{u}$ using the generalized Gram-Schmidt Orthogonalization:

$$\text{proj}_{\boldsymbol{p}}(\boldsymbol{u}) = \boldsymbol{u} - \frac{\langle \boldsymbol{u}, \boldsymbol{p} \rangle_l}{\langle \boldsymbol{p}, \boldsymbol{p} \rangle_l} \boldsymbol{p}. \tag{A.18}$$

Given $\langle \boldsymbol{p}, \boldsymbol{p} \rangle_l = -1$, we have:

$$\text{proj}_{\boldsymbol{p}}(\boldsymbol{u}) = \boldsymbol{u} + \langle \boldsymbol{p}, \boldsymbol{u} \rangle_l \boldsymbol{p}. \tag{A.19}$$

This completes the proof. $\square$

Prop. A.4 and Prop. A.3 together allow the RSGD steps in Eq. (19), Eq. (20), and Eq. (21) in Section 3.4.

### A.2.2   Poincaré Ball Model

**Definition A.2 ([52]).** *A $d$-dimensional Poincaré ball manifold is a Riemannian manifold defined as $\mathcal{P}^d := (\mathcal{B}^d, \boldsymbol{g}_p)$, where $\mathcal{B}^d := \{\boldsymbol{e} \in \mathbb{R}^d : \|\boldsymbol{e}\| < 1\}$ represents an open $d$-dimensional unit ball with the metric tensor:*

$$\boldsymbol{g}_p = \left( \frac{2}{1 - \|\boldsymbol{e}\|^2} \right)^2 \boldsymbol{I}, \tag{A.20}$$

*which is a conformal transformation of the Euclidean metric $\boldsymbol{I}$.*

The distance between two points $\boldsymbol{e}_{v_m}, \boldsymbol{e}_{v_n} \in \mathcal{P}^d$ reads [38]:

$$d_p(\boldsymbol{e}_{v_m}, \boldsymbol{e}_{v_n}) = \text{arcosh}\left( 1 + 2 \frac{\|\boldsymbol{e}_{v_m} - \boldsymbol{e}_{v_n}\|^2}{(1 - \|\boldsymbol{e}_{v_m}\|^2)(1 - \|\boldsymbol{e}_{v_n}\|^2)} \right). \tag{A.21}$$

It is straightforward to see that as $\|\boldsymbol{e}_{\boldsymbol{v_m}}\| \to 1$ or $\|\boldsymbol{e}_{\boldsymbol{v_n}}\| \to 1$, $d_p(\boldsymbol{e}_{v_m}, \boldsymbol{e}_{v_n}) \to +\infty$. Hence, the regions closer to the boundary of the Poincaré ball manifold have larger space capacity, thus allowing it to model data with hierarchical structures and power-law distributions, e.g., a causal graph.

On the other hand, as a point $\boldsymbol{e}$ approaches the origin, $\|\boldsymbol{e}\| \to 0$. And Eq. (A.21) suggests that, on average, it is closer to other points, thus can represent a casually more general node that connects to many its causal descendants. Meanwhile, $\|\boldsymbol{e}\|$ can be used to represent the causal specificity of the corresponding node.

### A.2.3   Transforming Hyperboloid manifold to Poincaré ball manifold via diffeomorphism

We first define diffeomorphism $f_d$ as follows.

**Definition A.3 (Diffeomorphism).** *Given two differentiable manifolds $\mathcal{M}_1$ and $\mathcal{M}_2$, a continuously differentiable map $f_d : \mathcal{M}_1 \to \mathcal{M}_2$ is a diffeomorphism if it is a bijection and its inverse $f_d^{-1} : \mathcal{M}_2 \to \mathcal{M}_1$ is differentiable as well. And we claim that $\mathcal{M}_1$ and $\mathcal{M}_2$ are diffeomorphic.*

The following proposition allows the bidirectional transformation between the hyperboloid manifold and the Poincaré ball manifold via diffeomorphism.

**Proposition A.5.** *The manifold $\mathcal{L}^d$ in Def. A.1 and manifold $\mathcal{P}^d$ in Def. A.2 are diffeomorphic, where the diffeomorphism $f_d : \mathcal{H}^d \to \mathcal{B}^d$ and its inverse $f_d^{-1} : \mathcal{B}^d \to \mathcal{H}^d$ are given by:*

$$f_d(\boldsymbol{p}) = \frac{(p^{(1)}, p^{(2)}, \cdots, p^{(d)})^\top}{p^{(0)} + 1}, \tag{A.22}$$

$$f_d^{-1}(\boldsymbol{e}) = \frac{(1 + \|\boldsymbol{e}\|^2, 2e^{(1)}, \cdots, 2e^{(d)})^\top}{1 - \|\boldsymbol{e}\|^2}. \tag{A.23}$$

*where $\boldsymbol{p} \in \mathcal{H}^d$ and $\boldsymbol{e} \in \mathcal{B}^d$.*

*Proof.* The proof consists of three parts:

1. $f_d$ is a mapping from $\mathcal{L}^d$ to $\mathcal{P}^d$;

2. $f_d^{-1}$ is a mapping from $\mathcal{P}^d$ to $\mathcal{L}^d$;

3. $f_d$ and $f_d^{-1}$ are differentiable and mutually inverse,

which are proved sequentially in the following proofs.

**Proof 1. $f_d$ is a mapping from $\mathcal{L}^d$ to $\mathcal{P}^d$.** For $\boldsymbol{p} \in \mathcal{H}^d$, it satisfies:

$$\langle \boldsymbol{p}, \boldsymbol{p} \rangle_l = -p^{(0)^2} + \sum_{t=1}^d p^{(t)^2} = -1. \tag{A.24}$$

Then:

$$\|f_d(\boldsymbol{p})\|^2 = \frac{\sum_{t=1}^d p^{(t)^2}}{\left(p^{(0)} + 1\right)^2} = \frac{p^{(0)^2} - 1}{\left(p^{(0)} + 1\right)^2} = 1 - \frac{2p^{(0)} + 2}{\left(p^{(0)} + 1\right)^2}. \tag{A.25}$$

Since $p^{(0)} > 0$, $\|f_d(\boldsymbol{p})\| < 1 \Rightarrow f_d(\boldsymbol{p}) \in \mathcal{B}^d$, which indicates that $f_d$ is a valid mapping function from $\mathcal{L}^d$ to $\mathcal{P}^d$.

**Proof 2. $f_d^{-1}$ is a mapping from $\mathcal{P}^d$ to $\mathcal{L}^d$.** Suppose $r = \|\boldsymbol{e}\|$, then:

$$\begin{aligned}
\langle f_d^{-1}(\boldsymbol{e}), f_d^{-1}(\boldsymbol{e}) \rangle_l &= -\left(\frac{1 + r^2}{1 - r^2}\right)^2 + \sum_{t=1}^d \left(\frac{2e^{(t)}}{1 - r^2}\right)^2 = -\left(\frac{1 + r^2}{1 - r^2}\right)^2 + \frac{4r^2}{(1 - r^2)^2} \\
&= \frac{-(1 + r^2)^2 + 4r^2}{(1 - r^2)^2} = \frac{-(1 - r^2)^2}{(1 - r^2)^2} = -1.
\end{aligned} \tag{A.26}$$

Therefore, $f_d^{-1}(\boldsymbol{e}) \in \mathcal{H}^d$, which completes this proof.

**Proof 3. $f_d$ and $f_d^{-1}$ are differentiable and mutually inverse.** Since $f_d$ and $f_d^{-1}$ are both rational functions and their denominators remain nonzero in their domains (e.g., $p^{(0)} + 1 > 0$, $1 - \|\boldsymbol{e}\|^2 > 0$), they are infinitely differentiable. Moreover, we have:

$$f_d^{-1}(f_d(\boldsymbol{p}))^{(0)} = \frac{1 + \|f_d(\boldsymbol{p})\|^2}{1 - \|f_d(\boldsymbol{p})\|^2} = \frac{2p^{(0)^2} + 2p^{(0)}}{2p^{(0)} + 2} = p^{(0)}, \tag{A.27}$$

$$f_d^{-1}(f_d(\boldsymbol{p}))^{(t)} = \frac{2p^{(t)}/(p^{(0)} + 1)}{1 - \|f_d(\boldsymbol{p})\|^2} = 2p^{(t)} \frac{1}{p^{(0)} + 1} \frac{\left(p^{(0)} + 1\right)^2}{2p^{(0)} + 2} = p^{(t)}, \ \forall t = 1, 2 \cdots, d. \tag{A.28}$$

Thus, $f_d^{-1}(f_d(\boldsymbol{p})) = \boldsymbol{p}$, and $f_d^{(-1)}$ is indeed the inverse function of $f_d$. Similarly, it is trivial to prove $f_d(f_d^{-1}(\boldsymbol{e})) = \boldsymbol{e}$, indicating that $f_d$ and $f_d^{-1}$ are mutually inverse. This completes the proof.

$\square$

## A.3 Solving the Query and Key Mapping Functions

We solve Eq. (22) and Eq. (23) in Section 3.5 in complex space. To this end, we begin with some preliminaries of the complex space.

**Proposition A.6.** *When $D$ is even, real space $\mathbb{R}^D$ and complex space $\mathbb{C}^{D/2}$ are diffeomorphic (Def. A.3), where the diffeomorphism $T : \mathbb{R}^D \to \mathbb{C}^{D/2}$ and its inverse $T^{-1} : \mathbb{C}^{D/2} \to \mathbb{R}^D$ are given by:*

$$
T(\boldsymbol{x}) = T \begin{bmatrix} x^{(1)} \\ x^{(2)} \\ \vdots \\ x^{(D)} \end{bmatrix} := \begin{bmatrix} x^{(1)} + \mathrm{i}x^{(2)} \\ x^{(3)} + \mathrm{i}x^{(4)} \\ \vdots \\ x^{(D-1)} + \mathrm{i}x^{(D)} \end{bmatrix}, \quad T^{-1}(\boldsymbol{z}) = T^{-1} \begin{bmatrix} z^{(1)} \\ z^{(2)} \\ \vdots \\ z^{(D/2)} \end{bmatrix} := \begin{bmatrix} \mathrm{Re}(z^{(1)}) \\ \mathrm{Im}(z^{(1)}) \\ \vdots \\ \mathrm{Re}(z^{(D/2)}) \\ \mathrm{Im}(z^{(D/2)}) \end{bmatrix}.
$$
(A.29)

*where $\boldsymbol{x} \in \mathbb{R}^D$ and $\boldsymbol{z} \in \mathbb{C}^{D/2}$.*

*Proof.* Given $T(\boldsymbol{x}) \in \mathbb{C}^{D/2}$ and $T^{-1}(\boldsymbol{z}) \in \mathbb{R}^D$, it is obvious to have $T^{-1}(T(\boldsymbol{x})) = \boldsymbol{x}$ and $T(T^{-1}(\boldsymbol{z})) = \boldsymbol{z}$. Moreover, each component of $T$ is a linear mapping as $(x^{(t)}, x^{(t+1)}) \mapsto x^{(t)} + \mathrm{i}x^{(t+1)}$, $t = 1, 3, \cdots, D-1$, and the same holds for $T^{-1}$. Therefore, $T$ is a bijection, and $T$ and $T^{-1}$ are differentiable and mutually inverse. This completes the proof. $\qquad\square$

Furthermore, the canonical inner product for two vectors $\boldsymbol{z}_1, \boldsymbol{z}_2 \in \mathbb{C}^{D/2}$ is defined as:

$$
\langle \boldsymbol{z}_1, \boldsymbol{z}_2 \rangle := \boldsymbol{z}_1^\top \boldsymbol{z}_2^* = \sum_{t=1}^{D/2} z_1^{(t)} z_2^{(t)*},
$$
(A.30)

where $\boldsymbol{z}_2^*$ denotes the complex conjugate of $\boldsymbol{z}_2$.

Next, as defined in Section 3.5, $v_m$ denotes a non-sequential feature, with $\boldsymbol{v}_m^i \in \mathbb{R}^D$ representing its position-agnostic feature embedding in the context of the $i$-th observation, and $\boldsymbol{e}_{v_m} \in \mathcal{B}^d$ representing its Poincaré ball positional encoding. We define the dimensionality of positional encodings as $d = D/2$, and give Eq. (23) in $\mathbb{C}^d$ as:

$$
\langle T\left[\mathcal{I}_q(\boldsymbol{v}_m^i, \boldsymbol{e}_{v_m})\right], T\left[\mathcal{I}_k(\boldsymbol{v}_n^i, \boldsymbol{e}_{v_n})\right] \rangle = \mathcal{A}\left(\boldsymbol{v}_m^i, \boldsymbol{v}_n^i, \boldsymbol{\gamma}(\boldsymbol{e}_{v_m}, \boldsymbol{e}_{v_n})\right),
$$
(A.31)

where $T : \mathbb{R}^D \to \mathbb{C}^d$ is the mapping function in Prop. A.6, and $\mathcal{A}$ is a scoring function. Both left- and right-hand sides of the above equation can be represented as exponential forms:

$$
T\left[\mathcal{I}_q(\boldsymbol{v}_m^i, \boldsymbol{e}_{v_m})\right] = \rho_q(\boldsymbol{v}_m^i, \boldsymbol{e}_{v_m}) \exp\left\{\mathrm{i}\boldsymbol{\theta}_q(\boldsymbol{v}_m^i, \boldsymbol{e}_{v_m})\right\},
$$
(A.32)

$$
T\left[\mathcal{I}_k(\boldsymbol{v}_n^i, \boldsymbol{e}_{v_n})\right] = \rho_k(\boldsymbol{v}_n^i, \boldsymbol{e}_{v_n}) \exp\left\{\mathrm{i}\boldsymbol{\theta}_k(\boldsymbol{v}_n^i, \boldsymbol{e}_{v_n})\right\},
$$
(A.33)

$$
\mathcal{A}\left(\boldsymbol{v}_m^i, \boldsymbol{v}_n^i, \boldsymbol{\gamma}(\boldsymbol{e}_{v_m}, \boldsymbol{e}_{v_n})\right) = \rho_{\mathcal{A}}\left(\boldsymbol{v}_m^i, \boldsymbol{v}_n^i, \boldsymbol{\gamma}(\boldsymbol{e}_{v_m}, \boldsymbol{e}_{v_n})\right) \mathbf{1}^\top \exp\left\{\mathrm{i}\boldsymbol{\theta}_{\mathcal{A}}\left(\boldsymbol{v}_m^i, \boldsymbol{v}_n^i, \boldsymbol{\gamma}(\boldsymbol{e}_{v_m}, \boldsymbol{e}_{v_n})\right)\right\},
$$
(A.34)

where $\rho_q, \rho_k : \mathbb{R}^D \times \mathbb{R}^d \to \mathbb{R}$, $\rho_{\mathcal{A}} : \mathbb{R}^D \times \mathbb{R}^D \times \mathbb{R}^d \to \mathbb{R}$ denote the radius functions to be solved; and $\boldsymbol{\theta}_q, \boldsymbol{\theta}_k : \mathbb{R}^D \times \mathbb{R}^d \to \mathbb{R}^d$, $\boldsymbol{\theta}_{\mathcal{A}} : \mathbb{R}^D \times \mathbb{R}^D \times \mathbb{R}^d \to \mathbb{R}^d$ denote the angle functions to be solved. Meanwhile, we have:

$$
\begin{aligned}
\langle e^{\mathrm{i}\boldsymbol{\theta}_q(\boldsymbol{v}_m^i, \boldsymbol{e}_{v_m})}, e^{\mathrm{i}\boldsymbol{\theta}_k(\boldsymbol{v}_n^i, \boldsymbol{e}_{v_n})} \rangle &= \sum_{t=1}^{d} \exp\left\{\mathrm{i}\boldsymbol{\theta}_q^{(t)}(\boldsymbol{v}_m^i, \boldsymbol{e}_{v_m})\right\} \exp\left\{\mathrm{i}\boldsymbol{\theta}_k^{(t)}(\boldsymbol{v}_n^i, \boldsymbol{e}_{v_n})\right\}^* \\
&= \sum_{t=1}^{d} \exp\left\{\mathrm{i}\boldsymbol{\theta}_q^{(t)}(\boldsymbol{v}_m^i, \boldsymbol{e}_{v_m}) - \mathrm{i}\boldsymbol{\theta}_k^{(t)}(\boldsymbol{v}_n^i, \boldsymbol{e}_{v_n})\right\} \\
&= \mathbf{1}^\top \exp\left\{\mathrm{i}\left[\boldsymbol{\theta}_q(\boldsymbol{v}_m^i, \boldsymbol{e}_{v_m}) - \boldsymbol{\theta}_k(\boldsymbol{v}_n^i, \boldsymbol{e}_{v_n})\right]\right\}.
\end{aligned}
$$
(A.35)

Substituting Eqs. (A.32) to (A.35) into Eq. (A.31) yields:

$$
\begin{aligned}
\rho_q(\boldsymbol{v}_m^i, \boldsymbol{e}_{v_m})\rho_k(\boldsymbol{v}_n^i, \boldsymbol{e}_{v_n}) \mathbf{1}^\top \exp\left\{\mathrm{i}\left[\boldsymbol{\theta}_q(\boldsymbol{v}_m^i, \boldsymbol{e}_{v_m}) - \boldsymbol{\theta}_k(\boldsymbol{v}_n^i, \boldsymbol{e}_{v_n})\right]\right\} \\
= \rho_{\mathcal{A}}\left(\boldsymbol{v}_m^i, \boldsymbol{v}_n^i, \boldsymbol{\gamma}(\boldsymbol{e}_{v_m}, \boldsymbol{e}_{v_n})\right) \mathbf{1}^\top \exp\left\{\mathrm{i}\boldsymbol{\theta}_{\mathcal{A}}\left(\boldsymbol{v}_m^i, \boldsymbol{v}_n^i, \boldsymbol{\gamma}(\boldsymbol{e}_{v_m}, \boldsymbol{e}_{v_n})\right)\right\}.
\end{aligned}
$$
(A.36)

A straightforward solution to the above equation reads:

$$
\begin{cases} \rho_q(\boldsymbol{v}_m^i, \boldsymbol{e}_{v_m})\rho_k(\boldsymbol{v}_n^i, \boldsymbol{e}_{v_n}) = \rho_{\mathcal{A}}\left(\boldsymbol{v}_m^i, \boldsymbol{v}_n^i, \boldsymbol{\gamma}(\boldsymbol{e}_{v_m}, \boldsymbol{e}_{v_n})\right), \\ \boldsymbol{\theta}_q(\boldsymbol{v}_m^i, \boldsymbol{e}_{v_m}) - \boldsymbol{\theta}_k(\boldsymbol{v}_n^i, \boldsymbol{e}_{v_n}) = \boldsymbol{\theta}_{\mathcal{A}}\left(\boldsymbol{v}_m^i, \boldsymbol{v}_n^i, \boldsymbol{\gamma}(\boldsymbol{e}_{v_m}, \boldsymbol{e}_{v_n})\right), \end{cases}
$$
(A.37)

where $\gamma(e_{v_m}, e_{v_n})$ is a function that reflects the relative information between $e_{v_m}$ and $e_{v_n}$. We further assume the relative self-information is a constant:

$$\gamma(e_{v_k}, e_{v_k}) = c, \quad \forall k = 1, 2, \cdots, M, \tag{A.38}$$

where $c \in \mathbb{R}^d$ is a constant vector. The explicit solutions of the radius and angle functions in Eq. (A.37) are given as follows.

**(1) Radius Functions.** For any $e_{v_m} = e_{v_n}$, we have:

$$\rho_q(v_m^i, e_{v_m})\rho_k(v_n^i, e_{v_m}) = \rho_q(v_m^i, \mathbf{0})\rho_k(v_n^i, \mathbf{0}) = \rho_{\mathcal{A}}\left(v_m^i, v_n^i, c\right), \tag{A.39}$$

which indicates that the radius functions are independent of $e_v$. This is a causal position analogue of the radius function used in RoPE [18], with the solutions:

$$\rho_q(v_m^i, e_{v_m}) = \|v_m^i\|, \quad \rho_k(v_n^i, e_{v_n}) = \|v_n^i\|, \tag{A.40}$$

$$\rho_{\mathcal{A}}\left(v_m^i, v_n^i, \gamma(e_{v_m}, e_{v_n})\right) = \|v_m^i\| \|v_n^i\|. \tag{A.41}$$

**(2) Angle Functions.** For any $e_{v_m} = e_{v_n}$, it always exists:

$$\boldsymbol{\theta}_q(v_m^i, e_{v_m}) - \boldsymbol{\theta}_k(v_n^i, e_{v_m}) = \boldsymbol{\theta}_q(v_m^i, \mathbf{0}) - \boldsymbol{\theta}_k(v_n^i, \mathbf{0}) = \boldsymbol{\theta}_{\mathcal{A}}\left(v_m^i, v_n^i, c\right). \tag{A.42}$$

It follows:

$$\boldsymbol{\theta}_q(v_m^i, e_{v_m}) - \boldsymbol{\theta}_q(v_m^i, \mathbf{0}) = \boldsymbol{\theta}_k(v_n^i, e_{v_m}) - \boldsymbol{\theta}_k(v_n^i, \mathbf{0}). \tag{A.43}$$

We simplify the solution by using the same function $\boldsymbol{\zeta}$ for both $\boldsymbol{\theta}_q$ and $\boldsymbol{\theta}_k$, yielding:

$$\boldsymbol{\zeta}(v_m^i, e_{v_m}) - \boldsymbol{\zeta}(v_m^i, \mathbf{0}) = \boldsymbol{\zeta}(v_n^i, e_{v_m}) - \boldsymbol{\zeta}(v_n^i, \mathbf{0}). \tag{A.44}$$

which indicates $\boldsymbol{\zeta}(v_m^i, e_{v_m}) - \boldsymbol{\zeta}(v_m^i, \mathbf{0})$ is independent of $v_{\{m,n\}}^i$ and can be expressed as:

$$\boldsymbol{\zeta}(v_m^i, e_{v_m}) - \boldsymbol{\zeta}(v_m^i, \mathbf{0}) = \phi(e_{v_m}), \tag{A.45}$$

where $\phi : \mathbb{R}^d \to \mathbb{R}^d$ is a function that captures the effects of $e_{v_m}$, which can also be viewed a causal position analogue of the angle function used in RoPE.

The following analysis will focus on Eq. (A.32) since it also applies to Eq. (A.33). Substituting Eqs. (A.40) and (A.45) into Eq. (A.32), we have:

$$T\left[\mathcal{I}_q(v_m^i, e_{v_m})\right] = \|v_m^i\| \exp\left\{i\boldsymbol{\theta}_q(v_m^i, \mathbf{0})\right\} \odot \exp\left\{i\phi(e_{v_m})\right\}, \tag{A.46}$$

where $\odot$ denotes the element-wise product. Since the term before $\odot$ on the RHS of the above equation depends only on $v_m^i$, we denote it as:

$$\|v_m^i\| \exp\left\{i\boldsymbol{\theta}_q(v_m^i, \mathbf{0})\right\} = T\left(\boldsymbol{W}_q v_m^i\right) = T\left(q_{v_m}^i\right) \in \mathbb{C}^d, \tag{A.47}$$

where $\boldsymbol{W}_q \in \mathbb{R}^{D \times D}$ are trainable weights for query or key mapping. Note that any complex exponential can be transformed into a rotation matrix in real space as shown in the lemma below.

**Lemma A.1.** *Given $z = z_1 + iz_2 \in \mathbb{C}$, $\theta \in \mathbb{R}$, and the mapping function $T : \mathbb{R}^2 \to \mathbb{C}$ defined as $T\left[(z_1, z_2)^\top\right] = z$, the following equivalence exists:*

$$\begin{aligned} T^{-1}\left(z \exp\{i\theta\}\right) &= T^{-1}\left[(z_1 + iz_2)\left(\cos(\theta) + i\sin(\theta)\right)\right] \\ &= T^{-1}\left[\cos(\theta)z_1 - \sin(\theta)z_2 + i\left(\sin(\theta)z_1 + \cos(\theta)z_2\right)\right] \\ &= \begin{bmatrix} \cos(\theta)z_1 - \sin(\theta)z_2 \\ \sin(\theta)z_1 + \cos(\theta)z_2 \end{bmatrix} = \begin{bmatrix} \cos(\theta) & -\sin(\theta) \\ \sin(\theta) & \cos(\theta) \end{bmatrix} \begin{bmatrix} z_1 \\ z_2 \end{bmatrix}. \end{aligned} \tag{A.48}$$

Lemma A.1 allows us to rewrite the solution of Eq. (A.46) as:

$$\begin{aligned} \mathcal{I}_q(v_m^i, e_{v_m}) &= T^{-1}\left[T\left(\boldsymbol{W}_q v_m^i,\right) \odot \exp\left\{i\phi(e_{v_m})\right\}\right] \\ &= \boldsymbol{R}\left(\phi(e_{v_m})\right) \boldsymbol{W}_q v_m^i \\ &= \boldsymbol{R}\left(\phi(e_{v_m})\right) q_{v_m}^i, \end{aligned} \tag{A.49}$$

where $\boldsymbol{R}$ is defined in Eq. (26) in Section 3.5, in which $\phi(e_{v_m})$ is replaced with $\boldsymbol{\varphi}_{v_m} := c e_{v_m}$. Put together, we reach the solutions for $\mathcal{I}_q$ and $\mathcal{I}_k$ in Eq. (24) in Section 3.5, whose validity in satisfying Eq. (23) are demonstrated in the following proposition.

**Proposition A.7.** *Given $\boldsymbol{v}_m^i, \boldsymbol{v}_n^i, \boldsymbol{e}_{v_m}, \boldsymbol{e}_{v_n}$, the $\mathcal{I}_q$ and $\mathcal{I}_k$ given in Eq. (24) satisfy:*

$$\langle \mathcal{I}_q(\boldsymbol{v}_m^i, \boldsymbol{e}_{v_m}), \mathcal{I}_k(\boldsymbol{v}_n^i, \boldsymbol{e}_{v_n}) \rangle = \mathcal{A}\left(\boldsymbol{v}_m^i, \boldsymbol{v}_n^i, \gamma(\boldsymbol{e}_{v_m}, \boldsymbol{e}_{v_n})\right) = (\boldsymbol{q}_{v_m}^i)^\top \boldsymbol{R}(\boldsymbol{\varphi}_{v_n} - \boldsymbol{\varphi}_{v_m}) \boldsymbol{k}_{v_n}^i, \quad \text{(A.50)}$$

*where $\gamma(\boldsymbol{e}_{v_m}, \boldsymbol{e}_{v_n}) := \boldsymbol{e}_{v_n} - \boldsymbol{e}_{v_m}$.*

*Proof.*

$$\begin{aligned}
\langle \mathcal{I}_q(\boldsymbol{v}_m^i, \boldsymbol{e}_{v_m}), \mathcal{I}_k(\boldsymbol{v}_n^i, \boldsymbol{e}_{v_n}) \rangle &= \left[ \boldsymbol{R}(\boldsymbol{\varphi}_{v_m}) \boldsymbol{W}_q \boldsymbol{v}_m^i \right]^\top \boldsymbol{R}(\boldsymbol{\varphi}_{v_n}) \boldsymbol{W}_k \boldsymbol{v}_n^i \\
&= {\boldsymbol{v}_m^i}^\top \boldsymbol{W}_q^\top \boldsymbol{R}(\boldsymbol{\varphi}_{v_m})^\top \boldsymbol{R}(\boldsymbol{\varphi}_{v_n}) \boldsymbol{W}_k \boldsymbol{v}_n^i,
\end{aligned} \quad \text{(A.51)}$$

where, based on Lemma A.2, $\boldsymbol{R}(\boldsymbol{\varphi}_{v_m})^\top \boldsymbol{R}(\boldsymbol{\varphi}_{v_n})$ can be written as:

$$\begin{aligned}
&\boldsymbol{R}(\boldsymbol{\varphi}_{v_m})^\top \boldsymbol{R}(\boldsymbol{\varphi}_{v_n}) \\
&= \begin{bmatrix} \boldsymbol{r}(\varphi_{v_m}^{(1)}) & \boldsymbol{0} & \cdots & \boldsymbol{0} \\ \boldsymbol{0} & \boldsymbol{r}(\varphi_{v_m}^{(2)}) & \cdots & \boldsymbol{0} \\ \vdots & \vdots & \ddots & \boldsymbol{0} \\ \boldsymbol{0} & \boldsymbol{0} & \cdots & \boldsymbol{r}(\varphi_{v_m}^{(d)}) \end{bmatrix}^\top \begin{bmatrix} \boldsymbol{r}(\varphi_{v_n}^{(1)}) & \boldsymbol{0} & \cdots & \boldsymbol{0} \\ \boldsymbol{0} & \boldsymbol{r}(\varphi_{v_n}^{(2)}) & \cdots & \boldsymbol{0} \\ \vdots & \vdots & \ddots & \boldsymbol{0} \\ \boldsymbol{0} & \boldsymbol{0} & \cdots & \boldsymbol{r}(\varphi_{v_n}^{(d)}) \end{bmatrix} \\
&= \begin{bmatrix} \boldsymbol{r}(\varphi_{v_m}^{(1)})^\top \boldsymbol{r}(\varphi_{v_n}^{(1)}) & \boldsymbol{0} & \cdots & \boldsymbol{0} \\ \boldsymbol{0} & \boldsymbol{r}(\varphi_{v_m}^{(2)})^\top \boldsymbol{r}(\varphi_{v_n}^{(2)}) & \cdots & \boldsymbol{0} \\ \vdots & \vdots & \ddots & \boldsymbol{0} \\ \boldsymbol{0} & \boldsymbol{0} & \cdots & \boldsymbol{r}(\varphi_{v_m}^{(d)})^\top \boldsymbol{r}(\varphi_{v_n}^{(d)}) \end{bmatrix} \\
&= \begin{bmatrix} \boldsymbol{r}(\varphi_{v_n}^{(1)} - \varphi_{v_m}^{(1)}) & \boldsymbol{0} & \cdots & \boldsymbol{0} \\ \boldsymbol{0} & \boldsymbol{r}(\varphi_{v_n}^{(2)} - \varphi_{v_m}^{(2)}) & \cdots & \boldsymbol{0} \\ \vdots & \vdots & \ddots & \boldsymbol{0} \\ \boldsymbol{0} & \boldsymbol{0} & \cdots & \boldsymbol{r}(\varphi_{v_n}^{(d)} - \varphi_{v_m}^{(d)}) \end{bmatrix} \\
&= \boldsymbol{R}(\boldsymbol{\varphi}_{v_n} - \boldsymbol{\varphi}_{v_m}) = \boldsymbol{R}\left[ c\left(\boldsymbol{e}_{v_n} - \boldsymbol{e}_{v_m}\right) \right],
\end{aligned} \quad \text{(A.52)}$$

where $\boldsymbol{r}(\varphi_v^{(t)})$ is defined in Eq. (26) in Section 3.5. Then, Eq. (A.51) reads:

$$\langle \mathcal{I}_q(\boldsymbol{v}_m^i, \boldsymbol{e}_{v_m}), \mathcal{I}_k(\boldsymbol{v}_n^i, \boldsymbol{e}_{v_n}) \rangle = {\boldsymbol{v}_m^i}^\top \boldsymbol{W}_q^\top \boldsymbol{R}\left[ c\gamma(\boldsymbol{e}_{v_m}, \boldsymbol{e}_{v_n}) \right] \boldsymbol{W}_k \boldsymbol{v}_n^i = \mathcal{A}\left(\boldsymbol{v}_m^i, \boldsymbol{v}_n^i, \gamma(\boldsymbol{e}_{v_m}, \boldsymbol{e}_{v_n})\right). \quad \text{(A.53)}$$

This completes the proof. $\square$

**Lemma A.2.** *The following equation always exists:*

$$\begin{bmatrix} \cos(\alpha) & -\sin(\alpha) \\ \sin(\alpha) & \cos(\alpha) \end{bmatrix}^\top \begin{bmatrix} \cos(\beta) & -\sin(\beta) \\ \sin(\beta) & \cos(\beta) \end{bmatrix} = \begin{bmatrix} \cos(\beta - \alpha) & -\sin(\beta - \alpha) \\ \sin(\beta - \alpha) & \cos(\beta - \alpha) \end{bmatrix}, \forall \alpha, \beta \in \mathbb{R}. \quad \text{(A.54)}$$

**Remark A.1.** *Since $\boldsymbol{\varphi}_{v_m} := c\boldsymbol{e}_{v_m}$ and $\boldsymbol{e}_{v_m} \in \mathcal{B}^d$ lies within the unit Poincaré ball, $\boldsymbol{\varphi}_{v_m}$ is bounded for any $v_m$. $c$ is a scale factor to control the range of angles. Here, we set $c = \pi/4$ to ensure that $\boldsymbol{\varphi}_{v_m}$ is a small but not negligible angle constrained to $[-\pi/4, \pi/4]$. This also brings several valuable properties as demonstrated in Supps. A.4 to A.6.*

**Remark A.2.** *Since Eq. (A.55) is sparse, it can be efficiently computed as [18]:*

$$\boldsymbol{R}(\boldsymbol{\varphi}_{v_m}) \boldsymbol{v} = \begin{bmatrix} \cos(\varphi_{v_m}^{(1)}) \\ \cos(\varphi_{v_m}^{(1)}) \\ \cos(\varphi_{v_m}^{(2)}) \\ \cos(\varphi_{v_m}^{(2)}) \\ \vdots \\ \cos(\varphi_{v_m}^{(D/2)}) \\ \cos(\varphi_{v_m}^{(D/2)}) \end{bmatrix} \odot \begin{bmatrix} v^{(1)} \\ v^{(2)} \\ v^{(3)} \\ v^{(4)} \\ \vdots \\ v^{(D-1)} \\ v^{(D)} \end{bmatrix} + \begin{bmatrix} \sin(\varphi_{v_m}^{(1)}) \\ \sin(\varphi_{v_m}^{(1)}) \\ \sin(\varphi_{v_m}^{(2)}) \\ \sin(\varphi_{v_m}^{(2)}) \\ \vdots \\ \sin(\varphi_{v_m}^{(D/2)}) \\ \sin(\varphi_{v_m}^{(D/2)}) \end{bmatrix} \odot \begin{bmatrix} -v^{(2)} \\ v^{(1)} \\ -v^{(4)} \\ v^{(3)} \\ \vdots \\ -v^{(D)} \\ v^{(D-1)} \end{bmatrix}, \quad \text{(A.55)}$$

*where $\odot$ is the element-wise product.*

## A.4  Causal Distance-Induced Attention Attenuation

**Proposition 4.1.** *Given $\boldsymbol{v}_m^i$ and $\boldsymbol{v}_n^i$, the attention scoring function $\mathcal{A}$ in Eq. (25) is bounded by $\mathcal{A}^+ > 0$ and $\mathcal{A}^- < 0$, satisfying:*

$$\frac{\partial \mathcal{A}^+ \left(\boldsymbol{v}_m^i, \boldsymbol{v}_n^i, \boldsymbol{e}_{v_m} - \boldsymbol{e}_{v_n}\right)}{\partial d_p(\boldsymbol{e}_{v_m}, \boldsymbol{e}_{v_n})} \leq 0, \quad \frac{\partial \mathcal{A}^- \left(\boldsymbol{v}_m^i, \boldsymbol{v}_n^i, \boldsymbol{e}_{v_m} - \boldsymbol{e}_{v_n}\right)}{\partial d_p(\boldsymbol{e}_{v_m}, \boldsymbol{e}_{v_n})} \geq 0, \tag{27}$$

*where $d_p(\boldsymbol{e}_{v_m}, \boldsymbol{e}_{v_n})$ is the distance between $\boldsymbol{e}_{v_m}$ and $\boldsymbol{e}_{v_n}$ on the Poincaré ball manifold, computed as:*

$$d_p(\boldsymbol{e}_{v_m}, \boldsymbol{e}_{v_n}) = \operatorname{arcosh}\left(1 + 2\frac{\|\boldsymbol{e}_{v_m} - \boldsymbol{e}_{v_n}\|^2}{(1 - \|\boldsymbol{e}_{v_m}\|^2)(1 - \|\boldsymbol{e}_{v_n}\|^2)}\right). \tag{28}$$

*The functions $\mathcal{A}^+$ and $\mathcal{A}^-$ are given in Supp. A.4.*

*Proof.* We first prove that for any $\boldsymbol{v}_m^i, \boldsymbol{v}_n^i$ and $\boldsymbol{\gamma}(\boldsymbol{e}_{v_m}, \boldsymbol{e}_{v_n})$, $\mathcal{A}$ is bounded by two functions $\mathcal{A}^+, \mathcal{A}^-$.

**(1) Boundedness.**  Recall that $\boldsymbol{q}_{v_m}^i = \boldsymbol{W}_q \boldsymbol{v}_m^i = \left(q_{v_m}^i{}^{(1)}, q_{v_m}^i{}^{(2)}, \cdots, q_{v_m}^i{}^{(D)}\right)^\top$ and $\boldsymbol{k}_{v_n}^i = \boldsymbol{W}_k \boldsymbol{v}_n^i = \left(k_{v_n}^i{}^{(1)}, k_{v_n}^i{}^{(2)}, \cdots, k_{v_n}^i{}^{(D)}\right)^\top$. Then, we have:

$$
\begin{aligned}
\mathcal{A}\left(\boldsymbol{v}_m^i, \boldsymbol{v}_n^i, \boldsymbol{e}_{v_m} - \boldsymbol{e}_{v_n}\right) &= \boldsymbol{q}_{v_m}^i{}^\top \boldsymbol{R}(\boldsymbol{\varphi}_{v_n} - \boldsymbol{\varphi}_{v_m})\boldsymbol{k}_{v_n}^i \\
&= \sum_{t=1}^d (q_{v_m}^i{}^{(2t-1)}, q_{v_m}^i{}^{(2t)})\boldsymbol{r}(\varphi_{v_n}^{(t)} - \varphi_{v_m}^{(t)})(k_{v_n}^i{}^{(2t-1)}, k_{v_n}^i{}^{(2t)})^\top \\
&= \sum_{t=1}^d \begin{bmatrix} \cos(\varphi_{v_n}^{(t)} - \varphi_{v_m}^{(t)})q_{v_m}^i{}^{(2t-1)} + \sin(\varphi_{v_n}^{(t)} - \varphi_{v_m}^{(t)})q_{v_m}^i{}^{(2t)} \\ -\sin(\varphi_{v_n}^{(t)} - \varphi_{v_m}^{(t)})q_{v_m}^i{}^{(2t-1)} + \cos(\varphi_{v_n}^{(t)} - \varphi_{v_m}^{(t)})q_{v_m}^i{}^{(2t)} \end{bmatrix}^\top \begin{bmatrix} k_{v_n}^i{}^{(2t-1)} \\ k_{v_n}^i{}^{(2t)} \end{bmatrix} \\
&= \sum_{t=1}^d \alpha_i^{(t)} \cos(\varphi_{v_m}^{(t)} - \varphi_{v_n}^{(t)}) + \beta_i^{(t)} \sin(\varphi_{v_m}^{(t)} - \varphi_{v_n}^{(t)}),
\end{aligned}
\tag{A.56}
$$

where $\alpha_i^{(t)} := q_{v_m}^i{}^{(2t-1)}k_{v_n}^i{}^{(2t-1)} + q_{v_m}^i{}^{(2t)}k_{v_n}^i{}^{(2t)}$, $\beta_i^{(t)} := q_{v_m}^i{}^{(2t)}k_{v_n}^i{}^{(2t-1)} - q_{v_m}^i{}^{(2t-1)}k_{v_n}^i{}^{(2t)}$. Eq. (A.56) is bounded as:

$$
\begin{aligned}
\mathcal{A}\left(\boldsymbol{v}_m^i, \boldsymbol{v}_n^i, \boldsymbol{e}_{v_m} - \boldsymbol{e}_{v_n}\right) &\leq \sum_{t=1}^d |\alpha_i^{(t)}| \cos(\varphi_{v_m}^{(t)} - \varphi_{v_n}^{(t)}) + |\beta_i^{(t)}| \\
&\leq |\alpha_i^*| \sum_{t=1}^d \cos(|\varphi_{v_m}^{(t)} - \varphi_{v_n}^{(t)}|) + \sum_{t=1}^d |\beta_i^{(t)}|,
\end{aligned}
\tag{A.57}
$$

where $|\alpha_i^*| := \max_t |\alpha_i^{(t)}|$. As $\varphi_{v_m}^{(t)}, \varphi_{v_n}^{(t)} \in [-\pi/4, \pi/4]$, we have $|\varphi_{v_m}^{(t)} - \varphi_{v_n}^{(t)}| \in [0, \pi/2]$. Over this interval, $\cos(|\varphi_{v_m}^{(t)} - \varphi_{v_n}^{(t)}|)$ is concave. Thus, Jensen's inequality indicates the inequality:

$$
\begin{aligned}
\frac{1}{d}\sum_{t=1}^d \cos(|\varphi_{v_m}^{(t)} - \varphi_{v_n}^{(t)}|) &\leq \cos\left(\frac{1}{d}\sum_{t=1}^d |\varphi_{v_m}^{(t)} - \varphi_{v_n}^{(t)}|\right) \\
&\leq \cos\left(\frac{1}{d}\|\boldsymbol{\varphi}_{v_m} - \boldsymbol{\varphi}_{v_n}\|\right),
\end{aligned}
\tag{A.58}
$$

which leads to the upper bound function of Eq. (A.56) as:

$$\mathcal{A}^+ \left(\boldsymbol{v}_m^i, \boldsymbol{v}_n^i, \boldsymbol{e}_{v_m} - \boldsymbol{e}_{v_n}\right) := (|\alpha_i^*|d) \cos\left(\frac{1}{d}\|\boldsymbol{\varphi}_{v_m} - \boldsymbol{\varphi}_{v_n}\|\right) + \sum_{t=1}^d |\beta_i^{(t)}|. \tag{A.59}$$

Since $\boldsymbol{\varphi}_{v_m} = \frac{\pi}{4}\boldsymbol{e}_{v_m}$, $\boldsymbol{\varphi}_{v_n} = \frac{\pi}{4}\boldsymbol{e}_{v_n}$ and $\boldsymbol{e}_{v_m}, \boldsymbol{e}_{v_n} \in \mathcal{B}^d := \{\boldsymbol{e} \in \mathbb{R}^d : \|\boldsymbol{e}\| < 1\}$, we have $\|\boldsymbol{\varphi}_{v_m} - \boldsymbol{\varphi}_{v_n}\| = \frac{\pi}{4}\|\boldsymbol{e}_{v_m} - \boldsymbol{e}_{v_n}\| \in [0, \pi/2]$, which further leads to $\mathcal{A}^+ > 0$. Similarly, the lower bound function $\mathcal{A}^- < 0$ can be defined as:

$$\mathcal{A}^- \left(\boldsymbol{v}_m^i, \boldsymbol{v}_n^i, \boldsymbol{e}_{v_m} - \boldsymbol{e}_{v_n}\right) := -(|\alpha_i^*|d) \cos\left(\frac{1}{d}\|\boldsymbol{\varphi}_{v_m} - \boldsymbol{\varphi}_{v_n}\|\right) - \sum_{t=1}^d |\beta_i^{(t)}|. \tag{A.60}$$

Subsequently, we prove that both $\mathcal{A}^+$ and $\mathcal{A}^-$ attenuate as $d_p(\boldsymbol{e}_{v_m}, \boldsymbol{e}_{v_n})$ increases.

**(2) Attenuation.** The partial derivative of $\mathcal{A}^+$ with respect to $\|\boldsymbol{e}_{v_m} - \boldsymbol{e}_{v_n}\|$ reads:

$$
\frac{\partial \mathcal{A}^+ \left(\boldsymbol{v}_m^i, \boldsymbol{v}_n^i, \boldsymbol{e}_{v_m} - \boldsymbol{e}_{v_n}\right)}{\partial \|\boldsymbol{e}_{v_m} - \boldsymbol{e}_{v_n}\|} = (d|\alpha_i^*|) \frac{\partial \cos\left(\frac{1}{d}\|\boldsymbol{\varphi}_{v_m} - \boldsymbol{\varphi}_{v_n}\|\right)}{\partial \|\boldsymbol{\varphi}_{v_m} - \boldsymbol{\varphi}_{v_n}\|} \frac{\partial \|\boldsymbol{\varphi}_{v_m} - \boldsymbol{\varphi}_{v_n}\|}{\partial \|\boldsymbol{e}_{v_m} - \boldsymbol{e}_{v_n}\|}
$$
$$
= -(\frac{\pi d}{4}|\alpha_i^*|) \cdot \sin\left(\frac{1}{d}\|\boldsymbol{\varphi}_{v_m} - \boldsymbol{\varphi}_{v_n}\|\right).
\tag{A.61}
$$

Given the Poincaré distance in Eq. (28), we have:

$$
\frac{\partial d_p(\boldsymbol{e}_{v_m}, \boldsymbol{e}_{v_n})}{\partial \|\boldsymbol{e}_{v_m} - \boldsymbol{e}_{v_n}\|} = \frac{\partial \operatorname{arcosh}\left(1 + 2C^{-1}\|\boldsymbol{e}_{v_m} - \boldsymbol{e}_{v_n}\|^2\right)}{\partial \|\boldsymbol{e}_{v_m} - \boldsymbol{e}_{v_n}\|}
$$
$$
= \frac{4C^{-1}\|\boldsymbol{e}_{v_m} - \boldsymbol{e}_{v_n}\|}{\sqrt{\left(1 + 2C^{-1}\|\boldsymbol{e}_{v_m} - \boldsymbol{e}_{v_n}\|^2\right)^2 - 1}} = \frac{2}{\sqrt{\|\boldsymbol{e}_{v_m} - \boldsymbol{e}_{v_n}\|^2 + C}},
\tag{A.62}
$$

where $C = (1 - \|\boldsymbol{e}_{v_m}\|^2)(1 - \|\boldsymbol{e}_{v_n}\|^2)$. This leads to:

$$
\frac{\partial \mathcal{A}^+ \left(\boldsymbol{v}_m^i, \boldsymbol{v}_n^i, \boldsymbol{e}_{v_m} - \boldsymbol{e}_{v_n}\right)}{\partial d_p(\boldsymbol{e}_{v_m}, \boldsymbol{e}_{v_n})} = \frac{\partial \mathcal{A}^+ \left(\boldsymbol{v}_m^i, \boldsymbol{v}_n^i, \boldsymbol{e}_{v_m} - \boldsymbol{e}_{v_n}\right)}{\partial \|\boldsymbol{e}_{v_m} - \boldsymbol{e}_{v_n}\|} \frac{\partial \|\boldsymbol{e}_{v_m} - \boldsymbol{e}_{v_n}\|}{\partial d_p(\boldsymbol{e}_{v_m}, \boldsymbol{e}_{v_n})}
$$
$$
= -\frac{\pi d}{4}|\alpha_i^*| \sin\left(\frac{1}{d}\|\boldsymbol{\varphi}_{v_m} - \boldsymbol{\varphi}_{v_n}\|\right) \cdot \frac{\sqrt{\|\boldsymbol{e}_{v_m} - \boldsymbol{e}_{v_n}\|^2 + C}}{2} \leq 0
\tag{A.63}
$$

where $\|\boldsymbol{\varphi}_{v_m} - \boldsymbol{\varphi}_{v_n}\| \in [0, \pi/2]$. Similarly, we can prove the attenuation for $\mathcal{A}^-$. This completes the proof. $\square$

**Remark A.3.** *As the causal distance $d_p(\boldsymbol{e}v_m, \boldsymbol{e}v_n) \to +\infty$, both $\mathcal{A}^+$ and $\mathcal{A}^-$ attenuate and converge towards smaller magnitudes (though not necessarily to 0). Since $\mathcal{A}$ is bounded between $\mathcal{A}^+$ and $\mathcal{A}^-$, its range of possible variation also shrinks significantly. In particular, when $\boldsymbol{q}$ and $\boldsymbol{k}$ are collinear, $\mathcal{A}$ exhibits a stronger attenuation property shown below.*

**Corollary A.1.** *Following the definition in Prop. 4.1, when $\boldsymbol{k} = c\boldsymbol{q}, c \in \mathbb{R}$ and $c \neq 0$, it exists:*

$$
\frac{\partial \mathcal{A}\left(\boldsymbol{v}_m^i, \boldsymbol{v}_n^i, \boldsymbol{e}_{v_m} - \boldsymbol{e}_{v_n}|\boldsymbol{k} = c\boldsymbol{q}\right)}{\partial d_p(\boldsymbol{e}_{v_m}, \boldsymbol{e}_{v_n})} \operatorname{sgn}(c) \leq 0,
\tag{A.64}
$$

*where $\operatorname{sgn}(c) := c/|c|$ is a sign function.*

*Proof.* According to Eq. (A.56), given $\boldsymbol{k} = c\boldsymbol{q}$, we have:

$$
\mathcal{A}\left(\boldsymbol{v}_m^i, \boldsymbol{v}_n^i, \boldsymbol{e}_{v_m} - \boldsymbol{e}_{v_n}|\boldsymbol{k} = c\boldsymbol{q}\right) = \sum_{t=1}^d c\left[\left(q_{v_m}^{i\ (2t-1)}\right)^2 + \left(q_{v_m}^{i\ (2t)}\right)^2\right] \cos(\varphi_{v_m}^{(t)} - \varphi_{v_n}^{(t)}).
\tag{A.65}
$$

When $c > 0$, we have $\alpha_i^{(t)} := c\left[\left(q_{v_m}^{i\ (2t-1)}\right)^2 + \left(q_{v_m}^{i\ (2t)}\right)^2\right] \geq 0$, and:

$$
\frac{\partial \mathcal{A}\left(\boldsymbol{v}_m^i, \boldsymbol{v}_n^i, \boldsymbol{e}_{v_m} - \boldsymbol{e}_{v_n}|\boldsymbol{k} = c\boldsymbol{q}\right)}{\partial \|\boldsymbol{e}_{v_m} - \boldsymbol{e}_{v_n}\|} = \frac{\partial \mathcal{A}\left(\boldsymbol{v}_m^i, \boldsymbol{v}_n^i, \boldsymbol{e}_{v_m} - \boldsymbol{e}_{v_n}|\boldsymbol{k} = c\boldsymbol{q}\right)}{\partial \|\boldsymbol{\varphi}_{v_m} - \boldsymbol{\varphi}_{v_n}\|} \frac{\partial \|\boldsymbol{\varphi}_{v_m} - \boldsymbol{\varphi}_{v_n}\|}{\partial \|\boldsymbol{e}_{v_m} - \boldsymbol{e}_{v_n}\|}
$$
$$
= \frac{\pi}{4} \sum_{t=1}^d \frac{\partial \ \alpha_i^{(t)} \cos(\varphi_{v_m}^{(t)} - \varphi_{v_n}^{(t)})}{\partial (\varphi_{v_m}^{(t)} - \varphi_{v_n}^{(t)})} \frac{\partial (\varphi_{v_m}^{(t)} - \varphi_{v_n}^{(t)})}{\partial \|\boldsymbol{\varphi}_{v_m} - \boldsymbol{\varphi}_{v_n}\|} = \frac{\pi}{4} \sum_{t=1}^d -\alpha_i^{(t)} \sin(\varphi_{v_m}^{(t)} - \varphi_{v_n}^{(t)}) \frac{\|\boldsymbol{\varphi}_{v_m} - \boldsymbol{\varphi}_{v_n}\|}{\varphi_{v_m}^{(t)} - \varphi_{v_n}^{(t)}}.
\tag{A.66}
$$

Combining Eq. (A.66) and Eq. (A.62), we obtain:

$$
\frac{\partial \mathcal{A}\left(\boldsymbol{v}_m^i, \boldsymbol{v}_n^i, \boldsymbol{e}_{v_m} - \boldsymbol{e}_{v_n}|\boldsymbol{k} = c\boldsymbol{q}, c > 0\right)}{\partial d_p(\boldsymbol{e}_{v_m}, \boldsymbol{e}_{v_n})}
$$
$$
= -\frac{\pi\sqrt{\|\boldsymbol{e}_{v_m} - \boldsymbol{e}_{v_n}\|^2 + C}}{8} \cdot \|\boldsymbol{\varphi}_{v_m} - \boldsymbol{\varphi}_{v_n}\| \sum_{t=1}^d \frac{\alpha_i^{(t)} \sin(\varphi_{v_m}^{(t)} - \varphi_{v_n}^{(t)})}{\varphi_{v_m}^{(t)} - \varphi_{v_n}^{(t)}} \leq 0.
\tag{A.67}
$$

Similarly, when $c < 0$, we have $\alpha_i^{(t)} < 0$, and:

$$\frac{\partial \mathcal{A}\left(\boldsymbol{v}_m^i, \boldsymbol{v}_n^i, \boldsymbol{e}_{v_m} - \boldsymbol{e}_{v_n} | \boldsymbol{k} = c\boldsymbol{q}, c < 0\right)}{\partial d_p(\boldsymbol{e}_{v_m}, \boldsymbol{e}_{v_n})} \geq 0. \tag{A.68}$$

This completes the proof. □

## A.5 Causal Generality-Induced Attention Attenuation

**Proposition 4.2.** *Given $\boldsymbol{v}_m^i$, $\boldsymbol{v}_n^i$, and fixed causal distance $d_p(\boldsymbol{e}_{v_m}, \boldsymbol{e}_{v_n})$ in the Poincaré ball manifold defined in Def. 4.1, $\mathcal{A}^+$ and $\mathcal{A}^-$ satisfy:*

$$\frac{\partial \mathcal{A}^+\left(\boldsymbol{v}_m^i, \boldsymbol{v}_n^i, \boldsymbol{e}_{v_m} - \boldsymbol{e}_{v_n}\right)}{\partial \psi_{v_m}} \leq 0, \quad \frac{\partial \mathcal{A}^-\left(\boldsymbol{v}_m^i, \boldsymbol{v}_n^i, \boldsymbol{e}_{v_m} - \boldsymbol{e}_{v_n}\right)}{\partial \psi_{v_m}} \geq 0. \tag{29}$$

*The same holds for $\psi_{v_n}$.*

*Proof.* As proved in Supp. A.4, $\mathcal{A}$ is bounded by the functions $\mathcal{A}^+$ and $\mathcal{A}^-$ in Eqs. (A.59) and (A.60). We have also shown in Eq. (A.61) that:

$$\frac{\partial \mathcal{A}^+\left(\boldsymbol{v}_m^i, \boldsymbol{v}_n^i, \boldsymbol{e}_{v_m} - \boldsymbol{e}_{v_n}\right)}{\partial \|\boldsymbol{e}_{v_m} - \boldsymbol{e}_{v_n}\|} = -(\frac{\pi d}{4}|\alpha_i^*|) \cdot \sin\left(\frac{1}{d}\|\boldsymbol{\varphi}_{v_m} - \boldsymbol{\varphi}_{v_n}\|\right). \tag{A.69}$$

The Euclidean norm of $\boldsymbol{e}_{v_m}$ and $\boldsymbol{e}_{v_n}$ can be expressed in terms of their Poincaré distance based on Eq. (28) as:

$$\|\boldsymbol{e}_{v_m} - \boldsymbol{e}_{v_n}\| = \sqrt{\frac{1}{2}\left[\cosh(d_p(\boldsymbol{e}_{v_m}, \boldsymbol{e}_{v_n})) - 1\right](1 - \|\boldsymbol{e}_{v_m}\|^2)(1 - \|\boldsymbol{e}_{v_n}\|^2)}. \tag{A.70}$$

Given fixed $d_p(\boldsymbol{e}_{v_m}, \boldsymbol{e}_{v_n})$, $C = \frac{1}{2}\left[\cosh(d_p(\boldsymbol{e}_{v_m}, \boldsymbol{e}_{v_n})) - 1\right] \geq 0$ represents a non-negative constant. Given $\psi_{v_m} = 1 - \|\boldsymbol{e}_{v_m}\|$, we have:

$$\begin{aligned}
\frac{\partial \|\boldsymbol{e}_{v_m} - \boldsymbol{e}_{v_n}\|}{\partial \psi_{v_m}} &= \frac{\partial \|\boldsymbol{e}_{v_m} - \boldsymbol{e}_{v_n}\|}{\partial (1 - \|\boldsymbol{e}_{v_m}\|)} = -\frac{\partial \|\boldsymbol{e}_{v_m} - \boldsymbol{e}_{v_n}\|}{\partial \|\boldsymbol{e}_{v_m}\|} \\
&= -\frac{\partial \sqrt{C(1 - \|\boldsymbol{e}_{v_m}\|^2)(1 - \|\boldsymbol{e}_{v_n}\|^2)}}{\partial \|\boldsymbol{e}_{v_m}\|} = \frac{\|\boldsymbol{e}_{v_m}\|}{\sqrt{1 - \|\boldsymbol{e}_{v_m}\|^2}}\sqrt{C(1 - \|\boldsymbol{e}_{v_n}\|^2)},
\end{aligned} \tag{A.71}$$

Combining Eq. (A.69) and Eq. (A.71), we obtain:

$$\begin{aligned}
&\frac{\partial \mathcal{A}^+\left(\boldsymbol{v}_m^i, \boldsymbol{v}_n^i, \boldsymbol{e}_{v_m} - \boldsymbol{e}_{v_n}\right)}{\partial \psi_{v_m}} \\
&= -(\frac{\pi d}{4}|\alpha_i^*|) \cdot \sin\left(\frac{1}{d}\|\boldsymbol{\varphi}_{v_m} - \boldsymbol{\varphi}_{v_n}\|\right) \cdot \frac{\|\boldsymbol{e}_{v_m}\|}{\sqrt{1 - \|\boldsymbol{e}_{v_m}\|^2}}\sqrt{C(1 - \|\boldsymbol{e}_{v_n}\|^2)} \leq 0, \quad \text{(A.72)}
\end{aligned}$$

where $\|\boldsymbol{\varphi}_{v_m} - \boldsymbol{\varphi}_{v_n}\| \in [0, \pi/2]$. Similarly, we can prove the attenuation for $\mathcal{A}^-$. The same proof also applies to $\psi_{v_n}$. This completes the proof. □

**Corollary A.2.** *When $\psi_{v_m} := 1 - \|\boldsymbol{e}_{v_m}\| \to 1$ and the causal distance $d_p(\boldsymbol{e}_{v_m}, \boldsymbol{e}_{v_n})$ is fixed, the upper bound function $\mathcal{A}^+\left(\boldsymbol{v}_m^i, \boldsymbol{v}_n^i, \boldsymbol{e}_{v_m} - \boldsymbol{e}_{v_n}\right)$ and lower bound function $\mathcal{A}^-\left(\boldsymbol{v}_m^i, \boldsymbol{v}_n^i, \boldsymbol{e}_{v_m} - \boldsymbol{e}_{v_n}\right)$ attenuate towards constants $a$ and $-a$, respectively, where*

$$a := (|\alpha_i^*|d)\cos\left(\frac{\pi}{4d}\sqrt{\frac{C}{C+1}}\right) + \sum_{t=1}^d |\beta_i^{(t)}|, \tag{A.73}$$

*where $C = \frac{1}{2}\left[\cosh(d_p(\boldsymbol{e}_{v_m}, \boldsymbol{e}_{v_n})) - 1\right] \geq 0$ represents a non-negative constant. Moreover, $a$ decreases monotonically with increasing causal distance.*

*Proof.* Since $e_{v_m} \to 0$ as $\psi_{v_m} \to 1$, we have the following limits of Eq. (A.70) :

$$\lim_{\psi_{v_m} \to 1} \|e_{v_m} - e_{v_n}\|^2 = \lim_{\psi_{v_m} \to 1} \sqrt{C(1 - \|e_{v_m}\|^2)(1 - \|e_{v_n}\|^2)}$$

$$\implies \lim_{\psi_{v_m} \to 1} \|e_{v_m} - e_{v_n}\| = \lim_{\psi_{v_m} \to 1} \|e_{v_n}\| = \lim_{\psi_{v_m} \to 1} \sqrt{C(1 - \|e_{v_n}\|^2)}. \tag{A.74}$$

Let $\|\overrightarrow{e_{v_n}}\| := \lim_{\psi_{v_m} \to 1} \|e_{v_n}\|$. Eq. (A.74) yields:

$$\frac{1}{C} \|\overrightarrow{e_{v_n}}\|^2 = 1 - \|\overrightarrow{e_{v_n}}\|^2$$

$$\implies \|\overrightarrow{e_{v_n}}\| = \sqrt{\frac{C}{C+1}} \tag{A.75}$$

Hence,

$$\lim_{\psi_{v_m} \to 1} \|\varphi_{v_m} - \varphi_{v_n}\| = \frac{\pi}{4} \sqrt{\frac{C}{C+1}}$$

$$\implies \lim_{\psi_{v_m} \to 1} \cos\left(\frac{1}{d} \|\varphi_{v_m} - \varphi_{v_n}\|\right) = \cos\left(\frac{\pi}{4d} \sqrt{\frac{C}{C+1}}\right) \tag{A.76}$$

$$\implies \lim_{\psi_{v_m} \to 1} \mathcal{A}^+\left(v_m^i, v_n^i, e_{v_m} - e_{v_n}\right) = (|\alpha_i^*|d) \cos\left(\frac{\pi}{4d} \sqrt{\frac{C}{C+1}}\right) + \sum_{t=1}^{d} |\beta_i^{(t)}| = a.$$

According to Prop. 4.2, $\mathcal{A}^+\left(v_m^i, v_n^i, e_{v_m} - e_{v_n}\right)$ asymptotically attenuates towards $a$ as $\psi_{v_m} \to 1$. Similarly, the asymptotical attenuation of $\mathcal{A}^-\left(v_m^i, v_n^i, e_{v_m} - e_{v_n}\right)$ towards $-a$ as $\psi_{v_m} \to 1$ can be proved. Finally, the monotonic decrease of $a$ w.r.t $d_p(e_{v_m}, e_{v_n})$ can be told from $a \propto f(g(C))$, where $f := \cos(\cdot)$ and $g := \frac{\pi}{4d} \sqrt{1 - \frac{1}{C+1}} \in [0, \frac{\pi}{4}]$ . It is straightforward to verify that $f$ is monotonically decreasing in $g$, and $g$ is monotonically increasing in $C$, which itself increases with $d_p(e_{v_m}, e_{v_n})$. Therefore, $a$ decreases monotonically as $d_p(e_{v_m}, e_{v_n})$ increases. This behavior aligns with the expectation that the range of the attention score between $v_m$ and $v_n$ shrinks around 0 as their causal distance grows. This completes the proof.

$\square$

**Corollary A.3.** *Following the definition in Prop. 4.2, when $k = cq, c \in \mathbb{R}$ and $c \neq 0$, it exists:*

$$\frac{\partial \mathcal{A}\left(v_m^i, v_n^i, e_{v_m} - e_{v_n} | k = cq\right)}{\partial \psi_{v_m}} \text{sgn}(c) \leq 0, \tag{A.77}$$

*where $\text{sgn}(c) := c/|c|$ is a sign function. And the same holds true for $\psi_{v_n}$.*

*Proof.* The proof follows that of Corollary A.1. $\square$

## A.6 Robustness to Positional Disturbances

**Proposition 4.3.** *Assume that the noise-perturbed Poincaré ball positional encoding of $v_j$ can be represented as $e'_{v_j} := e_{v_j} + \varepsilon_j$, where $\varepsilon_j \sim \mathcal{N}(\mu, I_j)$ is a small random Gaussian disturbance with $\mu \in \mathbb{R}^d$ and $I_j = \text{diag}(\sigma_{j1}^2, \sigma_{j2}^2, \cdots, \sigma_{jd}^2)$. Then, the noise-disturbed attention score $\mathcal{A}'$ remains robust to such disturbances in three aspects, including **Distinguishability** (Prop. A.8), **Unbiasedness** (Prop. A.9), and **Asymptotic Convergence** (Prop. A.10).*

### A.6.1 Distinguishability

Here, distinguishability refers to the property that the attention score between two feature embeddings, differing only due to random noise, should remain larger than the score between embeddings of two truly distinct features. Importantly, this property should be preserved even when the positional embeddings are perturbed. Formally, this is captured in the following proposition.

**Proposition A.8.** *Given embeddings of two distinct features* $(\boldsymbol{v}_m^i, \boldsymbol{v}_n^i)$, *the noisy embedding* $\widetilde{\boldsymbol{v}}_m^i = \boldsymbol{v}_m^i + \boldsymbol{\delta}$, *where* $\boldsymbol{\delta} \in \mathbb{R}^D$ *is a random noise with zero mean and finite second moment, and the noise-perturbed positional encodings* $(\boldsymbol{e}'_{v_m}, \boldsymbol{e}'_{v_n})$, *it exists:*

$$\mathbb{E}_{\boldsymbol{v}_m^i, \boldsymbol{\delta}}\left[\mathcal{A}\left(\boldsymbol{v}_m^i, \tilde{\boldsymbol{v}}_m^i, \boldsymbol{e}'_{v_m} - \boldsymbol{e}'_{v_n}\right)\right] > \mathbb{E}_{\boldsymbol{v}_m^i, \boldsymbol{v}_n^i}\left[\mathcal{A}\left(\boldsymbol{v}_m^i, \boldsymbol{v}_n^i, \boldsymbol{e}'_{v_m} - \boldsymbol{e}'_{v_n}\right)\right]. \tag{A.78}$$

*Proof.* We first define:

$$\begin{aligned}
\mathcal{D}(\boldsymbol{v}_m^i, \boldsymbol{v}_n^i, \boldsymbol{\delta}, \boldsymbol{e}'_{v_m} - \boldsymbol{e}'_{v_n}) &:= \mathcal{A}\left(\boldsymbol{v}_m^i, \boldsymbol{v}_m^i + \boldsymbol{\delta}, \boldsymbol{e}'_{v_m} - \boldsymbol{e}'_{v_n}\right) - \mathcal{A}\left(\boldsymbol{v}_m^i, \boldsymbol{v}_n^i, \boldsymbol{e}'_{v_m} - \boldsymbol{e}'_{v_n}\right) \\
&= \boldsymbol{v}_m^{i\top} \boldsymbol{W}_q^\top \boldsymbol{R}(\boldsymbol{\varphi}'_{v_n} - \boldsymbol{\varphi}'_{v_m}) \boldsymbol{W}_k(\boldsymbol{v}_m^i + \boldsymbol{\delta}) - \boldsymbol{v}_m^{i\top} \boldsymbol{W}_q^\top \boldsymbol{R}(\boldsymbol{\varphi}'_{v_n} - \boldsymbol{\varphi}'_{v_m}) \boldsymbol{W}_k \boldsymbol{v}_n^i,
\end{aligned} \tag{A.79}$$

where $\boldsymbol{\varphi}'_{v_m} = c\boldsymbol{e}'_{v_m}, \boldsymbol{\varphi}'_{v_n} = c\boldsymbol{e}'_{v_n}$. Let $\boldsymbol{\mu} = \mathbb{E}(\boldsymbol{v}_m^i) = \mathbb{E}(\boldsymbol{v}_n^i), \boldsymbol{\Sigma} = \text{Cov}(\boldsymbol{v}_m)$. Since both $v_m$ and $v_n$ are randomly sampled from $\mathcal{V}$, for any $\boldsymbol{e}'_{v_m} - \boldsymbol{e}'_{v_n}$, the expectation of $\mathcal{D}$ satisfies:

$$\begin{aligned}
&\mathbb{E}_{\boldsymbol{v}_m^i, \boldsymbol{v}_n^i, \boldsymbol{\delta}}\left[\mathcal{D}(\boldsymbol{v}_m^i, \boldsymbol{v}_n^i, \boldsymbol{\delta}, \boldsymbol{e}'_{v_m} - \boldsymbol{e}'_{v_n})\right] \\
&= \mathbb{E}_{\boldsymbol{v}_m^i, \boldsymbol{v}_n^i, \boldsymbol{\delta}}\left[\boldsymbol{v}_m^{i\top} \boldsymbol{W}_q^\top \boldsymbol{R}(\boldsymbol{\varphi}'_{v_n} - \boldsymbol{\varphi}'_{v_m}) \boldsymbol{W}_k(\boldsymbol{v}_m^i + \boldsymbol{\delta}) - \boldsymbol{v}_m^{i\top} \boldsymbol{W}_q^\top \boldsymbol{R}(\boldsymbol{\varphi}'_{v_n} - \boldsymbol{\varphi}'_{v_m}) \boldsymbol{W}_k \boldsymbol{v}_n^i\right] \\
&= \mathbb{E}_{\boldsymbol{v}_m^i}\left[\boldsymbol{v}_m^{i\top} \boldsymbol{W}_q^\top \boldsymbol{R}(\boldsymbol{\varphi}'_{v_n} - \boldsymbol{\varphi}'_{v_m}) \boldsymbol{W}_k \boldsymbol{v}_m^i\right] - \mathbb{E}_{\boldsymbol{v}_m^i, \boldsymbol{v}_n^i}\left[\boldsymbol{v}_m^{i\top} \boldsymbol{W}_q^\top \boldsymbol{R}(\boldsymbol{\varphi}'_{v_n} - \boldsymbol{\varphi}'_{v_m}) \boldsymbol{W}_k \boldsymbol{v}_n^i\right] \\
&= \text{tr}\left[\boldsymbol{W}_q^\top \boldsymbol{R}(\boldsymbol{\varphi}'_{v_n} - \boldsymbol{\varphi}'_{v_m}) \boldsymbol{W}_k \boldsymbol{\Sigma}\right] = \text{tr}\left[\boldsymbol{R}(\boldsymbol{\varphi}'_{v_m} - \boldsymbol{\varphi}'_{v_n}) \boldsymbol{W}_q \boldsymbol{\Sigma} \boldsymbol{W}_k^\top\right] \\
&= \text{tr}\left[\boldsymbol{R}(\boldsymbol{\varphi}'_{v_m} - \boldsymbol{\varphi}'_{v_n}) \text{Cov}(\boldsymbol{W}_q \boldsymbol{v}_m^i, \boldsymbol{W}_k \boldsymbol{v}_m^i)\right] \\
&= \sum_{t=1}^{d}\left[\text{Cov}\left(q_{v_m}^{i\ (2t-1)}, k_{v_m}^{i\ (2t-1)}\right) + \text{Cov}\left(q_{v_m}^{i\ (2t)}, k_{v_m}^{i\ (2t)}\right)\right] \cos(\varphi'_{v_m}{}^{(t)} - \varphi'_{v_n}{}^{(t)}),
\end{aligned} \tag{A.80}$$

where the third equal sign is achieved based on Lemma A.3. Since $\boldsymbol{q}_{v_m}^i$ and $\boldsymbol{k}_{v_m}^i$ are generated from the same embedding $\boldsymbol{v}_m$, it is safe to assume:

$$\text{Cov}\left(q_{v_m}^{i\ (t)}, k_{v_m}^{i\ (t)}\right) > 0, \quad \forall t = 1, 2, \cdots, d. \tag{A.81}$$

By substituting Eq. (A.81) into Eq. (A.80), we obtain:

$$\begin{aligned}
&\mathbb{E}_{\boldsymbol{v}_m^i, \boldsymbol{v}_n^i, \boldsymbol{\delta}}\left[\mathcal{D}(\boldsymbol{v}_m^i, \boldsymbol{v}_n^i, \boldsymbol{\delta}, \boldsymbol{e}'_{v_m} - \boldsymbol{e}'_{v_n})\right] \\
&= \sum_{t=1}^{d}\left[\underbrace{\text{Cov}\left(q_{v_m}^{i\ (2t-1)}, k_{v_m}^{i\ (2t-1)}\right)}_{>0} + \underbrace{\text{Cov}\left(q_{v_m}^{i\ (2t)}, k_{v_m}^{i\ (2t)}\right)}_{>0}\right] \underbrace{\cos(\varphi'_{v_m}{}^{(t)} - \varphi'_{v_n}{}^{(t)})}_{\in[-\pi/2, \pi/2]} \\
&> \sum_{t=1}^{d} \text{Cov}\left(q_{v_m}^{i\ (t)}, k_{v_m}^{i\ (t)}\right) * 0 \geq 0.
\end{aligned} \tag{A.82}$$

This completes the proof. $\qquad\square$

**Lemma A.3** (**Expectation of quadratic form**). *Given a random vector* $\boldsymbol{x} \in \mathbb{R}^D$ *and a constant matrix* $\boldsymbol{A} \in \mathbb{R}^{D \times D}$, *where* $\mathbb{E}(\boldsymbol{x}) = \boldsymbol{\mu}$ *and* $\text{Cov}(\boldsymbol{x}) = \boldsymbol{\Sigma}$, *it always exists:*

$$\begin{aligned}
\mathbb{E}_{\boldsymbol{x}}(\boldsymbol{x}^\top \boldsymbol{A} \boldsymbol{x}) &= \mathbb{E}_{\boldsymbol{x}}\left[\text{tr}(\boldsymbol{x}^\top \boldsymbol{A} \boldsymbol{x})\right] = \mathbb{E}_{\boldsymbol{x}}\left[\text{tr}(\boldsymbol{A} \boldsymbol{x} \boldsymbol{x}^\top)\right] = \text{tr}\left[\mathbb{E}_{\boldsymbol{x}}(\boldsymbol{A} \boldsymbol{x} \boldsymbol{x}^\top)\right] = \text{tr}\left[\boldsymbol{A}(\boldsymbol{\Sigma} + \boldsymbol{\mu}\boldsymbol{\mu}^\top)\right] \\
&= \text{tr}(\boldsymbol{A}\boldsymbol{\Sigma}) + \text{tr}(\boldsymbol{A}\boldsymbol{\mu}\boldsymbol{\mu}^\top) = \text{tr}(\boldsymbol{A}\boldsymbol{\Sigma}) + \boldsymbol{\mu}^\top \boldsymbol{A}\boldsymbol{\mu}.
\end{aligned} \tag{A.83}$$

### A.6.2 Approximate Unbiasedness

**Proposition A.9.** *Given the original and noise-perturbed positional encodings defined in Prop. 4.3 , the noise-disturbed attention score approximates the original score in expectation:*

$$\mathbb{E}_{\boldsymbol{\varepsilon}_m, \boldsymbol{\varepsilon}_n}\left[\mathcal{A}\left(\boldsymbol{v}_m^i, \boldsymbol{v}_n^i, \boldsymbol{e}'_{v_m} - \boldsymbol{e}'_{v_n}\right)\right] \approx \mathcal{A}\left(\boldsymbol{v}_m^i, \boldsymbol{v}_n^i, \boldsymbol{e}_{v_m} - \boldsymbol{e}_{v_n}\right). \tag{A.84}$$

*Proof.* Let $\boldsymbol{\delta}_{mn} := \boldsymbol{\varphi}_{v_m} - \boldsymbol{\varphi}_{v_n} + \boldsymbol{\varepsilon}_m - \boldsymbol{\varepsilon}_n$, then it has:

$$\boldsymbol{\delta}_{mn} \sim \mathcal{N}\left(\boldsymbol{\varphi}_{v_m} - \boldsymbol{\varphi}_{v_n}, \boldsymbol{I}_m + \boldsymbol{I}_n\right). \tag{A.85}$$

Each component $\delta_{mn}^{(t)}$, $t = 1, 2, \cdots, d$, in $\boldsymbol{\delta}_{mn}$ satisfies:

$$\delta_{mn}^{(t)} \sim \mathcal{N}\left(\varphi_{v_m}^{(t)} - \varphi_{v_n}^{(t)}, \sigma_{mt}^2 + \sigma_{nt}^2\right). \tag{A.86}$$

Utilizing the characteristic function of Gaussian distribution, we have:

$$
\begin{aligned}
\varphi_{\delta_{mn}^{(t)}}(x) &= \mathbb{E}_{\delta_{mn}^{(t)} \sim \mathcal{N}\left(\varphi_{v_m}^{(t)} - \varphi_{v_n}^{(t)}, \sigma_{mt}^2 + \sigma_{nt}^2\right)}\left[e^{\mathrm{i}x\delta_{mn}^{(t)}}\right] \\
&= \exp\left\{-\frac{x^2(\sigma_{mt}^2 + \sigma_{nt}^2)}{2}\right\} \exp\left\{\mathrm{i}x(\varphi_{v_m}^{(t)} - \varphi_{v_n}^{(t)})\right\} \\
&= \exp\left\{-\frac{x^2(\sigma_{mt}^2 + \sigma_{nt}^2)}{2}\right\}\left[\cos\left(x(\varphi_{v_m}^{(t)} - \varphi_{v_n}^{(t)})\right) + \mathrm{i}\sin\left(x(\varphi_{v_m}^{(t)} - \varphi_{v_n}^{(t)})\right)\right] \\
&= \mathrm{Re}\left\{\mathbb{E}_{\delta_{mn}^{(t)} \sim \mathcal{N}\left(\varphi_{v_m}^{(t)} - \varphi_{v_n}^{(t)}, \sigma_{mt}^2 + \sigma_{nt}^2\right)}\left[e^{\mathrm{i}x\delta_{mn}^{(t)}}\right]\right\} + \mathrm{Im}\left\{\mathbb{E}_{\delta_{mn}^{(t)} \sim \mathcal{N}\left(\varphi_{v_m}^{(t)} - \varphi_{v_n}^{(t)}, \sigma_{mt}^2 + \sigma_{nt}^2\right)}\left[e^{\mathrm{i}x\delta_{mn}^{(t)}}\right]\right\} \\
&= \mathbb{E}_{\delta_{mn}^{(t)} \sim \mathcal{N}\left(\varphi_{v_m}^{(t)} - \varphi_{v_n}^{(t)}, \sigma_{mt}^2 + \sigma_{nt}^2\right)}\left[\cos(x\delta_{mn}^{(t)})\right] + \mathrm{i}\mathbb{E}_{\delta_{mn}^{(t)} \sim \mathcal{N}\left(\varphi_{v_m}^{(t)} - \varphi_{v_n}^{(t)}, \sigma_{mt}^2 + \sigma_{nt}^2\right)}\left[\sin(x\delta_{mn}^{(t)})\right].
\end{aligned} \tag{A.87}
$$

Then, we can obtain:

$$\mathbb{E}_{\delta_{mn}^{(t)} \sim \mathcal{N}\left(\varphi_{v_m}^{(t)} - \varphi_{v_n}^{(t)}, \sigma_{mt}^2 + \sigma_{nt}^2\right)}\left[\cos(x\delta_{mn}^{(t)})\right] = \exp\left\{-\frac{x^2(\sigma_{mt}^2 + \sigma_{nt}^2)}{2}\right\}\cos\left(x(\varphi_{v_m}^{(t)} - \varphi_{v_n}^{(t)})\right), \tag{A.88}$$

$$\mathbb{E}_{\delta_{mn}^{(t)} \sim \mathcal{N}\left(\varphi_{v_m}^{(t)} - \varphi_{v_n}^{(t)}, \sigma_{mt}^2 + \sigma_{nt}^2\right)}\left[\sin(x\delta_{mn}^{(t)})\right] = \exp\left\{-\frac{x^2(\sigma_{mt}^2 + \sigma_{nt}^2)}{2}\right\}\sin\left(x(\varphi_{v_m}^{(t)} - \varphi_{v_n}^{(t)})\right). \tag{A.89}$$

For Eq. (A.56), the attention score calculated with noise-perturbed positional encodings can be expressed as:

$$\mathcal{A}\left(\boldsymbol{v}_m^i, \boldsymbol{v}_n^i, \boldsymbol{e}'_{v_m} - \boldsymbol{e}'_{v_n}\right) = \sum_{t=1}^{d} \alpha_i^{(t)}\cos(\delta_{mn}^{(t)}) + \beta_i^{(t)}\sin(\delta_{mn}^{(t)}). \tag{A.90}$$

Substituting Eqs. (A.88) and (A.89) into Eq. (A.90), we have:

$$
\begin{aligned}
\mathbb{E}_{\boldsymbol{e}'_{v_m}, \boldsymbol{e}'_{v_n}}\left[\mathcal{A}\left(\boldsymbol{v}_m^i, \boldsymbol{v}_n^i, \boldsymbol{e}'_{v_m} - \boldsymbol{e}'_{v_n}\right)\right] &= \mathbb{E}_{\boldsymbol{\delta}_{mn}}\left[\sum_{t=1}^{d} \alpha_i^{(t)}\cos(\delta_{mn}^{(t)}) + \beta_i^{(t)}\sin(\delta_{mn}^{(t)})\right] \\
&= \sum_{t=1}^{d} \alpha_i^{(t)}\mathbb{E}_{\delta_{mn}^{(t)}}\left[\cos(x\delta_{mn}^{(t)})\Big|x=1\right] + \beta_i^{(t)}\mathbb{E}_{\delta_{mn}^{(t)}}\left[\sin(x\delta_{mn}^{(t)})\Big|x=1\right] \\
&= \sum_{t=1}^{d} \exp\left(-\frac{\sigma_{mt}^2 + \sigma_{nt}^2}{2}\right)\left[\alpha_i^{(t)}\cos(\varphi_{v_m}^{(t)} - \varphi_{v_n}^{(t)}) + \beta_i^{(t)}\sin(\varphi_{v_m}^{(t)} - \varphi_{v_n}^{(t)})\right].
\end{aligned} \tag{A.91}
$$

Since $-\pi/4 \leq \varphi_{v_m}^{(t)} \leq \pi/4$, we can assume that $\{\sigma_{mt}\}_{t=1}^{d}$ are a series of small quantities, leading to:

$$\exp\left(-\frac{\sigma_{mt}^2 + \sigma_{nt}^2}{2}\right) \approx 1. \tag{A.92}$$

By substituting Eq. (A.92) into Eq. (A.91), we reach the Prop. A.9 . To be more rigorous, we have:

$$\lim_{\boldsymbol{I}_m, \boldsymbol{I}_n \to \boldsymbol{0}} \mathbb{E}_{\boldsymbol{e}'_{v_m}, \boldsymbol{e}'_{v_n}}\left[\mathcal{A}\left(\boldsymbol{v}_m^i, \boldsymbol{v}_n^i, \boldsymbol{e}'_{v_m} - \boldsymbol{e}'_{v_n}\right)\right] = \mathcal{A}\left(\boldsymbol{v}_m^i, \boldsymbol{v}_n^i, \boldsymbol{e}_{v_m} - \boldsymbol{e}_{v_n}\right). \tag{A.93}$$

This completes the proof. $\qquad\square$

**Remark A.4.** *We perform numerical experiments to verify Eq. (A.92). The accuracy of this approximation is defined as:*

$$\mathrm{Acc} := \exp\left(-\frac{\sigma_{mt}^2 + \sigma_{nt}^2}{2}\right) \in (0, 1]. \tag{A.94}$$

*According to the three-sigma rule [53], 99.73% of the samples will be within three standard deviations of the mean, indicating that the vast majority of $\varphi_{v_m}^{(t)} + \varepsilon_m^{(t)}$ are within $[\varphi_{v_m}^{(t)} - 3\sigma_{mt}, \varphi_{v_m}^{(t)} + 3\sigma_{mt}] \subset [-\pi/4, \pi/4]$. Therefore, the accuracy is computed with $\sigma_{mt}, \sigma_{nt} \in [-\pi/12, \pi/12]$. The surface plot of* $\mathrm{Acc}$ *against $\sigma_{mt}$ and $\sigma_{nt}$ is shown in Fig. S1, demonstrating that Eq. (A.92) guarantees at least 93.8% accuracy.*

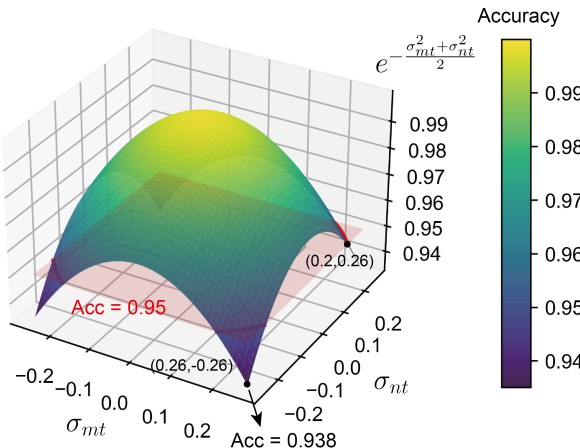

Figure S1: Surface plot of accuracy of approximation as $\sigma_{mt}, \sigma_{nt}$ change. The plane representing Acc=0.95 is marked in transparent red.

### A.6.3 Approximate Asymptotic Convergence

**Proposition A.10.** *We define the average effect of noise disturbance on attention score as:*

$$\xi_N := \frac{1}{N} \sum_{i=1}^{N} \mathcal{A}\left(\boldsymbol{v}_m^i, \boldsymbol{v}_n^i, \boldsymbol{e}'_{v_m} - \boldsymbol{e}'_{v_n}\right) - \mathcal{A}\left(\boldsymbol{v}_m^i, \boldsymbol{v}_n^i, \boldsymbol{e}_{v_m} - \boldsymbol{e}_{v_n}\right), \tag{A.95}$$

*where $N$ denotes the number of observations.*

*Given an error term $\epsilon > 0$, it approximately holds:*

$$\mathbb{P}\left(|\xi_N| \geq \epsilon\right) \leq 2\exp\left(-\frac{\epsilon^2 N}{8S}\right), \tag{A.96}$$

*where $S = \frac{1}{N} \sum_{i=1}^{N} \left(\|\boldsymbol{q}_{v_m}^i\|\|\boldsymbol{k}_{v_n}^i\|\right)^2$ is a constant independent of $\boldsymbol{e}_{v_m}, \boldsymbol{e}_{v_n}, \varepsilon_m$ and $\varepsilon_n$. Furthermore,*

$$\lim_{N \to +\infty} \mathbb{P}\left(|\xi_N| \geq \epsilon\right) = \lim_{N \to +\infty} 2\exp\left(-\frac{\epsilon^2 N}{8S}\right) = 0, \ \forall \epsilon > 0, \tag{A.97}$$

*which indicates the positional disturbance-induced bias on attention score asymptotically converges to 0 in probability, e.g., $\lim_{N \to +\infty} \xi_N \xrightarrow{P} 0$.*

*Proof.* The difference between noise-disturbed and original attention scores of the $i$-th observation is bounded as:

$$\mathcal{A}\left(\boldsymbol{v}_m^i, \boldsymbol{v}_n^i, \boldsymbol{e}_{v_m}' - \boldsymbol{e}_{v_n}'\right) - \mathcal{A}\left(\boldsymbol{v}_m^i, \boldsymbol{v}_n^i, \boldsymbol{e}_{v_m} - \boldsymbol{e}_{v_n}\right)$$

$$= \sum_{t=1}^d \alpha_i^{(t)} \cos(\delta_{mn}^{(t)}) + \beta_i^{(t)} \sin(\delta_{mn}^{(t)}) - \alpha_i^{(t)} \cos(\varphi_{v_m}^{(t)} - \varphi_{v_n}^{(t)}) - \beta_i^{(t)} \sin(\varphi_{v_m}^{(t)} - \varphi_{v_n}^{(t)})$$

$$= \sum_{t=1}^d \sqrt{\alpha_i^{(t)^2} + \beta_i^{(t)^2}} \left[\cos(\theta_i^{(t)} - \delta_{mn}^{(t)}) - \cos(\theta_i^{(t)} - \varphi_{v_m}^{(t)} + \varphi_{v_n}^{(t)})\right] \leq 2 \sum_{t=1}^d \sqrt{\alpha_i^{(t)^2} + \beta_i^{(t)^2}}$$

$$= 2 \sum_{t=1}^d \sqrt{\left[\left(q_{v_m}^i{}^{(2t-1)}\right)^2 + \left(q_{v_m}^i{}^{(2t)}\right)^2\right] \left[\left(k_{v_n}^i{}^{(2t-1)}\right)^2 + \left(k_{v_n}^i{}^{(2t)}\right)^2\right]}$$

$$\overset{\text{C.S.}}{\leq} 2 \sqrt{\left[\sum_{t=1}^d \left(q_{v_m}^i{}^{(2t-1)}\right)^2 + \left(q_{v_m}^i{}^{(2t)}\right)^2\right] \left[\sum_{t=1}^d \left(k_{v_n}^i{}^{(2t-1)}\right)^2 + \left(k_{v_n}^i{}^{(2t)}\right)^2\right]}$$

$$= 2\sqrt{\|\boldsymbol{q}_{v_m}^i\|^2 \|\boldsymbol{k}_{v_n}^i\|^2} = 2\|\boldsymbol{q}_{v_m}^i\|\|\boldsymbol{k}_{v_n}^i\|,$$

(A.98)

where C.S. denotes the Cauchy-Schwarz inequality, and $\theta_i^{(t)} \in [-\pi, \pi]$ is a directed angle defined as:

$$\cos(\theta_i^{(t)}) := \frac{\alpha_i^{(t)}}{\sqrt{\alpha_i^{(t)^2} + \beta_i^{(t)^2}}}, \qquad \sin(\theta_i^{(t)}) := \frac{\beta_i^{(t)}}{\sqrt{\alpha_i^{(t)^2} + \beta_i^{(t)^2}}}.$$

(A.99)

Applying Lemma A.4, we get:

$$\mathbb{P}\left(|\xi_N - \mathbb{E}(\xi_N)| \geq \epsilon\right) \leq 2 \exp\left(-\frac{\epsilon^2 N}{8S}\right),$$

(A.100)

where $S = \frac{1}{N} \sum_{i=1}^N \left(\|\boldsymbol{q}_{v_m}^i\|\|\boldsymbol{k}_{v_n}^i\|\right)^2$.

As shown by Eq. (A.91), the expectation of $\xi_N$ satisfies:

$$\mathbb{E}(\xi_N) = \frac{1}{N} \sum_{i=1}^N \mathbb{E}_{\boldsymbol{e}_{v_m}', \boldsymbol{e}_{v_n}'} \left[\mathcal{A}\left(\boldsymbol{v}_m^i, \boldsymbol{v}_n^i, \boldsymbol{e}_{v_m}' - \boldsymbol{e}_{v_n}'\right) - \mathcal{A}\left(\boldsymbol{v}_m^i, \boldsymbol{v}_n^i, \boldsymbol{e}_{v_m} - \boldsymbol{e}_{v_n}\right)\right]$$

$$= \frac{1}{N} \sum_{i=1}^N \sum_{t=1}^d \left[\exp\left(-\frac{\sigma_{mt}^2 + \sigma_{nt}^2}{2}\right) - 1\right] \left[\alpha_i^{(t)} \cos(\varphi_{v_m}^{(t)} - \varphi_{v_n}^{(t)}) + \beta_i^{(t)} \sin(\varphi_{v_m}^{(t)} - \varphi_{v_n}^{(t)})\right].$$

(A.101)

As analyzed in Prop. A.9 and Remark A.4, we have:

$$\exp\left(-\frac{\sigma_{mt}^2 + \sigma_{nt}^2}{2}\right) - 1 \approx 0,$$

(A.102)

which leads to:

$$\mathbb{P}\left(|\xi_N - \mathbb{E}(\xi_N)| \geq \epsilon\right) \approx \mathbb{P}\left(|\xi_N| \geq \epsilon\right) \leq 2 \exp\left(-\frac{\epsilon^2 N}{8S}\right).$$

(A.103)

To be more rigorous, we have:

$$\lim_{\boldsymbol{I}_m, \boldsymbol{I}_n \to \boldsymbol{0}} \mathbb{P}\left(|\xi_N - \mathbb{E}(\xi_N)| \geq \epsilon\right) = \mathbb{P}\left(|\xi_N| \geq \epsilon\right) \leq 2 \exp\left(-\frac{\epsilon^2 N}{8S}\right).$$

(A.104)

Since $S$ is the second moment of $\|\boldsymbol{q}_{v_m}^i\|\|\boldsymbol{k}_{v_n}^i\|$, which can be assumed to be finite empirically, when $N \to \infty$, we have $\exp\left(-\frac{\epsilon^2 N}{8S}\right) \to 0$. Thus $\mathbb{P}\left(|\xi_N| \geq \epsilon\right)$ asymptotically converges to zero. This completes the proof. $\square$

**Lemma A.4** (**Hoeffding's inequality**). *Given a series of i.i.d. random variables $\{X_i\}_{i=1}^n$, with $\mathbb{P}(X_i \in [a_i, b_i]) \approx 1$, $\forall i \in [1 \cdots n]$, the sum $S_n := \sum_{i=1}^n X_i$ satisfies:*

$$\mathbb{P}\left(|S_n - \mathbb{E}(S_n)| \geq \epsilon\right) \leq 2 \exp\left(-\frac{2\epsilon^2}{\sum_{i=1}^n (b_i - a_i)^2}\right),$$

(A.105)

*where $\epsilon > 0$ denotes the error term.*

## B    Measuring Causality-Generality of Nodes within Directed Causal Graph

We define the gregariousness ($\boldsymbol{\pi}_v$) of a node $v$ in a DAG $G(\mathcal{V}, \mathcal{E})$ as its propensity to establish outgoing connections to other nodes, which is equivalent to the degree of causal generality when $G$ represents a causal graph. While a node's out-degree can serve as a simple proxy for gregariousness, it only captures 1-hop local connectivity. For example, a node might connect to many immediate neighbors that themselves have no further outgoing connections. To capture the global connectivity, we adopt a PageRank-like approach. As shown in Eq. (18), we first compute a probability transition matrix $\boldsymbol{P}$ by transposing the absolute adjacency matrix $\boldsymbol{A}$ of $G$ normalized by its in-degrees, with $\boldsymbol{P}_{i,j}$ representing the probability that an incoming connection of $v_i$ comes from $v_j$. Note that $\boldsymbol{P}$ differs from a conventional transition matrix normalized by out-degrees, which instead represents outgoing connection probabilities. Next, a restart probability matrix $\frac{1}{M}$ is added to $\boldsymbol{P}$ to ensure that the resulting Markov chain is strongly connected and ergodic [54]. In this random walk, nodes with higher global influence (i.e., more gregarious nodes) will accumulate larger steady-state probabilities, reflecting their broader reach across the graph and higher gregariousness. This steady-state distribution, given by the left eigenvector of $\boldsymbol{P}$ corresponding to the largest eigenvalue $\lambda_{max} = 1$, defines the PageRank vector $\pi$, as shown in Eq. (18).

## C    Related Work

### C.1    Position Encoding

For sequential data, position encoding methods broadly fall into two paradigms: Absolute Positional Encoding (APE) and Relative Positional Encoding (RPE). The canonical APE approach [1] employs fixed sinusoidal functions of varying frequencies to encode each token's absolute position. Beyond this fixed design, various trainable absolute positional encoding schemes have been proposed to enhance performance [55–57]. However, these approaches often fails to generalize to sequences longer than those seen during training. To address this limitation, RPE methods [2, 5, 15–18, 58–60] modulate attention scores based on the relative distance between tokens. Among these, rotary positional encoding (RoPE) [18] applies position-dependent rotations to Query and Key vectors, using angles proportional to their absolute positions. These rotated vectors are directly involved in the computation of Query-Key attention scores. RoPE offers several key benefits, including that long-term decaying attention scores, compatibility with linear self-attention [18], and enhanced understanding of contextual knowledge [19]. Most recently, several RoPE-based methods have been proposed to improve the extrapolation ability of Transformers to longer contexts by modifying the frequency-domain representation of RoPE [60], incorporating decay-aware embeddings [61], or employing interpolation-based techniques [62]. However, these methods all assume a predefined sequential order among tokens, making them unsuitable for data without inherent ordering even when such data exhibit an implicit causal structure.

Positional encoding methods are limited and primarily developed for specialized domains. In the context of single-cell RNA sequencing (scRNA-seq), two main strategies have emerged for generating positional encodings for genes, which inherently lack a natural ordering. The first strategy [24, 25, 48] utilizes static pseudo-positional encodings derived from large scale gene co-expression data, capturing association patterns between genes. This approach is analogous to static word embeddings generated using models like CBOW, where positional encodings are assigned based on gene "proximity" in expression space. However, these encodings are not contextualized and fail to represent causal relationships between genes. The second strategy [22, 23, 63] generates contextualized pseudo-positional encodings by assigning gene orderings based on ranked expression levels within the dataset. Positional encodings, either static or trainable, are then constructed according to these induced pseudo-orders. While more adaptive, this approach still primarily reflects superficial relationships based on relative expression and cannot capture more complex dependencies, such as causal interactions among genes.

### C.2    Causal Structure Learning

Generally, there are four families of casual structure learning methods, including constraint-based methods, score-based methods, functional causal discovery, and gradient-based causal learning [41, 64]. Constraint-based methods [65–67] typically start with a fully-connected graph from which

they learn causal graph structure by leveraging the independence between graphical structures (e.g., chains, forks, and colliders). Score-based methods[68, 69], on the other hand, start with an empty graph and iteratively add or prune edges to maximize a scoring function (e.g., BIC) that measures how well the graph explains the observed distribution. A common drawback of both approaches is their computational inefficiency and limited ability to estimate causal effect strength. Functional causal discovery methods [70–72] assume explicit functional forms (e.g., spline regression) and distributional properties (e.g. non-Gaussianity) to recover both the causal structure and the strength of causal relationships. More recently, gradient-based causal discovery methods have advanced causal structure learning by formulating the task as a continuous optimization problem. For example, NOTEARS [26] enforces acyclicity through a smooth constraint embedded in a data reconstruction loss, allowing efficient gradient-based optimization without combinatorial search or independence testing. GOLEM [73] extends NOTEARS by incorporating a likelihood-based objective with sparsity regularization, while retaining acyclicity constraints. However, both methods are limited to modeling linear causal dependencies. In contrast, neural network-based approaches [27, 74, 75] introduce deep learning architectures to model complex nonlinear causal mechanisms, enabling scalable and flexible estimation of both structure and effect strengths in high-dimensional settings.

# D   Dataset Description

## D.1   Pre-Training Datasets

### D.1.1   Single-Cell Sequencing Data

We collect a wide variety of single-cell multi-omics datasets from *Homo sapiens* and *Mus musculus*, which are sourced from the CELLxGENE database [76] at `https://cellxgene.cziscience.com/`. This collection includes 1,465 datasets, encompassing around 91.5 million cells and covering approximately 900 different cell types, with data spanning several sequencing methods and omics modalities.

The datasets are primarily divided into two broad categories: single-cell transcriptomics and single-cell epigenomics, depending on the type of molecular feature being analyzed, such as RNA expression or chromatin modifications. All datasets are organized into a standardized high-dimensional matrix $X \in \mathbb{R}^{N \times n}$, where each element $x_{j,g}$ represents the gene expression values of gene $g$ in cell $j$. Here, $N$ denotes the total number of cells, and $n$ refers to the number of genes. It is noteworthy that the format of spatial transcriptomics data (e.g., Slide-seq) is processed consistently and does not take into account spatial coordinates and H&E images.

### D.1.2   DNA Methylation Data

We adopt the pretraining dataset released by MethylGPT [77], which consists of DNA methylation data collected from 154,063 human samples through the EWAS Data Hub [78] and Clockbase [79]. The dataset includes approximately 300,000 patients, with low-quality entries filtered. The cleaned data was deduplicated, ensuring no repetitions in the training set, and randomly sampled to cover 20 distinct tissue types. We specifically focus on 49,156 CpG sites selected for their biological relevance and array format compatibility, as detailed by the EWAS catalog. The data is structured into a matrix $X \in \mathbb{R}^{N \times M}$, where each element $X_{i,j}$ denotes the methylation level of CpG site $j$ in sample $i$. Here, $N$ is the number of samples and $M$ corresponds to the number of CpG sites.

## D.2   Held-Out Datasets

### D.2.1   Gene Perturbation Prediction

**Human Leukemia Cell Dataset**   The human leukemia cell dataset consists of three distinct datasets. We use the Norman dataset [47] for GPP expriment. The Norman perturbation dataset provides gene expression profiles from the K562 leukemia cell line treated with Perturb-seq. This dataset includes 131 dual-gene perturbations and 105 single-gene perturbations, with each perturbation represented by approximately 300 to 700 cells.

### D.2.2  Cell Type Annotation

**hPBMC**  The hPBMC [80] dataset, sourced from a healthy donor, contains gene expression profiles for 68,450 peripheral blood mononuclear cells (PBMCs). It includes eleven distinct cell types: CD4+ T cells, CD8+ T cells, B cells, natural killer (NK) cells, CD14+ monocytes, FCGR3A+ monocytes, dendritic cells, memory cells, helper2 cells, and megakaryocytes. These cells were processed using the 10x platform with scRNA-seq technology.

**hPancreas**  The hPancreas [81] dataset comprises 2,209 single cells from human pancreatic islets, collected from six healthy donors and four type 2 diabetes (T2D) donors. It includes both endocrine and exocrine cells, representing eight cell types: alpha, beta, gamma, delta, and epsilon endocrine cells, as well as acinar, ductal, and pancreatic stellate cells (PSCs). The cells were dissociated into single-cell suspensions, sorted via fluorescence-activated cell sorting (FACS), and subjected to RNA sequencing using the Smart-seq2 protocol.

**hBMMC**  The hBMMC dataset [82] includes 35,882 bone marrow mononuclear cells (BMMCs) from healthy donors, containing six distinct cell types: progenitor cells, B cells, T cells, NK cells, monocytes, and dendritic cells. These cells were profiled using single-cell assay for transposase-accessible chromatin sequencing (scATAC-seq) technology on the 10x platform.

**mOP**  The mOP [83] dataset provides a spatially resolved cell atlas of the mouse brain, containing molecular profiles for 338 major cell types from more than ten million cells, spanning eleven brain regions. It was generated using Multiplexed Error-Robust Fluorescence In Situ Hybridization (MER-FISH), a spatial transcriptomic technique that enables gene expression profiling while preserving the spatial organization of cells within tissue sections.

### D.2.3  Cell Clustering

**SCoPE2_Specht**  SCoPE2_Specht [45] is a representative single-cell proteomic dataset that quantifies 3,042 proteins in 1,490 cells using the SCoPE2 method. It includes two cell types: monocytes and macrophages. Notably, without polarizing cytokines, monocytes may adopt macrophage-like traits, increasing cell clustering difficulty due to their similarity.

**SCoPE2_Montalvo**  SCoPE2_Montalvo [84] quantifies 843 proteins in 508 cells using the SCoPE2 method. It contains five cell types: Vasculature, Beta 1 cells, Beta 2 cells, Delta cells, Alpha cells.

**pSCoPE_Leduc**  pSCoPE_Leduc [85] was generated by the pSCoPE technique. It quantifies 2,844 different proteins in 1,543 cells, comprising two cell types: melanoma cells and U-93 cells.

### D.2.4  Age Prediction

We use a widely used DNA methylation dataset for age prediction, collected by [86], which includes 13,505 samples (21,368 CpG sites) from multiple tissues. The dataset covers ages from 0 to 100 years, with the majority of samples derived from whole blood (47.2%) and brain tissue (34.5%).

## E  Evaluation Protocols

### E.1  Gene Perturbation Prediction

The Gene Perturbation Prediction (GPP) experiment leverages learned gene representations to predict the effects of targeted perturbations. These embeddings, generated by foundational models, serve as input to downstream classification heads that predict perturbation status, thereby elucidating gene function and regulatory network dependencies. For each foundational model incorporating distinct positional encodings, we computed the mean squared error (MSE) across the top 20 differentially expressed genes between pre- and post-perturbation expression profiles as the evaluation metric.

### E.2 Cell Type Annotation

As a standard classification task, we adopt the evaluation framework established in prior studies [22–25, 87]. Under the fine-tuning setting, we append an additional classifier to the cell embeddings generated by each model and perform supervised fine-tuning on the model parameters to optimize task-specific performance. Then, we employ accuracy and macro F1 score as the evaluation metrics.

### E.3 Cell Clustering

To assess the quality of the cell embeddings generated by our proposed method, we conducted a cell clustering experiment, which is a standard practice in single-cell proteomics [88, 89]. We employed the k-means algorithm to obtain the cell clusters and subsequently evaluated the clustering performance using three commonly used metrics, including ARI [90], NMI [91] and ASW [92].

### E.4 Age Prediction

Following established DNA methylation foundational models [77], we fine-tuned both our model and MethylGPT using a ResNet1D prediction head. During joint optimization, both the pre-trained MethylGPT and the downstream ResNet1D were trained end-to-end, with mean squared error (MSE) as the objective function. Other non-pre-trained models were also trained and evaluated on the same data splits. To robustly assess model performance, we employed the median absolute error (MedAE) as the evaluation metric.

## F Implementation Details

### F.1 Data Preprocessing

To ensure methodological consistency, we adopt a unified data processing workflow for each omics included in this study.

#### F.1.1 Single-Cell RNA Sequencing

**Gene List Mapping.** After collecting the single-cell datasets, we standardize their gene symbols to the HUGO Gene Nomenclature. Technological discrepancies between sequencing platforms occasionally result in absent gene annotations within specific datasets. To address this, unmapped genes are assigned zero expression values, thereby enforcing uniform gene symbol compatibility across all processed matrices.

**Quality Control and Normalization.** Quality control is performed using Scanpy [50] to eliminate low-information cells, defined as those with fewer than 200 detected genes. To mitigate technical variability, raw expression counts are normalized by scaling each cell's total transcript count to 10,000 (library size normalization). Subsequently, non-zero expression values undergo log1p transformation to stabilize variance and reduce skewness in the data distribution.

**Dataset Splitting.** We split the dataset in a similar way to previous studies [87]. Specifically, we collect the large-scale datasets for pre-training and several held-out datasets for evaluation. The former does not need to be split, while the latter needs to be split into a fine-tuning dataset and a test dataset in a 3:7 ratio according to different uses. Furthermore, these held-out datasets usually come from different experimental conditions, donors, and due to the common batch effects in single-cell data [93], they can be regarded as new data that differ from pre-training datasets.

#### F.1.2 Single-Cell Proteomics

The single-cell proteomic data were processed according to the SCoPE2 pipeline [45]. Raw MS files were analyzed in MaxQuant using the UniProt human proteome database [94], with TMT labeling modifications and 1% FDR filtering. Cells with <500 peptides or >20% mitochondrial proteins were excluded. Proteins detected in <10% of cells were removed. Missing values were imputed via k-nearest neighbors, and batch effects were corrected using LOESS normalization and ComBat. Analyses used R with SCoPE2 [45] and Seurat packages [95].

### F.1.3 DNA Methylomics

The DNA methylation data preprocessing followed the MethylGPT pipeline [77]. Initially, stringent quality control was applied to exclude samples with missing values exceeding 40% of CpG sites and remove duplicate entries. Subsequently, CpG sites were selected based on their biological relevance (associated with $\geq 5$ EWAS traits) and cross-platform compatibility (detected in $\geq 95\%$ of samples). The methylation $\beta$-values were standardized, with missing values intentionally preserved for downstream masked modeling tasks. The processed data were structured into a matrix $X \in \mathbb{R}^{N \times M}$, where $N$ and $M$ denote the number of samples and CpG sites, respectively, enabling systematic analysis of methylation patterns.

### F.2 Transformer-Based Backbone Models

We use the following two transformer backbones with CAPE-generated positional encodings to learn both feature and observation-level representations for tasks in Section 5.2 and Supp. G.2. For both backbones, CAPE-generated positional encodings are used in place of the original positional encodings as described below.

**scBERT** scBERT [24] discretizes continuous gene expression values via binning, mapping each to a learnable token embedding. To encode positional information, it assigns each gene a fixed embedding, which remains static during training. Finally, the expression embeddings and positional encodings are directly added and input into the transformer backbone (Performer [96]), and the expression embeddings are updated with masked reconstruction learning. Specifically, given the sample matrix $\boldsymbol{X} \in \mathbb{R}^{N \times M}$ where $N$ is the number of observations (cells), and $M$ is the number of features (genes). For each non-zero expression count $x_{ij}$ in each cell, it calculates the raw absolute values and divide them into $B$ consecutive intervals $[b_k, b_{k+1}]$, where $k = 1, 2, \cdots, B$, and each interval is assigned a feature embedding in the code book $\mathcal{C}$ with $B$ items. Then, $\mathcal{G}$ in Section 3.1, which is a function to generate contextualized, causality-agonistic intermediate feature embeddings, is defined as:

$$\boldsymbol{v}_j^i = \mathcal{G}(\boldsymbol{v}_j, \boldsymbol{x}_i) = \mathcal{C}(\text{bin}(x_{ij})), \quad \text{bin}(x_{ij}) = \begin{cases} k, & \text{if } x_{ij} > 0 \text{ and } x_{ij} \in [b_k, b_{k+1}], \\ 0, & \text{otherwise.} \end{cases} \quad (\text{F.106})$$

In the **original study**, the fusion function $\mathcal{F}$ to integrate feature embeddings and positional encodings is simply defined as:

$$\mathcal{F}(\boldsymbol{v}_j^i, \boldsymbol{\varphi}_{v_j}) = \boldsymbol{v}_j^i + \boldsymbol{\varphi}_{v_j}, \quad (\text{F.107})$$

where $\boldsymbol{\varphi}_{v_j} \in \mathbb{R}^D$ denotes the gene embedding of gene $j$ generated by pretrained gene2vec [97]. **In our study**, we instead use the CAPE-generated positional encodings $\boldsymbol{\varphi}_{v_j} \in \mathbb{R}^d$ and modify the $\mathcal{F}$ function as:

$$\mathcal{F}(\boldsymbol{v}_j^i, \boldsymbol{\varphi}_{v_j}) = \boldsymbol{R}(\boldsymbol{\varphi}_{v_j})\boldsymbol{v}_j^i, \quad (\text{F.108})$$

where $\boldsymbol{R}$ is the rotary matrix define as Eq. (26). Additionally, at the beginning of the input sequence $\boldsymbol{v}_1^i, \boldsymbol{v}_2^i, \cdots, \boldsymbol{v}_M^i$ of cell $i$, scBERT sets a special `<cls>` token, which uses the attention module to extract the cell-level embedding from $\{\boldsymbol{v}_j^i\}_{j=1}^M$.

**scGPT** scGPT [25] uses a similar architecture to scBERT, with the main differences being: (1) it uses a different positional encoding; (2) it is pre-trained on a wider range of datasets, making it suitable for multi-omics; and (3) it adopts a multi-task pre-training paradigm. In particular, in terms of positional encoding, scGPT sets a learnable gene embeddings for each gene $j$ and updates it during the training process. Therefore, scGPT maintains two different codebooks, $\mathcal{C}_{\text{bin}}$ with $B$ items and $\mathcal{C}_{\text{gene}}$ with $M$ items, one for assigning $\boldsymbol{v}_j^i$ and one for $\boldsymbol{\varphi}_{v_j}$, as:

$$\boldsymbol{v}_j^i = \mathcal{G}(\boldsymbol{v}_j, \boldsymbol{x}_i) = \mathcal{C}_{\text{bin}}(\text{bin}(x_{ij})), \quad \boldsymbol{\varphi}_{v_j} = \mathcal{C}_{\text{gene}}(j). \quad (\text{F.109})$$

Note that we use the CAPE-generated positional encodings as $\boldsymbol{\varphi}_{v_j}$ for scGPT in our study, as described in Equation (F.108).

In summary, scBERT is equivalent to using static absolute position encodings, while scGPT uses trainable absolute position encodings. When we practice CAPE on these models, we replace the positional encodings $\boldsymbol{\varphi}_{v_j}$ and fusion function $\mathcal{F}$ set by CAPE with the native ones, while keeping the other model architectures unchanged.

**General case** When measurement values of features are not on a comparable scale, the binning-based $\mathcal{G}$ functions in Eq. (F.106) and Equation (F.109) are no longer applicable for generating contextualized, causality-agonistic intermediate feature embeddings (e.g., $\boldsymbol{v}_j^i$). In such cases, the canonical transformer without positional encodings is used as the $\mathcal{G}$ function in Eq. (3) to generate these intermediate feature embeddings via a self-supervised reconstruction-based training objective.

## F.3 Benchmark Methods

### F.3.1 Positional Encoding

**Trainable Relative Position Encoding** We use the trainable relative position encoding proposed by [49] as our benchmark. Instead of relying on absolute position embeddings, this method represents the distance between query position $m$ and key position $n$ using a sinusoidal-based vector $\tilde{\boldsymbol{p}}_{m-n}$. Content vectors $\boldsymbol{x}_n$ and these relative encodings are projected separately (via $\boldsymbol{W}_k$ and $\hat{\boldsymbol{W}}_k$) and combined with two global bias vectors $u$ (content bias) and $v$ (position bias). The resulting attention score

$$\boldsymbol{q}_m^\top \boldsymbol{k}_n = \boldsymbol{x}_m^\top \boldsymbol{W}_q^\top \boldsymbol{W}_k \boldsymbol{x}_n + \boldsymbol{x}_m^\top \boldsymbol{W}_q^\top \hat{\boldsymbol{W}}_k \tilde{\boldsymbol{p}}_{m-n} + \boldsymbol{u}^\top \boldsymbol{W}_q^\top \boldsymbol{W}_k \boldsymbol{x}_n + \boldsymbol{v}^\top \boldsymbol{W}_q^\top \hat{\boldsymbol{W}}_k \tilde{\boldsymbol{p}}_{m-n} \quad \text{(F.110)}$$

ensures that attention depends only on relative distances, giving the model translation invariance and better generalization to longer sequences.

### F.3.2 Multi-Omics Analysis Benchmark Models

**KNN-ComBat** KNN-ComBat is a standard method in the existing single-cell proteomics data analysis pipeline [98], which combines KNN-based imputation with ComBat-based batch correction for routine data preprocessing.

**MAGIC** MAGIC [99] is a diffusion-based method for data cleaning in single-cell RNA sequencing, effectively imputing missing data and recovering gene interactions by sharing information across similar cells.

**AutoClass** AutoClass [100] is a deep neural network for cleaning single-cell RNA-seq data, using an autoencoder and classifier to remove noise and recover missing data, improving downstream analysis.

**Harmony** Harmony [101] effectively corrects batch effects by iteratively clustering cells and adjusting their positions in PCA space, ensuring the integration reflects biological rather than technical variation.

**Scanorama** Scanorama [102] is a tool for integrating single-cell RNA-seq data across multiple datasets while correcting for batch effects. It uses a fast, alignment-based method that projects data into a shared low-dimensional space, ensuring that the biological variation is preserved while mitigating technical variability.

**scPROTEIN** scPROTEIN [88] framework addresses peptide uncertainty, missing data, batch effects, and noise in single-cell proteomics. It uses multitask heteroscedastic regression for peptide uncertainty and graph contrastive learning for cell embedding, enhancing clustering, batch correction, and annotation.

**MethylGPT** MethylGPT [77] is a transformer-based foundation model for DNA methylation analysis, demonstrates superior performance across key tasks including age prediction, disease risk prediction and missing data imputation.

**AltumAge** AltumAge [86] is a deep learning-based epigenetic clock designed to predict human age using DNA methylation data from multiple tissues. It outperforms traditional linear models by leveraging a neural network architecture capable of capturing complex interactions between CpG sites.

**ElasticNet**   ElasticNet [103] is a linear regression model widely used in the construction of epigenetic clocks. By applying regularization to DNA methylation data, it effectively selects CpG sites related to age prediction in high-dimensional data.

**Horvath's clock**   Horvath's clock [104] is a DNA methylation-based biomarker developed by Steve Horvath to estimate the biological age of skin and blood cells.

### F.4   Training Details

**Causal Structure Learning (Step I)**   Given a preprocessed matrix $X \in \mathbb{R}^{N \times M}$, we parameterize the causal graph as a learnable matrix $A \in \mathbb{R}^{M \times M}$. Both encoder and decoder are 1–64–1 MLPs. We train $A$ via Eq. (9) with regularization coefficient $\lambda_s = 1$, where we use AdamW with a batch size of 128, a learning rate of 3e-3, and 100 epochs for optimization. After training, we apply a pruning threshold of $\tau = 0.2$ to obtain the final adjacency matrix.

**Mapping Causal Structure to Hyperbolic Space (Step II)**   Given the trained $A \in \mathbb{R}^{M \times M}$ from Step I, we map each variable into a $d$-dimensional hyperbolic space, where $d = D/2$, and the dimensionality of variable embeddings $D$ is determined by the selected transformer backbones (e.g., $D = 200$ for scBERT and $D = 512$ for scGPT). Then, $k$ hop in the graph contrastive learning Eq. (16) is set as 2, while the regularization weight $\lambda_g$ is set as 0.1, and the relative weight for the restart matrix $w$ is set as 0.15. Finally, we also choose the AdamW optimizer with a batch size of 32, a learning rate of 1e-3, and 1000 epochs for optimization.

**Transforming Hyperbolic Positional Encoding to Rotary Form (Step III)**   For scBERT, we set the dimension of feature embeddings to 200 and the backbone network adopts the performer architecture. The pre-training process is consistent with the values in the original scBERT study, that is, epochs is set to 100, batch size is 3, learning rate is 1e-4, and Adam is used for optimization. For scGPT, we set the dimension of feature embeddings to 512. The backbone network has 4 transformer blocks, each with 8 attention heads. The pre-training process is consistent with the values in the original scBERT study, that is, epochs is set to 60, batch size is 5, learning rate is 1e-4, and Adam is used for optimization.

## G   Additional Experiments

### G.1   Empirical Evaluation of CAPE's Properties

In this empirical analysis, we evaluate the effectiveness of CAPE in enhancing both the causal awareness and robustness of the self-attention mechanism. Across all experiments, the query and key vectors are fixed and generated as 128-dimensional random vectors: $q_{v_m}, k_{v_n} \in \mathbb{R}^D \sim \mathcal{N}(0, \mathbf{1}_D)$, where $D = 128$. The dimensionality of the Poincaré ball positional encodings $e_{v_m}, e_{v_n}$ is set to $d = D/2 = 64$.

#### G.1.1   Attention Attenuation Induced by Causal Distance and Causal Generality

In this analysis, pairs of $\{e_{v_m}, e_{v_n}\}$ are sampled from an isotropic Gaussian in $\mathbb{R}^d$ and subsequently normalized to lie within the unit Poincaré ball. As a result, the norm $r = \|e_{v_m}\| = \|e_{v_n}\|$ varies within the open interval $(0, 1)$, and the Poincaré distance $d_p(e_{v_m}, e_{v_n})$ spans the range $[1, 5]$. The upper bound of the attention score, $\mathcal{A}^+$, is computed for various combinations of $d_p(e_{v_m}, e_{v_n})$ and $r$, and visualized as a 3D surface in Fig. S2. On one hand, for fixed values of $r$, $\mathcal{A}^+$ monotonically decreases as the Poincaré distance (causal distance) increases, consistent with the causal distance-induced attention attenuation stated in Prop. 4.1. On the other hand, for fixed values of $d_p(e_{v_m}, e_{v_n})$, $\mathcal{A}^+$ also monotonically decreases as the causal generality $(1 - r)$ increases, aligning with the causal generality-induced attention attenuation stated in Prop. 4.2.

#### G.1.2   Robustness to Positional Disturbances

Here, we sample a single pair of $\{e_{v_m}, e_{v_n}\}$ as described in Supp. G.1.1. To simulate perturbations, we generate a varying number ($N \in [1, 100]^{\mathbb{Z}}$) of Gaussian noise pairs $\{\varepsilon_{v_m}, \varepsilon_{v_n}\} \sim \mathcal{N}(0, Diag(\sigma))$, which are added to $e_{v_m}$ and $e_{v_n}$ to obtain perturbed positional encodings $\{e'_{v_m}, e'_{v_n}\}$, from which we

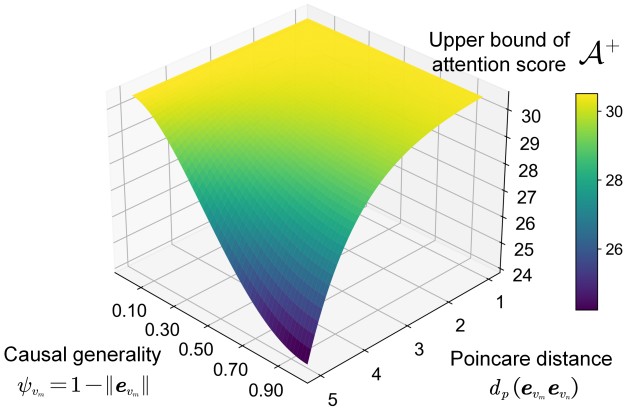

Figure S2: 3D surface showing the effect of Poincaré distance (causal distance) and causal generality on the upper bound of attention score $\mathcal{A}^+$. As Poincaré distance and causal generality increase, attention attenuation decreases.

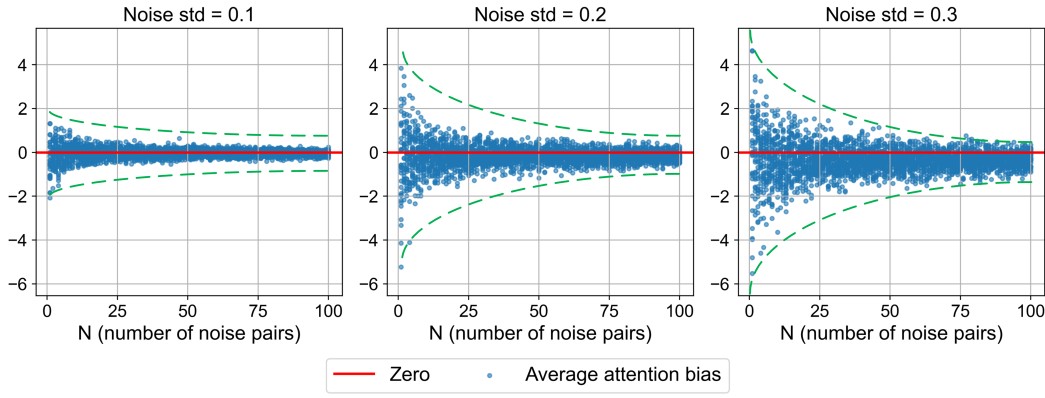

Figure S3: Robustness of CAPE-derived attention scores to positional noise. Each subplot shows the average attention bias against the number of the noise pairs $N$ under three Gaussian noise levels($\sigma = 0.1, 0.2, 0.3$). The red horizontal line marks the zero bias.

compute the average attention bias $\xi_N$ as defined in Eq. (A.95). For each value of $N$, the experiment is repeated for $T = 100$ times, and the results are visualized as scattered plots in Fig. S3. Each panel in Fig. S3 corresponds to a different noise level, with standard deviations $\sigma = 0.1, 0.2, 0.3$. We observe that as $N$ grows from 1 to 100, the distribution of $\xi_N$ becomes increasingly concentrated around zero across all noise levels. This empirical trend aligns with the theoretical result $\lim_{N \to +\infty} \xi_N \xrightarrow{P} 0$ stated in Prop. A.10, confirming the asymptotic robustness of CAPE-derived attention scores to random perturbations.

## G.2 Multi-Omics Analysis

In non-sequential data (e.g., single-cell multi-omics), learning high quality feature embeddings is critical for improving observation-level representations. For instance, in single-cell analysis, robust embeddings of genes or proteins inherently capture latent biological states (e.g., cell types or developmental trajectories), which directly enhance downstream tasks like clustering or classification. This aligns with practices in natural language processing (NLP): models such as BERT leverage a [CLS] token to aggregate sequence-level semantics for sentence classification. To validate CAPE's capability in bridging feature embeddings and observation-level semantics, we further applied CAPE to three representative observation-level tasks spanning multiple omics: (1) cell type annotation

Table S1: Performance comparison of cell type annotation on scRNA-seq datasets. Acc and MF1 denote accuracy and macro F1-score (%), respectively.

| Methods | Pos Encoding | hPBMC | | hPancreas | | hBMMC | | mOP | |
|---|---|---|---|---|---|---|---|---|---|
| | | Acc↑ | MF1↑ | Acc↑ | MF1↑ | Acc↑ | MF1↑ | Acc↑ | MF1↑ |
| scBERT | Static absolute[†] | 75.74 | 67.34 | 69.21 | 67.03 | 67.09 | 59.25 | 74.37 | 70.22 |
| | Trainable relative | 77.51 | 70.66 | 73.48 | 71.14 | 69.79 | 66.90 | 77.64 | 71.79 |
| | CAPE | 80.71 | 72.32 | 78.07 | 74.31 | 74.49 | **71.55** | 85.27 | 80.41 |
| scGPT | Trainable absolute[†] | 84.48 | 75.39 | 70.76 | 68.03 | 67.18 | 60.93 | 80.14 | 77.03 |
| | Trainable relative | 84.47 | 77.01 | 74.87 | 72.42 | 75.65 | 73.91 | 85.37 | 79.40 |
| | CAPE | **85.14** | **77.09** | **82.27** | **75.10** | **78.14** | 70.76 | **87.62** | **82.07** |

in scRNA-seq data, (2) cell clustering in single cell proteomics, and (3) age prediction in DNA methylomics.

We begin with cell type annotation, the most common task in single-cell foundational model to evaluate the cell embeddings generated by models. Following a similar experimental setup as described in Section 5.2, cell embeddings are learned using scGPT and scBERT with three types of positional encoding across three human datasets (hPBMC, hPancreas, and hBMMC) and one mouse dataset (mOP) (See Supp. D.2 for details). We find that both models, scGPT and scBERT, when combined with CAPE, achieve the best performance.

For single cell proteomics, we evaluate the cell embeddings in the cell clustering task, applying it to two datasets: SCoPE2_Specht and SCoPE2_Montalvo (see Supp. D.2 for details). Given the absence of transformer-based models designed for single cell proteomics, we leveraged scGPT to generate cell embeddings. Although scGPT is originally designed for scRNA-seq data, we fine-tuned it on the pSCoPE_Leduc dataset to adapt it for proteomics data. After fine-tuning, we used scGPT to obtain the cell embeddings for clustering, respectively using its default position encoding and CAPE for comparison.

Due to the lack of established foundational models and the scarcity of single cell proteomic computational methods, current work in single cell proteomics often uses methods originally developed for scRNA-seq as baseline. To ensure comprehensive benchmarking, we evaluated our method against: (1) scPROTEIN, a state-of-the-art representation learning framework specifically designed for single-cell proteomics [88]; (2) the common proteomics analysis pipeline (KNN-ComBat) combining KNN-based imputation with ComBat batch correction [98]; and (3) established scRNA-seq computational methods adapted for proteomic data, including [99–102] (See Supp. F.3 for details). As shown in Tab. S2, scGPT with CAPE significantly outperforms both the original scGPT and other baseline methods across all evaluated metrics in clustering. This demonstrates that CAPE-derived cell embeddings preserve substantially richer biological information.

We further assess CAPE's performance in predicting age from DNA methylation patterns. Similar to the experimental setup in GPP (Section 5.2), we utilize a transformer-based DNA methylation foundational model, MethylGPT [77], to generate cell embeddings, using its default position encoding strategy along with CAPE. Similar to the position encoding used in single-cell foundational models like scGPT [25] and scBERT [24], MethylGPT assigns embeddings to each CpG site, similar to how genes are represented in scGPT. Additionally, methylation values, much like gene expression counts in scRNA-seq, are also assigned embeddings, which are then summed and subsequently used as input to the transformer blocks. For a comprehensive assessment, we benchmark CAPE's age prediction performance against three widely use methods including AltumAge [86], ElasticNet [103] and Horvath's clock [104].

After fine-tuning for age prediction, MethylGPT, enhanced with CAPE, achieved superior accuracy and exhibited the lowest median absolute error among all methods(Tab. S3). This demonstrates CAPE's capacity to capture the hidden causal structure between CpG sites, effectively learning biologically meaningful age-related patterns.

### G.3 Sensitivity Analysis

In this section, we conduct sensitivity analysis for three key hyperparameters of CAPE's training objective, including the DAG pruning threshold $\tau$ (see Section 3.3), the $k$-hop neighborhood size for graph contrastive learning (Eq. (16)), and the weight $\lambda_g$ of the causal generality penalty term

Table S2: Performance comparison of cell clustering on single-cell proteomics.

| Methods | SCoPE2_Specht | | | SCoPE2_Montalvo | | |
|---|---|---|---|---|---|---|
| | ARI | NMI | ASW | ARI | NMI | ASW |
| KNN-ComBat | 0.317 | 0.066 | 0.375 | 0.274 | 0.053 | 0.536 |
| MAGIC | 0.245 | 0.375 | 0.339 | 0.452 | 0.389 | 0.693 |
| AutoClass | 0.02 | 0.313 | 0.211 | 0.316 | 0.253 | 0.421 |
| Harmony | 0.406 | 0.230 | 0.422 | 0.443 | 0.132 | 0.682 |
| Scanorama | 0.013 | 0.215 | 0.003 | 0.001 | 0.138 | 0.006 |
| scPROTEIN | 0.435 | 0.428 | 0.469 | 0.502 | 0.465 | **0.689** |
| scGPT | 0.396 | 0.405 | 0.156 | 0.482 | 0.377 | 0.383 |
| scGPT w/ CAPE | **0.513** | **0.497** | **0.475** | **0.516** | **0.572** | 0.631 |

Table S3: Performance comparison of age prediction on DNA methylomics datasets. MedAE denotes Median Absolute Error.

| Methods | MethylGPT | AltumAge | ElasticNet | Horvath's clock | MethylGPT w/ CAPE |
|---|---|---|---|---|---|
| MedAE | 4.59 | 6.53 | 5.16 | 6.88 | **4.07** |

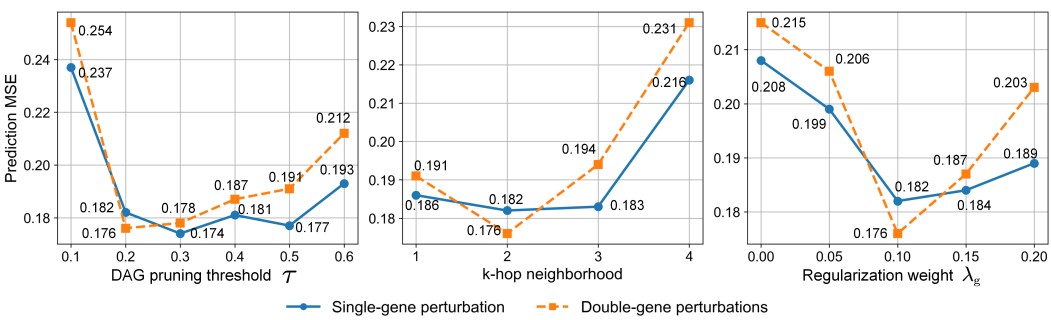

Figure S4: Sensitivity analysis about DAG pruning threshold, k-hop neighborhood, and regularization weight.

$\Omega$ in Eq. (16). Fig. S4 illustrates the Prediction Mean Squared Error (MSE) on single-gene and double-gene perturbation tasks as these hyperparameters are varied.

In the leftmost panel, we observe that increasing $\tau$ from 0.1 to 0.2 significantly improves prediction accuracy. However, as $\tau$ continues to increase beyond 0.2, performance gradually declines. This pattern reflects a trade-off: a low threshold ($\tau = 0.1$) fails to sufficiently eliminate noisy, false-positive causal edges, whereas a high threshold ($\tau > 0.3$) may excessively prune true causal edges, thus degrading the quality of the learned causal structure.

The middle panel shows that performance peaks at $k = 2$. A small $k$ (e.g., $k = 1$) may incorrectly designate features with a strong 2-hop causal relationship as negative causal pairs. Conversely, a large $k$ (e.g., $k \geq 4$) risks misclassifying weakly or non-causally related features as positive causal pairs. Both extremes undermine the effectiveness of the graph contrastive loss in preserving salient causal relationships, leading to suboptimal positional encodings.

The rightmost panel displays a V-shaped trend with respect to $\lambda_g$, with optimal accuracy achieved at $\lambda_g = 0.1$. A diminutive $\lambda_g$ may not sufficiently regularize causally general features towards the origin, thereby weakening the encoding of causal specificity. Conversely, an excessively large $\lambda_g$ could force all features towards the origin, collapsing causal distances and diminishing the model's capacity to discern varying causal strengths. Empirically, setting $\lambda_g$ in the range of 0.1 to 0.15 appear to offer a favorable balance, preserving both causal specificity and relative causal distances in the positional encodings.

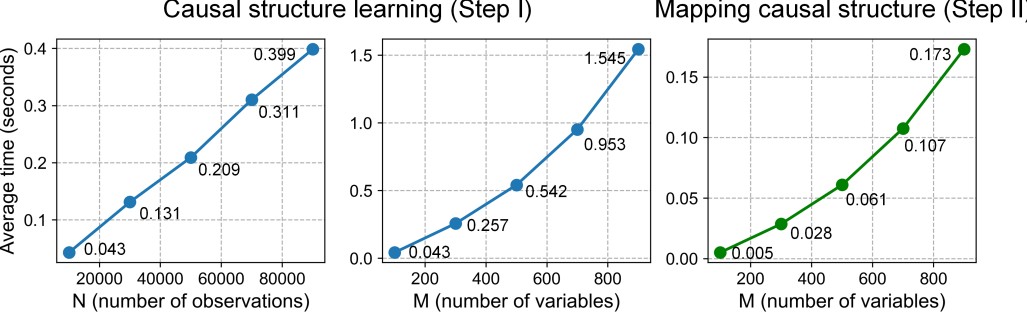

Figure S5: Runtime per epoch during training Step I and Step II on sampled sub-datasets.

### G.4 Complexity Analysis

We begin by analyzing the computational complexity of CAPE. In Step I, given a data matrix $X \in \mathbb{R}^{N \times M}$ with $N$ observations and $M$ features, CAPE implements the encoder and decoder functions (Eqs. (6) and (7)) using MLPs, and solves the nonlinear SEM through a trainable adjacency matrix $A \in \mathbb{R}^{M \times M}$. This step is analogous to training a graph neural network [27], with a time complexity of $\mathcal{O}(NM^2)$ [105]. To enforce acyclicity, CAPE introduces a smooth constraint term $h(A)$, which involves computing a matrix exponential and incurs a time complexity of $\mathcal{O}(M^3)$ [106]. To mitigate this computational bottleneck, we adopt the low-rank approximation strategy proposed by Dong, et al. [107] in our implementation. Specifically, $A$ is approximated as $UV^\top$ with $U, V \in \mathbb{R}^{M \times r}$ and rank $r = 40$. This approximation reduces the computation complexity of acyclicity constraint to $\mathcal{O}(M^2 r)$, yielding an overall complexity of $\mathcal{O}((N + r)M^2)$ for Step I. Thereby, CAPE achieves significantly improved scalability while maintaining its expressiveness. In Step II, CAPE maps each feature in the DAG to a $d$-dimensional hyperbolic embedding. The dominant cost in this step arises from graph constrastive learning, which has a time complexity of $\mathcal{O}(dM^2)$ [108].

We empirically assess CAPE's scalability with respect to the number of samples ($N$) and features ($M$). To evaluate the impact of $N$, we subsample $N = 1, 3, 5, 7, 9 \times 10^4$ instances from the GPP dataset (Tab. 2) with $M = 100$ fixed. To assess the effect of $M$, we vary $M = 100, 300, 500, 700, 900$ while fixing $N = 10^4$. For each configuration, we learn a separate adjacency matrix $A \in \mathbb{R}^{M \times M}$ in Step I and reuse it in Step II. We repeat each experiment 10 times and report the average runtime in Fig. S5. As expected, runtime scales linearly with $N$, and approximately quadratically with $M$, reflecting the theoretical complexities of $\mathcal{O}(NM^2)$ and $\mathcal{O}(dM^2)$ in Steps I and II, respectively.

## H   Limitations

While CAPE offers a general and theoretically grounded solution for encoding causal structure in non-sequential data, its effectiveness currently relies on the quality of the inferred causal graph. Although we adopt a robust variational formulation for causal discovery, inaccuracies may arise in extremely noisy or undersampled settings. Additionally, our current implementation assumes feature-wise causal structure to be static across samples, which may not fully capture sample-specific heterogeneity in highly dynamic systems. These limitations point to promising directions for future work, such as incorporating uncertainty-aware causal discovery or adapting CAPE to sample-dependent causal structures.

## I   Broader Impacts

By enabling transformers to model non-sequential yet causally related features, CAPE has the potential to advance representation learning in a wide range of scientific domains where causal structure is key—such as biomedicine, economics, and environmental science. In particular, our method may assist researchers in uncovering interpretable, causally grounded representations from high-dimensional

biological data, potentially informing therapeutic target discovery or precision medicine. As with any causal inference technique, misuse or overinterpretation of inferred relationships remains a risk, especially in domains where observational biases are strong. We encourage responsible use of CAPE in conjunction with domain expertise, and highlight the importance of open datasets, reproducible code, and transparent evaluation to mitigate unintended consequences.

