# OpenReview forum: "Causality-Induced Positional Encoding for Transformer-Based Representation Learning of Non-Sequential Features"
_NeurIPS.cc/2025/Conference — NeurIPS 2025 poster_

### Official Review · Reviewer_Jacd · 2025-06-11

**Clarity:** 3
**Significance:** 3
**Originality:** 4
**Rating:** 5
**Confidence:** 4

**Summary:**

This paper proposes a novel method to embed the structure of a causal graph into a Transformer model. It firstly encodes causal graphs into hyperbolic space, translating causal strength and hierarchy into geometric distance and position, then converts them into a rotary form. Experimental results verify the effectiveness of the proposed method.

**Questions:**

1. To my understanding, causal structure learning usually requires some fundamental assumptions, e.g., causal faithfulness and causal sufficiency. It is unknown whether these assumptions remain true in the context of real-world datasets. Could the proposed approach remain effective when these assumptions are not satisfied, e.g., when latent confounders exist?
2. A generalized nonlinear SEM model is used to infer the causal DAG. However, in many real-world scenarios, the DAG is non-identifiable from the data, so I think the inferred causal DAG should contain unavoidable errors under these scenarios. If this is correct, how does the proposed approach solve this problem? Moreover, if the proposed method can be effective under inaccurate causal DAG, could it be used in traditional sequential features?
3. RoPE is not used as benchmark as they rely on predefined feature order. Is it possible to use RoPE or other order-based methods with the topological order of the causal DAG? I think the comparison could make the experimental results more comprehensive.
4. As also mentioned in Sec. C.2, there are many casual structure learning methods. Why nonlinear SEM method is used in the causal structure learning step? Are there any other alternatives?

**Ethical Concerns:**

["NO or VERY MINOR ethics concerns only"]

**Final Justification:**

Thanks for the authors' detailed response. I have carefully read the authors' rebuttal as well as the comments from other reviewers. I believe the authors' response has clarified most of my questions. Overall, this is an excellent piece of work, and I have decided to raise the final score to 5 points (accept).

**Limitations:**

Yes

**Quality:**

4

**Strengths And Weaknesses:**

## Strengths:
- The motivation of applying causality-induced positional encoding is clear and significant.
- The method is clearly described and the paper is well-written.
## Weakness:
- This paper could benefit from more discussions on the correctness of the causal structure learning step. See questions for details.

---

> ### Author Rebuttal · Authors · 2025-07-31
>
> We sincerely thank you for the thoughtful and constructive feedback.
> ***
> **Question 1**
>
> Thank you for raising this fundamental and important point. We agree that our causal discovery step, like most SEM-based methods, assumes causal sufficiency and faithfulness. Therefore, in the presence of strong latent confounders or faithfulness violations, the learned DAG may not be the "true" underlying causal graph (please see our response to the next question for some tentative solutions). Since the primary contribution of our work is to introduce a method for creating causality-aware positional encodings from a learned causal structure, rather than proposing a novel, assumption-free causal discovery algorithm, we adopt the current methodology,  which extends NOTEARS that is known for easy optimization and balanced computational efficiency and performance. Our Ablation Studies (Supplementary G.3) provide strong empirical support for this approach. The results show that using this learned DAG, even if it is an approximation of the true causal structure, provides a far more effective basis for representation learning than assuming no structure or using a simple similarity graph. We agree that explicitly handling violations of these assumptions is a critical direction for future research. The modular design of CAPE is well-suited for this, as it would allow for the future integration of more advanced causal discovery algorithms that are robust to such violations. We will add this to our discussion.
> ***
> **Question 2**
>
> This is an important consideration and we agree that in case of unidentifiable causal effects, CAPE-inferred DAG will inevitably contain errors. Actually, how to accurately infer causal DAG while maintaining the computational efficiency remains a significant challenge currently, particularly when using real-word datasets. Here, we tentatively propose several possible solutions:
>
> i) **Improve CAPE's causal structure learning**. The nonlinear SEM can be modified to incorporate hidden confounding variables $h_i$ for each variable $x_i$ in addition to its parents $PA(x_i)$, leading to $x_i=f_i(PA(x_i),h_i)+z_i$. During training, the VAE encoder learns to approximate the posterior distribution of all the latent confounders ($q_\theta(H|X)$) given the observed data. It takes the observed data $x_i$ as input and outputs the parameters (e.g., mean and variance) for the distributions of the latent variables $h_i$. The decoder reconstructs the original data $X$ with $\tilde{H}$ sampled from the variational distribution. If the model can achieve a better objective score (a higher ELBO) by using a latent variable to explain the correlation, it will infer the presence of a confounding arc. Meanwhile, the SEM can be improved by assuming non-Gaussian noises with asymmetrical distribution that will help to correctly identify the direction of node links, given our current method might just return a sample DAG from a Markov equivalence class.
>
>  ii) **Improve the ex post CAPE-derived DAG**. We can apply conditional independence tests (e.g., Fisher's Z-test) against the chain, fork, and collider structures within *ex post* CAPE-inferred DAG, thus reducing the incorrect node links. Additionally, suspicious graph structures in the DAG can be validated using various refutation tests.
>
> iii) **Robustify the mapping of DAG to hyperbolic space**. Confidence scores can be assigned to node links within the CAPE-derived DAG based on the possibility of their association with hidden confoundings and the statistical significance of their independence and refutation tests. During the subsequent mapping of DAG to hyperbolic embeddings, variable pairs are assigned weights based on their confidence scores so that accurate causal structures in the DAG dominate over those less accurate. However, these solutions can come with substantial computational overheads, particularly when dealing with large DAG. Thus, they should be adopted only when we are certain about the presence of impactful hidden confoundings, and when the accuracy of inferred DAG and causal positional encodings are critical. Due to the limited rebuttal time, we will implement and test these solutions in the future and add the results to our manuscript.
>
> Your suggestion on applying CAPE to traditional sequential features is very interesting. Yet, we believe it may not be the most practical application of CAPE in its current form. This is because for traditional sequential features, their predefined order already provides a strong and computationally efficient causal prior—that an element is primarily caused by the elements that precede it. This is the principle behind successful causal autoregressive models like GPT. While applying CAPE may capture more complex, non-local causal relationships, the significant computational overhead of the causal discovery step would likely outweigh the marginal performance benefits. We believe the main strength of CAPE lies in domains where no such natural order exists. In our future work, we will try to migrate CAPE to sequential features, improving its computational efficiency and effectiveness by utilizing sequential order information.
> ***
> **Question 3**
>
> This is an excellent suggestion. We can use algorithms such as the Kahn's algorithm to perform a topological sorting on the inferred DAG to derive feature nodes' topological order, which can be used by RoPE. However, this method has several limitations. First, the topological order also predicates on the correctness of the inferred DAG, thus can be erroneous. Second, a topological sort flattens the DAG into a single, discrete sequence. This process cannot capture causal strength; two causally related nodes might be placed far apart in the sequence if there are many other nodes with stronger causal relation with them. In contrast, CAPE's continuous geometric embedding explicitly models these strengths via hyperbolic distances. Third, topological sorting algorithms can have major limitations. For example, in Kahn's algorithm, if multiple nodes have no incoming edges, their relative order is arbitrary, leading to ambiguous topological order. As per your suggestion, we ran a new ablation experiment using a topological sort of the inferred DAG to generate positional encodings. The results are shown below:
>
> | Models | Single-gene perturbation | Double-gene perturbations |
> | -- | -- | -- |
> | Original CAPE | 0.182 ($\pm$0.005) | 0.176 ($\pm$0.008) |
> | Topological order | 0.201 ($\pm$0.012) | 0.210 ($\pm$0.015) |
> | Similarity | 0.209 ($\pm$0.010) | 0.213 ($\pm$0.011) |
> | Non-positional | 0.234 ($\pm$0.014) | 0.238 ($\pm$0.017) |
>
> As expected, although topological order performs better than non-positional encodings and similarity graph, CAPE significantly outperforms it.
> ***
> **Question 4**
>
> We thank the reviewer for this important question. The choice of a nonlinear SEM was a deliberate decision based on the specific goals of our work and the practical trade-offs among different classes of causal discovery algorithms. While alternatives exist that can relax certain assumptions, they often come with significant limitations. For example, many constraint-based methods are more robust to hidden confounders but are typically computationally intractable and rely on discrete statistical tests, making them difficult to integrate into a gradient-based deep learning framework. Other approaches like functional causal models may be limited by strong assumptions about the data-generating process, such as linearity. Our primary goal is not to solve the general problem of causal discovery but to learn a plausible DAG to serve as an effective inductive bias for the Transformer. For this purpose, our approach was chosen for three key reasons: First, it is based on gradient-based continuous optimization, which not only allows for easy implementation but also natural integration with neural network to model complex nonlinear causal relationships. Second, our method is an end-to-end causal framework that simultaneously retrieves causal structure and causal strength. Finally, our method extends NOTEARS, a well-received causal discovery method known for well-balanced computational efficiency and performance. We will add these discussions to the Related Work Section.

---

### Official Review · Reviewer_wHJ8 · 2025-06-28

**Clarity:** 3
**Significance:** 4
**Originality:** 4
**Rating:** 5
**Confidence:** 4

**Summary:**

This paper introduces CAPE, a positional encoding method that learns causal structures among non-sequential features to provide Transformer with positional information related to causal structure rather than sequential structure. CAPE first uses generalized structural equation model (SEM) to identify causal directed acyclic graphs (DAGs) among features. It then embeds the identified DAGs into hyperbolic space and uses the hyperbolic surface model to preserve two key properties: causal strength and causal specificity. Finally, the hyperbolic embedding is converted into a rotated positional encoding form, which is compatible with the self-attention mechanism of Transformer. The authors provide theoretical analysis demonstrating that CAPE possesses desirable properties including causal distance-induced attenuation, causal generality-induced attenuation, and robustness to positional disturbances, and validate the method's effectiveness on synthetic data and multi-omics datasets.

**Questions:**

1. Computational efficiency - How does the computational efficiency compare? Can you provide complexity analysis?
2. Generalization - Is the method still robust when dealing with larger causal graphs or noisy conditions? Does this approach still work in a causal diagram that only shows a partial set of causal relationships? Is it possible to modeling different causal strengths?

**Ethical Concerns:**

["NO or VERY MINOR ethics concerns only"]

**Final Justification:**

I have carefully reviewed the authors' rebuttal and the comments provided by other reviewers. The authors have addressed the majority of my concerns effectively. In particular, I appreciate the clarification regarding the algorithm's complexity and its behavior on sparse graphs. Overall, I believe this is a promising piece of work, and I have decided to maintain my final score of 5.

**Limitations:**

The effectiveness in other domains remains to be confirmed.

**Quality:**

4

**Strengths And Weaknesses:**

Strengths:
1. Well-motivated - Addresses the important practical problem that traditional positional encoding cannot handle non-sequential causally-related features, with significant application value
2. Novel approach - Cleverly combines causal discovery, hyperbolic geometry, and rotary attention mechanisms
3. Solid theoretical foundation and well-designed experiments - Provides rigorous theoretical analysis with comprehensive experimental validation

Weaknesses:
1. Limited baseline comparisons - Lacks comparison with more diverse baseline methods, with experiments confined primarily to biomedical domains
2. Potential computational overhead - May increase computational costs, uncertain practical utility

---

> ### Author Rebuttal · Authors · 2025-07-31
>
> We greatly appreciate your careful evaluation and constructive comments. To better respond to your concerns, we have summarized and responded to the following points.
> ***
> **Limited baseline comparisons and effectiveness in other domains**
>
> Thank you for this constructive comment about applying CAPE to other domains. We focus on the biomedical domain as the rapid development in omics technologies in recent years allows the accumulation of enormous data that CAPE can be applied to. So we consider the biomedical domain the main battleground for CAPE. However, we agree that the application of CAPE to other domains is also important. Due to the limited rebuttal time, we conducted a simple experiment using the MovieLens-32M dataset (with 87,585 movies and 200,948 users), which is commonly used in test recommendation algorithms. This data has a table format with rows representing users, columns movies, and entries ratings. Our goal is to learn causality-aware embeddings of movies for their multi-label genre classification (20 genres). The benchmark methods include Transformer w/o positional encodings (Transformer-w/o) and an MLP-based classifier that uses the raw data. For CAPE-Transformer and Transformer-w/o, linear probing is adopted to train a projection head for multi-label classification using the derived movie embeddings of the training set. The benchmark MLP classifier is trained in a supervised manner using columns of the raw data as inputs. All movies are partitioned into training (80\%), validation (10\%), and test (10\%) sets.
>
> | Inputs | Classifier | Jaccard Index | Macro-F1 |
> | -- | -- | -- | -- |
> | Raw Column Data | MLP | 0.457 | 0.583 |
> | Transformer-w/o-generated movie embeddings | Projection head | 0.633 | 0.659 |
> | **CAPE-Transformer**-generated movie embeddings | Projection head | **0.678** | **0.690** |
>
> ***
> **Computational efficiency**
>
> We provide a detailed complexity analysis in Supplementary Section G.5. The computational cost of CAPE is primarily concentrated in two preprocessing steps: Step I (Causal Discovery) and Step II (Hyperbolic Embedding). Specifically, in Step I, the main computational bottleneck arises from enforcing the acyclicity constraint during causal structure learning. To mitigate this, we adopt a low-rank approximation of the adjacency matrix, reducing the constraint’s complexity to $\mathcal{O}(M^2 r)$, where $r \ll M$ is the rank parameter. As detailed in our analysis, this results in an overall complexity of $\mathcal{O}((N + r) M^2)$ for this step, where $N$ is the number of samples. Step II, which performs hyperbolic embedding of the discovered graph, has a complexity of $\mathcal{O}(d M^2)$, where $d$ is the embedding dimension. The causal encodings generated in these steps are then fed into a standard transformer architecture, where attention is computed across $M$ feature embeddings for each of the $N$ samples, with a complexity of $\mathcal{O}(N M^2)$. Thus, although CAPE introduces a modest additional cost during the construction of positional encodings, it preserves the overall complexity of standard transformers. The total computational cost remains $\mathcal{O}(N M^2)$, with only a constant-factor increase, rather than a higher-order growth such as $\mathcal{O}(M^3)$. Finally, our \textbf{empirical results} support the theoretical complexity analysis, showing that the runtime of CAPE scales linearly with $N$ and quadratically with $M$.
> ***
> **Robustness and Generalization.**
>
> This is a great and constructive question.
>
> **i) Graph size**. We agree that accuracy can decrease as the graph size increases due to the exponentially increased search space and the elevated risk of converging to local optima. However, in practice, the local solutions can often be high-quality as demonstrated by other gradient-based causal discovery methods like NOTEARS. Furthermore, performance is often better on sparse graphs due to constrained search space, which are common in real-world systems like gene regulatory networks. CAPE leverages regularization and thresholding on the adjacency matrix to promote sparsity.
>
> **ii) Noisy condition**. CAPE is designed to be robust to additive Gaussian noise, as this is an explicit component of the SEM framework. However, we acknowledge two limitations: First, performance can degrade as the noise variance increases, reducing the signal-to-noise ratio in the data. Second, performance may decline if the true noise follows a non-Gaussian distribution, as this constitutes a model misspecification.
>
> **iii) Effectiveness with partial causal graphs.** CAPE's sparsity promoting of inferred DAG can result in partial causal graphs, which may affect accuracy of Poincaré positional encodings during the hyperbolic mapping.
>
> To empirically validate these points, we conducted three experiments: for **i)**, we simulated four pairs of DAGs with the number of nodes increasing from 10 to 100 using the Erdos-Renyi algorithm. DAGs in the same pair differ in their sparsity. For **ii)**, we increased the standard deviation of Gaussian noise of simulated data from 1 to 3. We aslo simulated data with non-Gaussian noises (Gumble noise). For **iii)**, we randomly drop edges from the simulated DAG with a increasing rate from 10\% to 40\% edges. The first two experiments are evaluated using structural Hamming distance (SHD) and false discovery rate (FDR), while the third one is evaluated by checking the average changes in Euclidean distances between the two nodes with removed edges. The results are shown below.
>
> | Number of nodes | Low-sparsity SHD | Low-sparsity FDR | High-sparsity SHD | High-sparsity FDR |
> | -- | -- | -- | -- | -- |
> | 10 | 5.8 ($\pm$1.3) | 0.22 ($\pm$0.10) | 2.9 ($\pm$1.1) | 0.09 ($\pm$0.08) |
> | 30 | 17.9 ($\pm$5.2) | 0.24 ($\pm$0.05) | 14.5 ($\pm$3.2) | 0.15 ($\pm$0.10) |
> | 50 | 29.0 ($\pm$6.5) | 0.21 ($\pm$0.05) | 22.5 ($\pm$4.2) | 0.17 ($\pm$0.08) |
> | 100 | 60.0 ($\pm$10.0) | 0.26 ($\pm$0.04) | 45.0 ($\pm$8.0) | 0.11 ($\pm$0.06) |
>
> | Noise pattern | SHD | FDR |
> | -- | -- | -- |
> | Gaussian(0,1) | 5.8 ($\pm$1.3) | 0.22 ($\pm$0.10) |
> | Gaussian(0,2) | 5.6 ($\pm$1.7) | 0.25 ($\pm$0.12) |
> | Gaussian(0,3) | 6.0 ($\pm$2.0) | 0.27 ($\pm$0.14) |
> | Gumbel(0,1) | 6.3 ($\pm$1.5) | 0.25 ($\pm$0.10) |
>
> | Removal rate | Average changes |
> | -- | -- |
> | 10% | 0.1589 ($\pm$0.0269) |
> | 20% | 0.1858 ($\pm$0.0281) |
> | 40% | 0.3530 ($\pm$0.0362) |
> PS: The Euclidean distance on the Poincaré ball ranges from 0 to 2.
>
> Generally, the results align with our expectations explained above. Interestingly, in the third experiment, the difference in average Poincaré distances is relatively smaller when the edge dropout rate is below 20\% and rapidly increases when the rate is 40\%. This is probably due to that when the dropout rate is low, the causal relationships between two nodes of a removed edge can be maintained via other paths until those paths are eliminated by the increasing dropout rate. We also emphasize that, since CAPE's primary goal is proposing a methodology of creating causality-aware positional encodings from a learned causal structure, rather than proposing a novel causal discovery algorithm, we adopt a modular design for CAPE's causal discovery component, thus allowing for the future integration of more robust causal discovery algorithms.
>
> Finally, CAPE accounts for causal strength, which are explicitly modeled by its SEM and translated into Poincaré distances during the hyperbolic mapping.

---

### Official Review · Reviewer_TuE7 · 2025-06-29

**Clarity:** 4
**Significance:** 3
**Originality:** 4
**Rating:** 5
**Confidence:** 4

**Summary:**

The paper introduces an end-to-end deep neural network for 1) causal discovery/causal strength estimation 2) causal graph encoding 3) self attention of non-sequential feature with causal positional encoding. It provides theoretical properties of the attention with respect to the positional encoding, verified by ablation experiments, and shows experimentally how the introduced encoding improves a transformer network compared to other encoding methods.

**Questions:**

Other than making sure that the theoretical properties are correct, and that the plain text explanation properly reflects the strength of the guarantee (which would be my main concerns), I have one other question.

The greatest limitation of the encoding method used is that rotation matrices do not guarantee that distance on the pointcarre space result into null attention. This is due to the necessity of gamma being function of the difference in positional encoding, which results in the radius function being independent of the positional encoding. **What if gamma was function of the encoding difference AND causal generality?** While this differs from the usual setting, this could be fundamental in bridging Pointcarré space with positional encoding. Note that I did not carefully check if this suggestion is viable.

**Ethical Concerns:**

["NO or VERY MINOR ethics concerns only"]

**Final Justification:**

Authors responded and amended their statements as asked. Other reviews are positive, I believe that the paper should be accepted.

**Limitations:**

On top of the limitations identified in appendix (which should be put in the main paper), the authors should mention that the rotation-based encoding does not entirely reflect the information encoded in pointcarré space. It might be possible to obtain better guarantees with a different positional encoding transformation.

**Quality:**

3

**Strengths And Weaknesses:**

**Strong points**
 - Clarity: the paper is precise, well detailled, organized, and the extensive proof rely on unified and clear notations.
 - Quality: The theoretical results underline desired properties that are validated in ablation experiments, despite some theoretical guarantees remaining somewhat weak. Real data experiments were reported with standard deviation that seems to indicate that the improvements are statistically significant.
 - Originality: While the encoding of graph in Pointcarré space is not novel, its use to create positional encoding from *causal* information is, to my knowledge, novel.
 - Significance: Combining causal analysis and the more popular architectures of Deep Neural Network is an unsolved yet high impact problem.

**Weak points**

**Clarity**
 - Lines 212+: pointcarre encodings denote a point on a disk, so why is it that phi = c.e is an angle vector? The multiplication does not change the angle, just the magnitude of e, since e was defined using cartesian coordinates. Therefore, phi does not correspond to a coordinate system that involves angles.
 - Typo line 929, should be an inequality. The proof remains sound.

**Quality**
 - Line 831+: There is nothing guaranteeing that A+ and A- tend to 0. In fact, the term Beta_i only depends on W_q, W_k, v_q and v_k, none of which involve the positonal causal aware encoding.
 - In fact, if a first point has pointcarre embedding 0 and another is (a,0,0,0...) with a varying from 0 to 1, the changes to matrix R will only affect the first 2D block. This countradicts A tending to 0 when dp tends to infinity.
 - The same example shows that for those two points, the attention (line 246) may have large upper bound (lower bound by symmetry) despite causal generality.
 - Going further, since feature encoding v and positional encoding phi are computed separately, and since rotation matrices have determinant 1, for every feature embedding v1 and rotation matrix, there exists and embedding v2 (perhaps unused), such that the attention score is ||v1||. To do so, it is enough to invert R and multiply by v1. Relying only on a rotation matrix is not sufficient to guarantee that the causal proximity affects each attention scores. Attention score bounds may only be **probabilistic**.
 - Proposition A.8 also falls short of the stated goal. It relies on that vm and vn are of same mean, are independent, vm and delta are also independent, so their attention cancel out on average. The only term left is the attention of vm with itself, reasonably positive. The problem is that we would like that for any vm and vn, reasonably far from each other, vm+delta has higher attention score.

---

> ### Author Rebuttal · Authors · 2025-07-31
>
> We sincerely thank you for your insightful comments and careful check over our theoretical parts, which greatly improve the quality and clarity of our work.
> ***
> **Weak points Clarity 1**
>
> We appreciate the reviewer for pointing out that the term "angle vector" might be misleading. In our formulation, the word “angle” refers not to the polar coordinate of a point on the Poincaré disk, but to the **rotary phase** employed by RoPE. For every 2-D sub-space $(2t-1,2t)$ , the $t$-th element of $\varphi_{v}$ serves as the rotation phase in the rotation matrix $R(\varphi_{v})$. Thus, when computing the attention score (Eq. 25), the factor $R(\varphi_{v_m}-\varphi_{v_n})$ lets tokens $v_m$  and $v_n$ to be aware of the difference for *each coordinate* of their Poincaré embeddings and utilize this positional information during their training. To avoid confusion, we revise the description of $\varphi_{v}$ on Line 212 to read: "$\ldots$where $\varphi_{v}\coloneqq ce_v$ denotes a vector whose components are used as the rotation angles in the subsequent rotary embedding$\ldots$".
> ***
> **Weak points Clarity 2**
>
> The typo on line 929 has been corrected.
> ***
> **Weak points Quality 1&2**
>
> You are entirely correct in catching this mistake due to our negligence. As $d_p(e_{v_m}, e_{v_n})\rightarrow+\infty$, the correct attenuation bounds for $\mathcal{A}^+$ and $\mathcal{A}^-$ are $| \alpha_i^* | d \cos\left(\frac{2\pi}{d}\right) + \sum_{t=1}^d | \beta_i^{(t)} | $ and $-| \alpha_i^* | d \cos\left(\frac{2\pi}{d}\right) - \sum_{t=1}^d\lvert\beta_i^{(t)}\rvert$ , respectively. In fact, we realized these bounds intuitively are more sensible for several reasons:
>
> i) **Semantic component persists.** $ \alpha_i^* $ and $\beta_i^{(t)}$  encode the *semantic* similarity between tokens through $W_qv_m$ and $W_kv_n$. They are independent of positional information and therefore do not vanish with causal distance.
>
> ii) **Causal attenuation is still present.** The rotary phase affects only the $\alpha_i^*$ term via the cosine factor shown above, ensuring that, for a fixed semantic similarity, the attention score is attenuated when the Poincaré distance grows.
>
> iii) **Desirable behavior.** The attention score should not vanish for two features that are semantically related, even if they are causally distant. For instance, two genes may belong to the same protein family (high semantic similarity) but be regulated by entirely different pathways (large causal distance). Their attention score should remain non-zero to reflect their shared function, even as it is attenuated by their causal separation. We have corrected this mistake in the Remark A.3. section in our manuscript.
> ***
> **Weak points Quality 3**
>
> You are correct that the absolute magnitude of the attention score's bounds is not determined by causal generality alone, and can indeed be large. Our intention was not to claim that high generality guarantees a low attention score in all cases, but rather to describe its specific attenuating effect. We will clarify this in the manuscript and below:
>
> As shown in Eq. A.73, the two bounds depend on $ \alpha_i^* $, $\beta_i^{(t)}$, which encode the *semantic* similarity between tokens,  and causal distance $d_p(e_{v_m},e_{v_n})$ (in $C$) .  Thus, the attention score between $v_m$ and $v_n$ can remain significant even under extreme causal generality since causal generality is not the sole factor that determines the attention. The main point is, given fixed semantic similarity and causal generality, it is desirable that the attention score is attenuated when causal generality grows. Crucially, the raw attention scores are not the final weights used by the model. These scores are inputs to a **softmax function**, which normalizes them across all features in the sequence to produce a probability distribution. This means the **relative values** of the attention scores are more important than their absolute magnitudes. While a highly general feature might still produce a large raw score with a semantically similar neighbor, the causality-induced attenuation ensures this score is *relatively lower* than it would be for a more specific feature at the same causal distance. This relative difference is preserved and utilized by the softmax function to weigh the final feature representations.
> ***
> **Weak points Quality 4**
>
> This is a sharp theoretical observation. You are correct that for any feature encoding $v_1$ and rotation matrix $R$, it is mathematically possible to construct an encoding $v_2=R^Tv_1 $that perfectly cancels the rotational effect in their dot product. However, our framework's effectiveness relies on what the model is incentivized to learn during end-to-end training, not on this theoretical edge case. There are two main reasons why this is not a concern in practice:
>
> i) **Inductive Bias and the Learning Objective:** The core premise of CAPE is that injecting causal geometry provides a powerful **inductive bias**. That is, the feature encodings are not fixed but trained to minimize a loss on a downstream task, being rewarded for leveraging the information provided by the causal positional encodings to make better predictions. It is highly improbable that the optimal solution for minimizing the global loss would involve learning a complex set of feature embeddings that systematically "conspire" to cancel out this useful causal structural information. The model is far more likely to learn representations that benefit from the provided geometry.
>
> ii) **Empirical Validation from Ablation Studies:** Our experiments provide strong evidence that the model does, in fact, learn to use the causal information effectively. As shown in our ablation study (Table S4), the performance of the full CAPE model is substantially better than variants where the causal or rotary components are removed. This empirically demonstrates that the causal-rotary mechanism is not being ignored or counteracted by the learned feature embeddings but is critical to the model's success.
> ***
> **Weak points Quality 5**
>
> We are very grateful for this insightful critique. Our original intent with this proposition was to show that the rotary attention mechanism could distinguish genuine semantic differences from random noise. However, you have correctly identified the proposition's limitations. We agree that its proof relies on strong statistical assumptions that cause the attention terms to cancel out on average, and it does not guarantee that it is true for any pair of features. Upon reflection, we concur that this "in-expectation" result falls short of its goal. To improve the focus and rigor of our paper, we have removed Proposition A.8. This allows our theoretical analysis to center on the more impactful, instance-level guarantees of causal attenuation. We sincerely thank you for this comment, which has helped us to improve the quality of our paper.
> ***
> **Questions**
>
> We thank the reviewer for this thought-provoking suggestion. Indeed, in our current formulation, the function $\gamma$ is dependent only on the difference in positional encodings. While our method does not explicitly factor causal generality into the $\gamma$ function, we argue that this information is **implicitly encoded and available to the model**. A feature's causal generality is defined by its proximity to the origin of the Poincaré disk. This centrality is inherently reflected in its set of pairwise causal distances to all other features; a more general feature will, on average, be closer to all others. Since the model has access to all pairwise causal positional differences during training, it can therefore learn to recognize a feature's generality from this context. This is conceptually similar to how kernel methods can recover geometric properties from a matrix of pairwise similarities.
> ***
> **Limitations**
>
> We understand this limitation also refers to the same point you raised in the *Questions* section, which we have responded above. Please let us know if it refers to something else.

---

> > ### Comment · Reviewer_TuE7 · 2025-08-03
> >
> > Thank you for your detailed rebuttal. I greatly appreciate the clarity with which you explain the changes you will introduce to the paper. I believe all concerns of my review were adressed, and I stand corrected on Quality 3 and 4. The rationale behind the attenuation without convergence to 0 in Quality 1&2 is a core point to the main paper body, as it gives crutial understanding on the positional encoding properties.
> >
> > Concerning other reviews, I believe that the authors have adressed their concerns to a great extent. The only remaining main concern would be how different causal discovery and inference methods and assumptions would impact the performance of the encoding scheme and learned model. In my opinion, this would be very valuable in a data-centric evaluation, but remains out of the scope of what can be expected in a 10 page contribution.

---

> > > ### Author Response · Authors · 2025-08-03
> > >
> > > We sincerely thank you for your thoughtful engagement with our rebuttal and for the constructive feedback throughout the review process. We're glad our clarifications helped address the concerns raised. We also agree that exploring the impact of different causal discovery methods and modeling assumptions on CAPE's performance is an important direction for future research. While this lies beyond the scope of the current work, we believe it is a valuable extension, especially given CAPE’s modular design and compatibility with various structure learning algorithms. We plan to pursue this in future work!

---

### Decision · Program_Chairs · 2025-09-17

**Decision:**

Accept (poster)

**Comment:**

Capturing causation in transformers is an especially important direction, and using causation in positional encoding is an innovative approach.  The paper is technically sound, the reviewers and authors had a great discussion, and the reviewers are unanimous in recommending acceptance.